# Role of the primate ventral striatum as a neural hub bridging option valuation and action selection

Masafumi Nejime [1], Mengxi Yun [1], Yawei Wang[1], Takashi Kawai[1], Jun Kunimatsu [1,2], Hiroshi Yamada [1,2], Ken-ichi Inoue[3], Masahiko Takada [3] & Masayuki Matsumoto [1,2,3]

Making appropriate decisions relies on the brain's capacity to evaluate the expected outcomes of available options and select the most rewarding action. The ventral striatum and midbrain dopamine neurons have been implicated in the option valuation process, consistent with the brain's reinforcement learning theory in which these brain structures encode and update value representations of expected outcomes. Extending beyond this framework, we found that the dopamine–ventral striatum system plays a more proactive role in action selection. We recorded single-unit activity from ventral striatum neurons in macaque monkeys as they sequentially evaluated an option, decided whether to perform an action to choose it, and expressed that motor action. The activity of these neurons initially reflected the value of the option but gradually shifted to reflect monkey's action selection, as if the ventral striatum translates the value information into the action. Moreover, optogenetic facilitation of dopamine input to the ventral striatum as well as electrical stimulation of this region altered monkey's action selection. Our findings reveal a previously unappreciated function of the ventral striatum as a neural hub that bridges option valuation and action selection, and demonstrate the contribution of dopamine in the process leading to action selection within this region.

To select the most rewarding action in a given environment, the brain is thought to rely on principles of reinforcement learning[1,2]. A central concept in this framework is *value function*, which represents the expected reward value associated with each available option. This value representation is continuously updated based on the difference between actual and expected outcomes, a process driven by *reward prediction errors* (RPEs). RPE, which encodes the outcome difference, works as a teaching signal for the update. The more accurately the value function represents the true reward value of each option, the more effectively an agent can select the most beneficial action.

The striatum, particularly its ventral subdivision, and midbrain dopamine neurons have been identified as key neural substrates supporting the value representation and its update. Neurons in the ventral striatum encode expected reward values[3–6] and adjust these representations in accordance with past reward experiences[7,8], mirroring the value function computational mechanisms proposed by reinforcement learning theory. Midbrain dopamine neurons exhibit phasic increases in firing in response to outcomes that exceed expectations, and decreases when outcomes fall short, consistent with their role in signaling RPEs[9–13]. Since dopamine neurons project densely to the ventral striatum, RPE signals are transmitted to this region and are

[1]Institute of Medicine, University of Tsukuba, Tsukuba, Ibaraki, Japan. [2]Transborder Medical Research Center, University of Tsukuba, Tsukuba, Ibaraki, Japan. [3]Center for the Evolutionary Origins of Human Behavior, Kyoto University, Inuyama, Aichi, Japan. ✉e-mail: matsumoto.masayuki.4w@kyoto-u.ac.jp

thought to update value representations through dopaminergic modulation of synaptic plasticity[14].

According to the reinforcement learning framework, such precise value representations in the ventral striatum enable the brain to select the most valuable action in a given environment. However, a critical question remains: how does the brain link these internal value representations to the actual selection of action? A different line of research may provide insight into this question. As part of the limbic network, which integrates cortical and subcortical regions involved in reward processing[15,16], the ventral striatum projects to the globus pallidus[17–19], which in turn connects to the motor and premotor systems[20–22]. This anatomical organization has led to the idea that the ventral striatum serves as a limbic–motor interface, translating motivational and emotional signals into motor outputs[23–25]. Supporting this notion, previous studies have shown that the ventral striatum contributes to effortful motor behaviors, such as precision gripping in animals with spinal cord injuries, by integrating motivational drive with motor coordination[26].

The ventral striatum's role as a limbic–motor interface may also be relevant to decision-making contexts. The functional linkage between the limbic and the motor systems suggests that the ventral striatum not only represents the value of available options but may also help bridge the value representation and action selection. It could, for instance, provide the premotor system, including prefrontal and parietal cortical areas, with information about the value of available options and support its selection of the most valuable action. Alternatively, it may play a more active role in converting the value information directly into the decision through interaction with the premotor system. To explore these possibilities, we employed a value-based decision-making task in which macaque monkeys evaluated an option, decided whether to perform an action to select it, and executed that action. We recorded single-unit activity from ventral striatum neurons in monkeys performing this task. The activity of these neurons reflected the value of the option immediately after it was presented. Over time, however, their activity progressively shifted to reflect monkey's action selection (i.e., whether to perform the action to choose the option). This neural dynamics suggests that the value signal in the ventral striatum gradually evolved into the action selection signal as the decision was being made. Moreover, optogenetic facilitation of dopamine input to the ventral striatum as well as electrical stimulation of this region altered monkey's action selection, demonstrating a causal role of the ventral striatum and its dopaminergic modulation in decision-making. Our findings reveal a previously unappreciated function of the ventral striatum as a neural hub that bridges option valuation and action selection. They further highlight the contribution of dopamine to the process leading to action selection within this region.

## Results

### Value-based decision-making task and monkey's behavior

To examine the roles of the dopamine-ventral striatum system in option valuation and action selection, we employed a value-based decision-making task in which macaque monkeys evaluated an option and then immediately decided whether to choose it (Fig. 1a). We trained three monkeys (monkey E: *Macaca mulatta*, male, 11.7 kg, 15 years old; monkey A: *Macaca mulatta*, male, 8.4 kg, 11 years old; monkey M: *Macaca fuscata*, female, 6.3 kg, 6 years old) to perform this task. At the beginning of each trial, the monkey gazed at a central fixation point and pressed a button in front. Six visual objects were associated with different amounts of a liquid reward (value 1 = 0.12 ml, value 2 = 0.18 ml, value 3 = 0.24 ml, value 4 = 0.30 ml, value 5 = 0.36 ml, value 6 = 0.42 ml), and two of these objects were sequentially presented. The first object was presented as an option, and the monkey was required to decide whether to choose or not to choose the first object. If the monkey released the button during the presentation of

the first object, this was interpreted as a decision to choose that object. Conversely, if the monkey kept the button pressed down, it was considered a decision not to choose the object. Thus, the monkey's decision was regarded as whether to perform a specific action (i.e., button release) to choose the first option. After this decision, the second object was presented. If the monkey had decided to choose the first object, the animal was unable to choose the second object, even if the second object was better than the first object. Then, the reward associated with the first object was delivered. If the monkey had decided not to choose the first object, the animal had to release the button within the presentation of the second object. Then, the reward associated with the second object was delivered.

We confirmed that the three monkeys decided whether to release the button to choose the first object based on its value (Fig. 1b). The higher the value of the first object, the more likely the monkeys were to choose it (i.e., the logistic regression slope between the value and the choice rate was significantly larger than zero; monkey E: $n = 54$ sessions, mean $\pm$ SEM = $11.5 \pm 2.4$, $z = 7.3$, $p = 2.4 \times 10^{-13}$; monkey A: $n = 71$ sessions, mean $\pm$ SEM = $13.5 \pm 3.3$, $z = 6.4$, $p = 1.3 \times 10^{-4}$; monkey M: $n = 18$ sessions, mean $\pm$ SEM = $3.3 \pm 0.2$, $z = 3.7$, $p = 1.9 \times 10^{-4}$; two-tailed Wilcoxon signed-rank test). The latency of button release to choose the first object was also influenced by the value of the first object (Fig.1c). It significantly decreased as the object value increased (i.e., the regression slope between the value and the latency was significantly smaller than zero; monkey E: $n = 54$ sessions, mean $\pm$ SEM = $-27.6 \pm 1.0$, $z = -7.3$, $p = 2.4 \times 10^{-13}$; monkey A: $n = 71$ sessions, mean $\pm$ SEM = $-20.7 \pm 2.6$, $z = -5.6$, $p = 2.7 \times 10^{-8}$; monkey M: $n = 18$ sessions, mean $\pm$ SEM = $-18.0 \pm 2.3$, $z = -3.7$, $p = 2.0 \times 10^{-4}$; two-tailed Wilcoxon signed-rank test).

Because the task design allowed the monkeys to either commit early to the first object or wait for the second one, their decisions could, in principle, be influenced by individual risk attitudes. For example, risk-seeking monkeys might reject the first object more often in anticipation of receiving a larger reward from the second object. We quantified risk attitude as the value of the first object that was chosen with 50% probability. A higher value indicated a stronger risk-seeking tendency. However, no significant differences in risk attitude were observed across the three monkeys ($n = 54$, 71, and 18 sessions for monkeys E, A and M, respectively, $p = 0.07$, $F = 2.78$; one-way ANOVA) (Supplementary Fig. 1a). This indicates that the observed choice patterns were not influenced by individual differences in risk preference but were instead a consistent consequence of option valuation and action selection within the task.

Notably, the button-release latency was significantly longer when the monkey released the button for the first object than when the monkey released the button for the second object (monkey E: $n = 54$ sessions, $df = 322$, $F = 873.24$, $p = 1.0 \times 10^{-93}$; monkey A: $n = 71$ sessions, $df = 424$, $F = 399.61$, $p = 4.1 \times 10^{-63}$; monkey M: $n = 18$ sessions, $df = 106$, $F = 596.89$, $p = 2.4 \times 10^{-45}$; one-way ANOVA) (Fig. 1c). This result is consistent with our task design, which required the monkey to make a decision when releasing the button for the first object but only to respond reflexively to the appearance of the second object. Even for the second object, the button-release latency decreased as its value increased, although this trend was not statistically significant in one monkey (regression slope; monkey E: $n = 54$ sessions, mean $\pm$ SEM = $-16.4 \pm 0.7$, $z = -7.3$, $p = 2.4 \times 10^{-13}$; monkey A: $n = 71$ sessions, mean $\pm$ SEM = $-11.7 \pm 1.0$, $z = -6.3$, $p = 2.8 \times 10^{-10}$; monkey M: $n = 18$ sessions, mean $\pm$ SEM = $-1.5 \pm 1.2$, $z = -1.0$, $p = 0.33$; two-tailed Wilcoxon signed-rank test) (Fig. 1c). These data indicate that the monkey's decision was influenced by the value of the objects, and that the monkeys properly linked option valuation to action selection in our value-based decision-making task.

We also found that the monkey's decision was influenced by recent experience. Specifically, the value of the second object in the previous trial negatively predicted whether the monkey would choose

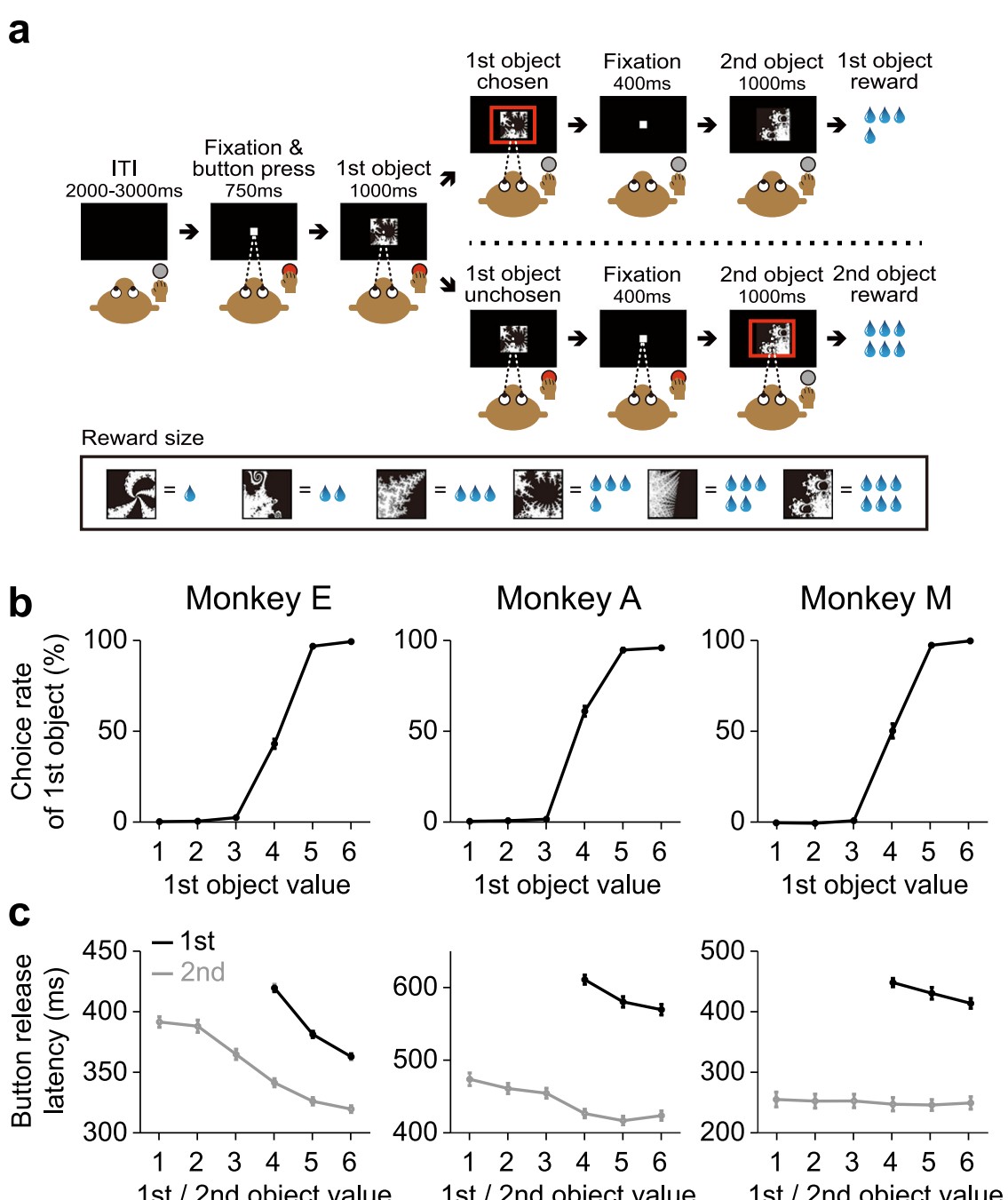

**Fig. 1 | Value-based decision-making task and monkey's behavior. a** Value-based decision-making task. ITI, intertrial interval. **b** Choice rate of the first object in monkey E (*n* = 54 sessions) (left), monkey A (*n* = 71 sessions) (center) and monkey M (*n* = 18 sessions) (right). **c** Latency of button release to choose the first object (black plots) and to respond to the appearance of the second object (gray plots). The number of sessions for each monkey is the same as in (**b**). Error bars in (**b**) and (**c**) indicate SEM, which are very small and hidden in some cases. (**a**) is adapted from Yun et al., Science Advances (2020)[28], licensed under CC BY-NC 4.0 (https://creativecommons.org/licenses/by-nc/4.0/). The figure has been modified from the original.

the first object in the current trial. A logistic regression analysis revealed a weak but significantly negative regression coefficient between the value of the previous second object and the choice rate of the current first object (*n* = 143 sessions, *z* = −7.2, *p* = 8.4 × 10⁻¹³; two-tailed Wilcoxon signed-rank test), while the regression coefficient for the current first object value was significantly positive (*n* = 143 sessions, *z* = 10.4, *p* = 3.2 × 10⁻²⁵; two-tailed Wilcoxon signed-rank test) (Supplementary Fig. 1b). Thus, the higher the value of the previous second object, the less likely the monkeys were to choose the current first object. This opposing effect of the current and previous values suggests that when the previous second object had been highly

rewarding, the monkeys anticipated the possibility of another valuable second object and were therefore more inclined to reject the current first object.

## Neural signatures of option valuation and action selection in the ventral striatum

While monkeys E and A were performing the decision-making task, we recorded single unit activity from 125 neurons in the ventral striatum (monkey E: *n* = 54 neurons; monkey A: *n* = 71 neurons) (Supplementary Fig. 2). In this task, option valuation, action selection, and action execution occurred in close succession during the presentation of the

first object, providing an opportunity to examine the neural mechanisms underlying each process and the transitions between them. We analyzed neuronal activity during this period.

Although the ventral striatum has attracted attention for its role in signaling reward value information, we found that ventral striatum neurons did not simply represent reward value in our decision-making task. For example, a representative neuron shown in Fig. 2a increased the activity as the value of the first object increased but only when the monkey would decide to choose this object (i.e., only when the monkey decided to release the button to obtain the reward associated with this object), indicating that the activity reflected not only the first object value but also the monkey's action selection. These neurons seem to represent a decision variable called "chosen value", which has been reported to be signaled in prefrontal areas[27], because they represented the value of the "chosen" first object. The activity of some other recorded neurons reflected the monkey's action selection but not the first object value. These neurons changed their activity depending on whether the monkey would decide to release the button to choose the first object (see Fig. 2b for a representative neuron). Especially, when the value of the first object was 4, the monkey sometimes chose the object and sometimes did not. In this value condition, the activity changed depending on the action selection regardless of the same value.

As mentioned above, many of the recorded ventral striatum neurons did not simply represent the value. Their activity reflected the monkey's action selection to varying degrees for each neuron. We further found that their activity changed over time. For example, a representative neuron shown in Fig. 2c increased its activity as the first object value increased, independent of the monkey's action selection, immediately after the onset of the first object, suggesting that the activity of this neuron simply reflected the value during the early period. However, its activity gradually shifted to reflect not only the first object value but also the monkey's action selection. Its activity increased as the first object value increased, but only when the monkey decided to release the button to choose the object during the later period.

To statistically characterize the activity of ventral striatum neurons, we conducted a model comparison analysis, in which we fitted the activity of each neuron with two models representing value and choice (Fig. 2d). We did not use a conventional multiple regression approach with value and choice as predictors in this main analysis, because these variables were highly correlated (see Fig. 1b), making it difficult to obtain reliable parameter estimates (called "multicollinearity" problem). The model comparison analysis, which is not affected by this issue, has been successfully used in our previous study to characterize neuronal activity in other brain regions[28]. In this analysis, the value model assumes that neuronal activity reflects the value of the first object, while the choice model assumes that neuronal activity reflects whether the monkey chose to release the button to select the first object. We compared their coefficients of determination (R-square) for each neuron (Fig. 2e). If neurons showed a significantly better fit with the value model than the choice model, their activity was considered to be more largely modulated by the value of the first object ($p < 0.05$; two-tailed Monte Carlo test) (blue area in Fig. 2e, hereafter called "value-modulated" signal). If neurons showed a significantly better fit with the choice model than the value model, their activity was considered to be more largely modulated by the monkey's action selection ($p < 0.05$; two-tailed Monte Carlo test) (red area in Fig. 2e, hereafter called "choice-modulated" signal). If neurons did not show a significantly better fit with either model ($p > 0.05$; two-tailed Monte Carlo test) but showed significant fits with both models ($p < 0.05$; two-tailed F test), their activity was considered to exhibit an intermediate modulation between the value and the choice models (cyan area in Fig. 2e, hereafter called "intermediate" signal). To examine the temporal profile of neuronal activity, this model

comparison analysis was conducted throughout the presentation of the first object using a 150-ms sliding window with a 1-ms step. Consequently, we identified 42 neurons with the value-modulated signal, 56 neurons with the choice-modulated signal, and 81 neurons with the intermediate signal (Fig. 2f, see also Supplementary Fig. 3a and 3b for individual monkeys). Notably, some of the recorded neurons exhibited two or three types of signals because these neurons represented distinct signals for different periods during the object presentation.

To confirm the validity of this classification, we conducted an additional analysis using "ridge regression", which can partly mitigate the issue of multicollinearity among predictors. In this model, the firing rate of each neuron was predicted by the value of the first object (*value*) and whether the monkey released the button to choose the object (*choice*), with an additional regularization term in the loss function that penalizes large coefficient estimates arising from correlated predictors. This regularization stabilizes parameter estimation and enables a more reliable valuation of each predictor's influence. We fitted the ridge model separately to the activity of each neuron classified as value- or choice-modulated based on the model comparison analysis. The activity was measured within the time window during which each neuron exhibited the value- or choice-modulated signal. The neuronal activity classified as value-modulated showed significantly larger regression coefficients for *value* than did the activity classified as choice-modulated ($z = 3.1$, $p = 2.3 \times 10^{-3}$; two-tailed Wilcoxon rank-sum test), whereas the neuronal activity classified as choice-modulated showed significantly larger coefficients for *choice* than did the activity classified as value-modulated ($z = 3.2$, $p = 1.4 \times 10^{-3}$; two-tailed Wilcoxon rank-sum test) (Supplementary Fig. 4). These results demonstrate that our model comparison approach reliably dissociates the value- and choice-modulated signals, despite their inherent correlation. The consistency of these findings with the ridge regression further supports the robustness of our model comparison approach.

Using another ridge regression, we tested whether the modulation of each neuron could be accounted for by the variation in reaction time (i.e., the latency of button release). Here, the firing rate of each neuron was predicted by the value of the first object (*value*) and the button-release latency (*latency*), which were also correlated (Fig. 1c). We excluded the predictor of whether the monkey released the button to choose the first object (*choice*) from this analysis, because the button-release latency existed only when the monkey released the button (i.e., when the monkey did not release the button, making it impossible to include such trials in the regression analysis). Whereas the proportion of neurons with a significant regression coefficient for *value* was significantly higher than chance for all calculation time windows (each 150-ms window after the first object onset) except immediately after the onset ($p > 0.05$; two-tailed Monte Carlo test) (Supplementary Fig. 5a), the proportion of neurons with a significant regression coefficient for *latency* was very close to chance over time, slightly above for some time windows ($p < 0.05$; two-tailed Monte Carlo test) but below for others ($p > 0.05$; two-tailed Monte Carlo test) (Supplementary Fig. 5b). These results suggest that the influence of button-release latency on neural activity, if any, was minimal.

We calculated the latencies of the value-modulated, choice-modulated, and intermediate signals for each neuron (Fig. 2g, see also Supplementary Fig. 3c and d for individual monkeys). The order of these latencies corresponded to the time course of the decision-making process in which the monkey evaluated the first object and then decided whether to release the button to choose it. That is, the value-modulated signal appeared earliest, followed by the intermediate signal (statistical significance of the latency difference between the value-modulated signal and the intermediate signal; $z = 2.1$, $p = 0.03$; two-tailed Wilcoxon rank-sum test). Then, the choice-modulated signal appeared (statistical significance of the latency difference between the intermediate signal and the choice-modulated

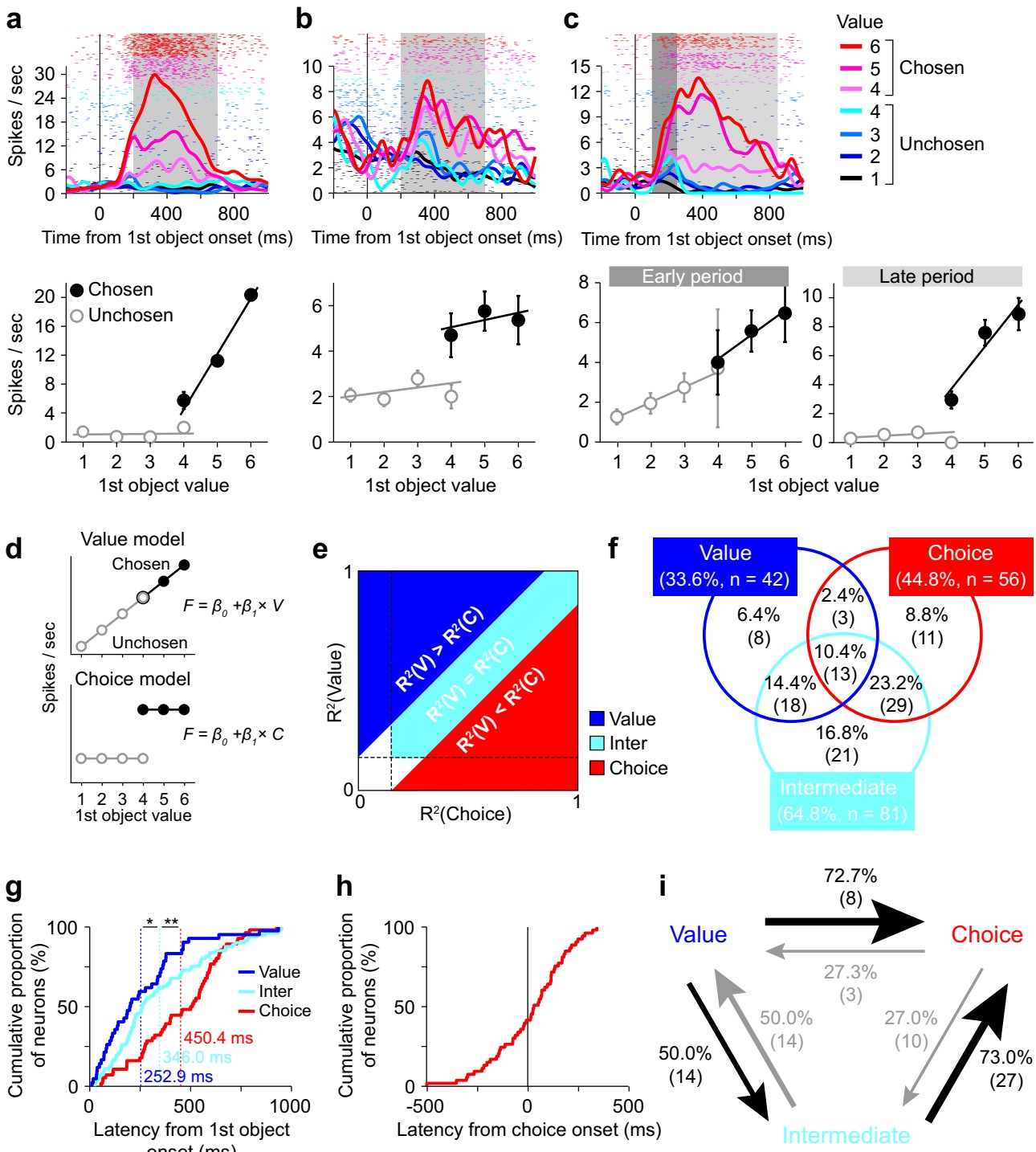

**Fig. 2 | Neural signatures of option valuation and action selection in the ventral striatum. a–c** Activity of three representative ventral striatum neurons ($n = 298$ trials in (**a**); $n = 347$ trials in (**b**); $n = 345$ trials in (**c**)). Top, spike density functions (SDFs) and raster plots aligned at the first object onset are shown for each value and for chosen and unchosen trials. Gray shaded areas indicate the time window to calculate the magnitude of neuronal activity. Bottom, magnitude of neuronal activity plotted against the first object value shown for chosen (filled black circles) and unchosen trials (open gray circles). Error bars indicate SEM. **d** Value and choice models in the model comparison analysis. **e** Schematic diagram of the model comparison analysis to identify value-modulated (blue), intermediate (cyan), and choice-modulated signals (red). **f** Proportion of neurons with value-modulated, intermediate, and choice-modulated signals. Numbers in parentheses indicate the number of neurons. **g** Cumulative histograms of the latencies of the value-modulated, intermediate, and choice-modulated signals aligned at the onset of the first object. The averaged latencies were described. Single and double asterisks indicate a significant difference between the latencies ($p = 0.03$ and $0.004$, respectively; two-tailed Wilcoxon signed-rank test). **h** Cumulative histograms of the latencies of the choice-modulated signal aligned at the onset of the monkey's choice action (i.e., button release). **i** Transition probability among the value-modulated, intermediate, and choice-modulated signals of the neurons that represent at least two of these signals. The directions of signal transition are indicated by an arrow. The size of the arrow represents the size of the transition probability. Numbers in parentheses indicate the number of neurons.

signal; $z = 2.9$, $p = 0.004$; two-tailed Wilcoxon rank-sum test). Furthermore, many of the neurons with the choice-modulated signal exhibited an earlier onset of their signal than the onset of the monkey's action execution (i.e., the onset of button release) (Fig. 2h). These results suggest that the signal dynamics of ventral striatum neurons reproduced the process from option valuation to action selection in value-based decision-making, and that this dynamics occurred within the period during which a decision was being made.

The signal dynamics corresponding to the time course of the decision-making process was observed even in individual neurons. For example, the representative neuron shown in Fig. 2c first represented the value-modulated signal, then the intermediate signal. Here, we calculated the transition probability among the value-modulated, intermediate and choice-modulated signals of the neurons that represented more than two of these signals (Fig. 2i). For the transition between the value-modulated and intermediate signals, the value-to-intermediate and intermediate-to-value transition probabilities were the same (i.e., 50%). However, for the transition between the intermediate and choice-modulated signals and the transition between the value-modulated and choice-modulated signals, the intermediate-to-choice and the value-to-choice transition probabilities were more than 70%. Furthermore, the time at which the transition occurred (i.e., the latency of the later signal) was shorter for the value-to-intermediate transition compared to the intermediate-to-choice and value-to-choice transitions. However, these differences did not reach statistical significance, possibly due to the small sample size (value-to-intermediate: $n = 14$, mean ± SD = 491 ± 209 ms; intermediate-to-choice: $n = 27$, mean ± SD = 497 ± 191 ms; value-to-choice: $n = 8$, mean ± SD = 543 ± 192 ms; value-to-intermediate vs. intermediate-to-choice: $z = -0.11$, $p = 0.91$; value-to-intermediate vs. value-to-choice: $z = -0.72$, $p = 0.47$; two-tailed Wilcoxon rank-sum test) (Supplementary Fig. 6). These signal transitions correspond to the time course of the decision-making process that computes the value of the first object and then decides whether to choose it based on the value information. The equal transition probability between the value-modulated and intermediate signals suggests that these two states may not form a strictly sequential process but could reflect bidirectional or overlapping computations during value valuation. This transition is therefore likely to be coordinated at the population level rather than within single neurons.

Although the neurons with the choice-modulated signal seem to represent whether the monkey would decide to choose or not to choose the first object, it remains possible that these neurons reflect a simple motor action (i.e., button release) associated with choosing the first object. To test this possibility, we recorded the activity of 56 neurons with the choice-modulated signal while the monkey was performing a simple button-release motor task (Fig. 3a). In this task, the monkey was not required to make decisions and was required just to release the button. We observed that only 7 of these 56 neurons exhibited a significant modulation around the button release onset in the button-release task ($p < 0.05$; two-tailed Wilcoxon signed-rank test). On average, these 56 neurons exhibited a smaller modulation in the button-release task compared to the decision-making task (see Fig. 3b and c for 33 neurons with positive modulation and Supplementary Fig. 7a, b for 23 neurons with negative modulation). The difference in modulation magnitude between the tasks was statistically significant (neurons with positive modulation: $z = 2.46$, $p = 0.014$; neurons with negative modulation: $z = 2.10$, $p = 0.036$; two-tailed Wilcoxon signed-rank test) (Fig. 3d and Supplementary Fig. 7c). These results suggest that the choice-modulated signal do not merely reflect the motor action of button release.

To understand the neural mechanisms underlying option valuation, action selection, and their transition, we analyzed neuronal activity encoding the value of the first object and whether the monkey decided to release the button to choose it during the decision-making period (i.e., during the presentation of the first object). However, other neural processes contributing to decision formation might also occur

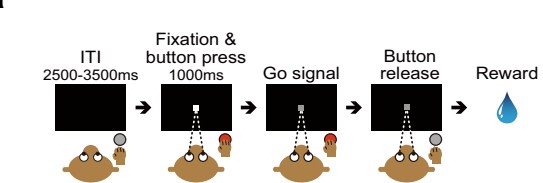

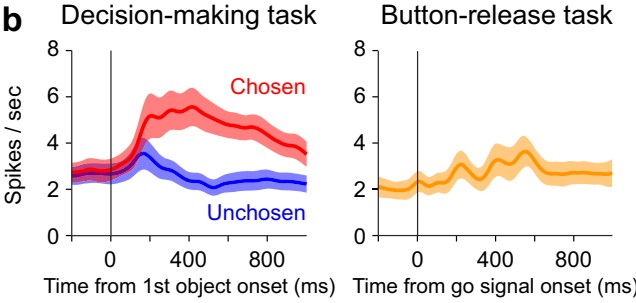

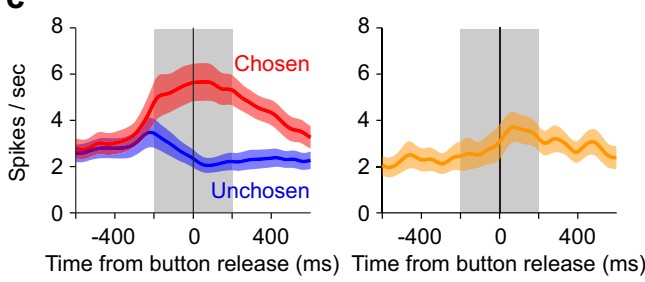

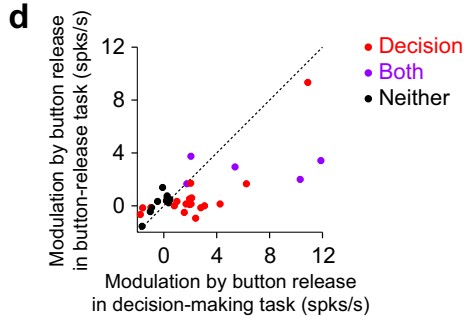

**Fig. 3 | Neuronal modulation evoked by simple motor action. a** Simple button-release motor task. ITI, intertrial interval. **b, c** Averaged activity of 33 neurons representing choice-modulated signal with a positive modulation aligned at the first object onset (**b**) and the button release onset (**c**) in the decision-making task (left) and the button-release task (right). Spike density functions (SDFs) are shown for chosen (red) and unchosen trials (blue) in the decision-making task. To calculate the SDF in unchosen trials (i.e., trials in which the monkey did not release the button), we randomly selected the onsets of button release in chosen trials with replacement and assigned the onsets to the unchosen trials. Color-shaded areas indicate SEM. Gray-shaded areas indicate the time window used to calculate the magnitude of neuronal activity. **d** Comparison of the modulation magnitude evoked by button release between the decision-making task (x-axis) and the button-release task (y-axis) ($n = 33$ neurons). Red and orange plots represent neurons with significant modulation exclusively in the decision-making task and button-release task, respectively ($p < 0.05$; two-tailed Wilcoxon signed-rank test); however, no orange plots are present here (see Supplementary Fig. 7). Purple plots denote neurons with significant modulation in both tasks ($p < 0.05$; two-tailed Wilcoxon signed-rank test), while black plots represent neurons with no significant modulation in either task ($p > 0.05$; two-tailed Wilcoxon signed-rank test). (**a**) is adapted from Yun et al., Science Advances (2020)[28], licensed under CC BY-NC 4.0 (https://creativecommons.org/licenses/by-nc/4.0/). The figure has been modified from the original.

in the ventral striatum during the same period. For example, we observed that not only the value of the first object, but also the value of the second object in the previous trial influenced the monkey's action selection (Supplementary Fig. 1b), suggesting that valuation or maintenance of the previous second object might also take place in the ventral striatum during decision making. To test this possibility, we conducted another ridge regression analysis in which the firing rate of each neuron was predicted by the value of the previous second object (*previous value*), as well as the value of the current first object (*current value*) and whether the monkey chose the first object (*choice*). As expected from the model comparison analysis, the proportions of neurons with significant regression coefficients for *current value* and *choice* increased after the first object onset and became significantly higher than chance ($p < 0.05$, two-tailed Monte Carlo test) (Supplementary Fig. 8a and b). On the other hand, the proportion of neurons with a significant regression coefficient for *previous value* was close to the chance level over time, slightly above for some time windows ($p < 0.05$; two-tailed Monte Carlo test) but below for others ($p > 0.05$; two-tailed Monte Carlo test) (Supplementary Fig. 8c), suggesting that the ventral striatum may not strongly represent or maintain the value of the previous second object. However, at the behavioral level, the influence of the previous second object value on the monkey's choices was much weaker than that of the current first object value. Therefore, it may be difficult to detect neuronal activity corresponding to such a weak behavioral effect of the previous second object value in the ventral striatum.

So far, we have focused on the neuronal modulation evoked by the first object and identified ventral striatum neurons with value-modulated, intermediate, and choice-modulated signals. How do these neurons with different types of signals respond to the second object? We did not observe clear differences in their modulations evoked by the second object across the three neuron groups (Supplementary Fig. 9). When the monkey did not choose the first object, i.e., when the second object was available and predicted the reward outcome, neuronal activity in all three groups reflected the value of the second object (Supplementary Fig. 9a–f). The magnitude of this modulation, measured as the regression coefficient between neuronal activity and the second object value, did not differ significantly across the three groups (neurons with positive modulation: $p = 0.41$, $F = 0.90$; neurons with negative modulation: $p = 0.41$, $F = 0.89$; one-way ANOVA) (Supplementary Fig. 9g and h). In contrast to the clear modulation evoked by the available second object, when the monkey chose the first object, i.e., when the second object was unavailable and no longer predicted reward, neurons of all groups showed little modulation (Supplementary Fig. 9a–h). Together, these results suggest that ventral striatum neurons flexibly change their roles depending on context, such as free-choice situations (when the monkey decided whether to choose the first object) versus forced-choice situations (when the monkey had no alternative but to obtain the reward associated with the second object).

### Effect of electrical stimulation of the ventral striatum on monkey's action selection

We next examined the causal contribution of neuronal activity in the ventral striatum to the monkey's action selection using monkeys E and A. We electrically stimulated the ventral striatum during the presentation of the first object, coinciding with the period when the monkey was making a decision (Fig. 4a and b). During this period, we found that ventral striatum neurons exhibited value-modulated, intermediate, and choice-modulated signals, which dynamically evolved as the decision process progressed from option valuation to action selection. The stimulation was applied in half of the trials, while the remaining trials served as controls without stimulation. We conducted the electrical stimulation experiment at 56 sites of the ventral striatum (monkey E: $n = 20$ sites; monkey A: $n = 36$ sites) and observed

that the electrical stimulation influenced the monkey's action selection. At some stimulation sites, the monkey was more likely to release the button to choose the first object in stimulation trials compared to non-stimulation trials (Fig. 4c for a representative stimulation site). At other sites, the monkey was less likely to release the button for the choice in stimulation trials (Fig. 4d for a representative stimulation site). Notably, these positive and negative effects were particularly evident in trials in which the first object had a medium value (i.e., value 4). Under this value condition, the monkey's decision fluctuated, sometimes selecting the first object and sometimes not. The stimulation sites that positively or negatively influenced the choice rate were distributed throughout the ventral striatum without clustering in any specific region (Supplementary Fig. 10a). However, the negative effect was slightly but significantly stronger in the posterior portion (correlation between Δchoice rate and anterior-posterior position; $n = 56$ sites, $r = 0.328$, $p = 0.014$; two-tailed Pearson's correlation test) (Supplementary Fig. 10b).

To statistically validate the effect of electrical stimulation on the monkey's action selection, we calculated the difference in choice rate between stimulation and non-stimulation trials (Δchoice rate) for each object value. In trials in which the first object value was 4, both positive and negative effects were observed, so the difference in choice rate did not significantly differ from zero as population ($z = 0.90$, $p = 0.37$; two-tailed Wilcoxon signed-rank test) (Fig. 4e). We confirmed that the direction of these positive and negative effects was consistent across the early and late halves of the trials for each stimulation site (Supplementary Fig. 11), suggesting that these effects were not due to random trial-by-trial fluctuations in the monkey's action selection. We calculated the absolute difference in choice rate, which reflects the net effect of electrical stimulation on the monkey's action selection (even if positive and negative effects are mixed), and observed that this absolute difference was significantly larger when the first object value was 4 compared with when the value was other than 4 ($n = 56$ sites, $df = 335$, $F = 74.9$, $p = 3.0 \times 10^{-52}$; one-way ANOVA and post-hoc test) (Fig. 4f, see also Supplementary Fig. 12a and b for individual monkeys). Furthermore, the distribution width of the choice rate difference, which also reflects the net effect, was significantly larger than the chance-level distribution width calculated by shuffling stimulation and non-stimulation trials when the first object value was 4 ($p = 0.006$; two-tailed Monte Carlo test) (Fig. 4g, see also Supplementary Fig. 12c and d for individual monkeys). On the other hand, when the first object value was other than 4, the distribution width was not significantly different from the chance-level distribution width ($p > 0.05$; two-tailed Monte Carlo test). These results suggest that electrical stimulation of the ventral striatum influenced the monkey's action selection specifically when the monkey was uncertain about choosing the medium-value option.

Although we observed that electrical stimulation of the ventral striatum influenced the choice rate of the first object when its value was 4, it remains possible that this effect could be attributed to the impact of the electrical stimulation on motor action rather than decision-making. The monkey's decisions were indicated by either releasing the button or keeping the button pressed down, so it is conceivable that the stimulation influenced the hand movements, which in turn altered the choice rate. To test this possibility, we conducted an electrical stimulation experiment using the simple button-release motor task, in which the monkey was required only to release the button and did not need to make decisions. Electrical stimulation with the same parameters (including duration, current and frequency) was applied to the ventral striatum from 1000 ms to 0 ms before the go signal in half of the trials, while no stimulation was delivered in the other half (Supplementary Fig. 13a). We calculated the difference in button-release latency between stimulation and non-stimulation trials to quantify the effect of electrical stimulation on the motor action. The latency difference was not significantly different from zero as

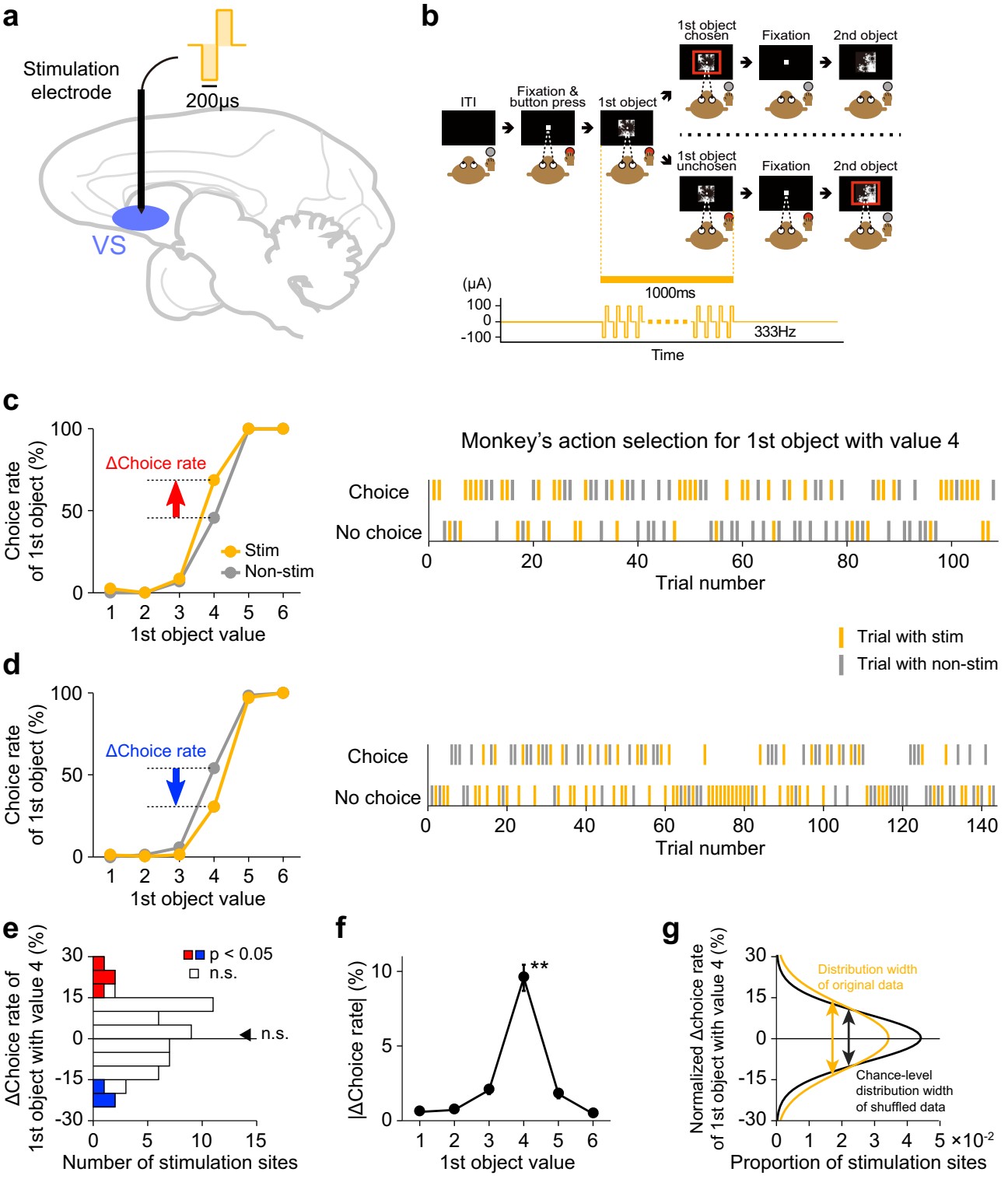

population ($z = -0.45$, $p = 0.65$; two-tailed Wilcoxon signed-rank test) (Supplementary Fig. 13b). Notably, there was no significant correlation between the difference in button-release latency and the difference in choice rate for the first object with value 4 ($n = 56$ sites, $r = -0.13$, $p = 0.34$; two-tailed Pearson's correlation test) (Supplementary Fig. 13c), indicating that the effect of electrical stimulation on the motor action (if any) and the effect on the monkey's action selection were independent. These results suggest that the effect of electrical stimulation of the ventral striatum on the monkey's action selection cannot be explained by its impact on the motor action. Rather,

neuronal activity in the ventral striatum appears to play a causal role in value-based decision-making.

Even in the decision-making task, electrical stimulation did not clearly affect button-release latency (Supplementary Fig. 14a). Although the latency difference between stimulation and non-stimulation trials reached marginal significance at the population level ($n = 56$ sites, $z = -2.45$, $p = 0.014$; two-tailed Wilcoxon signed-rank test), the difference was small, and only one of the 56 stimulation sites showed a significant effect ($p < 0.05$; two-tailed Wilcoxon rank-sum test). Moreover, there was no significant correlation between the

**Fig. 4 | Effect of electrical stimulation of the ventral striatum on monkey's action selection. a** Schematic diagram of the electrical stimulation experiment. VS, ventral striatum. **b** Electrical stimulation period during the value-based decision-making task. ITI, intertrial interval. **c, d** Effect of electrical stimulation at two representative sites on the monkey's choice of the first object ($n = 906$ trials in (**c**); $n = 1262$ trials in (**d**)). Left, choice rate of the first object. Yellow and grey plots represent the choice rates in stimulation and non-stimulation trials, respectively. Red and blue arrows indicate the difference in the choice rates between stimulation and non-stimulation trials (Δchoice rate). Right, raster plots showing, on a trial-by-trial basis, whether the monkey released the button to choose the first object during the session. Only trials in which the first object value was 4 were included. The upper row indicates trials in which the monkey chose the first object, and the lower row indicates trials in which it did not. Yellow and gray marks represent stimulation and non-stimulation trials, respectively. **e** Distribution of Δchoice rates

of the first object with value 4 ($n = 56$ sites). Red and blue bars represent Δchoice rates that are significantly positive and negative, respectively, compared to zero ($p < 0.05$; two-tailed chi-square test). The black arrow indicates the mean Δchoice rate. n.s. denotes no significance ($p = 0.37$; two-tailed Wilcoxon signed-rank test). **f** Mean absolute Δchoice rate for each first object value ($n = 56$ sites). Double asterisk denotes a significant difference ($p = 3.0 \times 10^{-52}$; one-way ANOVA and post-hoc test). Error bars indicate SEM, which are very small and hidden in most cases. **g** Gaussian functions fitted to the Δchoice rate distribution of the original data (yellow curve) and the chance-level distribution of shuffled data (black curve). The centers of the Gaussian functions are normalized to zero. Each arrow indicates the width of each distribution. (**b**) is adapted from Yun et al., Science Advances (2020)[28], licensed under CC BY-NC 4.0 (https://creativecommons.org/licenses/by-nc/4.0/). The figure has been modified from the original.

button-release latency difference in the decision-making task and the choice-rate difference for the first object with value 4 ($n = 56$ sites, $r = 0.08$, $p = 0.54$; two-tailed Pearson's correlation test) (Supplementary Fig. 14b). These results suggest that electrical stimulation of the ventral striatum might weakly influence the latency to reach a decision, but its impact (if any) is minimal and independent of which action the monkey selects.

## Effect of optogenetic facilitation of dopamine input to the ventral striatum on monkey's action selection

One of the main sources of input to the ventral striatum originates from midbrain dopamine neurons[29,30]. Within the framework of reinforcement learning in the brain, these neurons are thought to encode RPEs[9–13] and contribute to updating value representations in the ventral striatum[14]. Such reinforcement effects typically occur when animals obtain a reward (i.e., after a decision has been made) and shape future decision-making. However, our recent findings show that dopamine neurons are also activated at the time of option presentation (i.e., prior to decision-making) and encode multiple decision-related variables, including the value of the option and the monkey's action selection[28]. The ventral striatum likely receives these dopamine signals while a decision is being made. To test whether these dopamine signals influence decision-related processing in the ventral striatum, we conducted optogenetics experiments in which dopamine projections to the ventral striatum were selectively stimulated (Fig. 5a).

To deliver a red-light-sensitive channelrhodopsin gene (*ChRmine*) into dopamine neurons, we injected an adeno-associated virus vector (AAV2.1-hTHp2.6S-ChRmine-HA) into the substantia nigra pars compacta (SNc) and the ventral tegmental area (VTA) of monkeys E and M. We subsequently confirmed that dopamine neurons near the injection sites were activated when these neurons were stimulated with a 635-nm red laser light via optrode (an optic fiber coupled with a recording electrode) (Fig. 5c). At the end of the optogenetics experiments, we histologically verified the expression of the ChRmine-HA chimeric protein in dopamine neuron cell bodies (Fig. 5d) and dopamine projections in the ventral striatum (Fig. 5e).

In the optogenetics experiments, a 635-nm red laser light was delivered into the ventral striatum to stimulate dopamine projections during the presentation of the first object, coinciding with the period when the monkey was making a decision (Fig. 5b). Laser light stimulation was applied in half of the trials, while the remaining trials served as controls without stimulation. This stimulation schedule was the same as that of the electrical stimulation experiment. We conducted the optogenetics experiment at 50 sites within the ventral striatum (monkey E: $n = 32$ sites; monkey M: $n = 18$ sites) and observed that the laser light stimulation influenced the monkey's action selection.

The laser light stimulation of dopamine projections in the ventral striatum evoked effects similar to those of electrical stimulation of the ventral striatum. At some stimulation sites, the monkey was more likely to release the button to choose the first object in stimulation trials

compared to non-stimulation trials (Fig. 5f for a representative stimulation site). At other sites, the monkey was less likely to release the button for the choice in stimulation trials (Fig. 5g for a representative stimulation site). These positive and negative effects were particularly evident in trials in which the first object value was 4. The stimulation sites that positively or negatively influenced the choice rate were distributed throughout the ventral striatum and did not cluster in any specific region (Supplementary Fig. 15a).

We performed the same analyses to statistically validate the effect of laser light stimulation on the monkey's action selection as we did for electrical stimulation. In trials in which the object value was 4, the difference in choice rate between stimulation and non-stimulation trials (Δchoice rate) did not significantly differ from zero as population because the stimulation induced both positive and negative effects across stimulation sites ($z = 0.35$, $p = 0.72$; two-tailed Wilcoxon signed-rank test) (Fig. 5h). We confirmed that the direction of these positive and negative effects was consistent across the early and late halves of the trials for each stimulation site (Supplementary Fig. 16), suggesting that these effects were not due to random trial-by-trial fluctuations in the monkey's action selection. The absolute difference in choice rate, which reflects the net effect of laser light stimulation, was significantly larger when the first object value was 4 compared with when the value was other than 4 ($n = 50$ sites, $df = 299$, $F = 51.0$, $p = 6.2 \times 10^{-38}$; one-way ANOVA and post-hoc test) (Fig. 5i, see also Supplementary Fig. 17a and b for individual monkeys). Furthermore, the distribution width of the choice rate difference, which also reflects the net effect, was significantly larger than the chance-level distribution width calculated by shuffling stimulation and non-stimulation trials when the first object value was 4 ($p = 0.002$; two-tailed Monte Carlo test) (Fig. 5j, see also Supplementary Fig. 17c and d for individual monkeys). On the other hand, when the first object value was other than 4, the distribution width was not significantly different from the chance-level distribution width ($p > 0.05$; two-tailed Monte Carlo test). Thus, as observed in the electrical stimulation experiments, laser light stimulation of dopamine projections in the ventral striatum influenced the monkey's action selection specifically when the monkey was uncertain about choosing the medium-value option.

We confirmed that the effect of laser light stimulation could not be attributed to its impact on motor action, as we demonstrated in the electrical stimulation experiment by the simple button-release motor task. A laser light stimulation with the same parameters was applied to stimulate dopamine projections in the ventral striatum from 1000 ms to 0 ms before the go signal in half of the trials, while no stimulation was delivered in the other half (Supplementary Fig. 18a). The difference in button-release latency between stimulation and non-stimulation trials was not significantly different from zero as population ($z = 0.56$, $p = 0.57$; two-tailed Wilcoxon signed-rank test) (Supplementary Fig. 18b). Please note that there was no significant correlation between the difference in button-release latency and the difference in

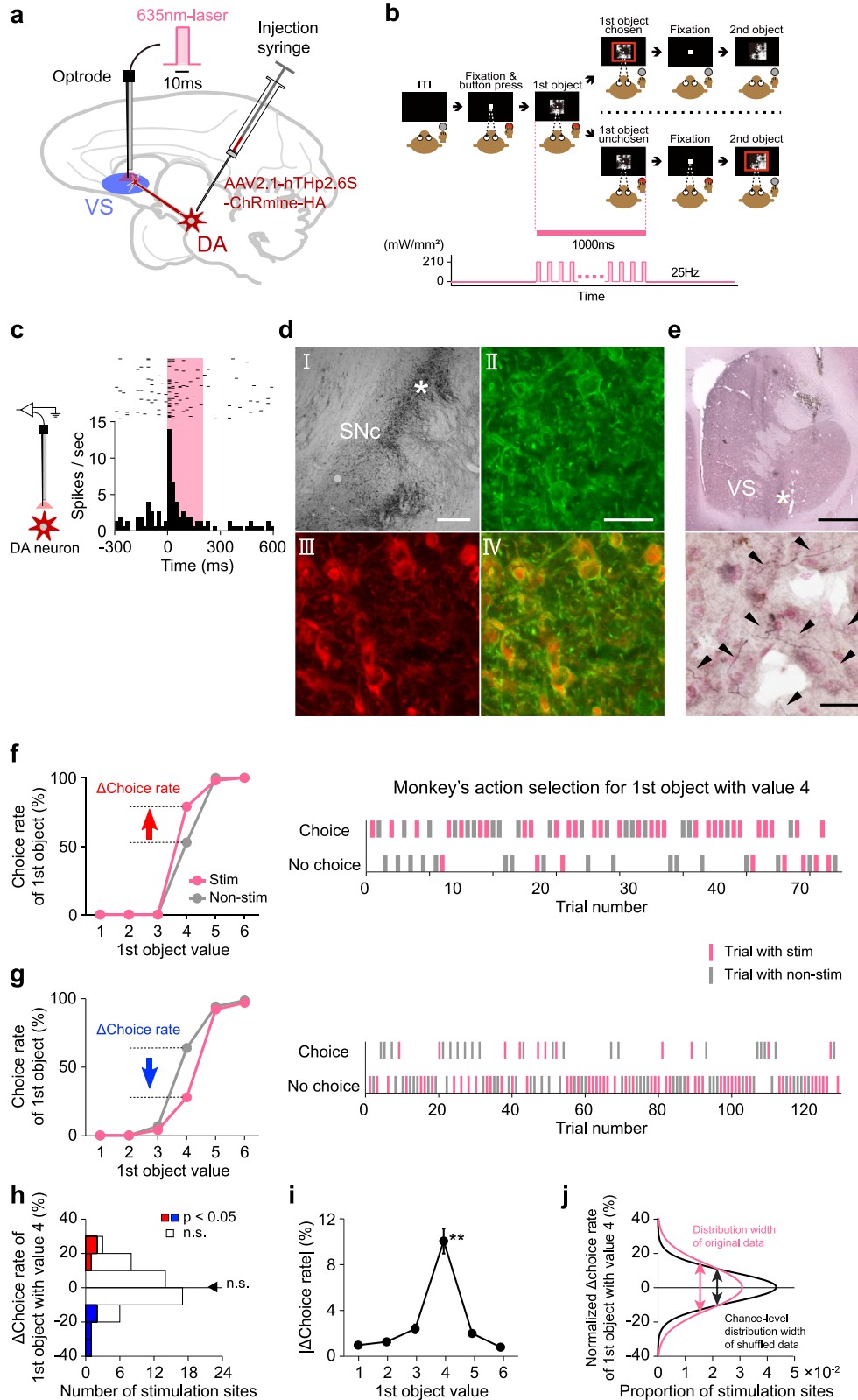

choice rate of the first object with value of 4, although the weak negative correlation was observed ($n = 50$ sites, $r = -0.23$, $p = 0.11$; two-tailed Pearson's correlation test) (Supplementary Fig. 18c). This subtle trend indicates only a limited association, if any, between the effect of laser light stimulation on the motor action and the effect on the monkey's action selection. These results therefore suggest that the effect of laser light stimulation on dopamine projections on the

monkey's action selection is unlikely to be fully explained by its impact on motor action.

As in the button-release task, laser light stimulation did not affect button-release latency in the decision-making task (Supplementary Fig. 19a). The latency difference between stimulation and non-stimulation trials was not significant ($n = 50$ sites, $z = -0.56$, $p = 0.57$; two-tailed Wilcoxon signed-rank test), and no stimulation sites showed

**Fig. 5 | Effect of optogenetic facilitation of dopamine input to the ventral striatum on monkey's action selection. a** Schematic diagram of the optogenetics experiment. VS, ventral striatum. DA, dopamine neurons. **b** Laser light stimulation period during the value-based decision-making task. ITI, intertrial interval. **c** Activation of a representative dopamine neuron evoked by a laser light stimulation. Pink area represents the period during which the laser light stimulation was delivered. Raster plots and peristimulus-time histogram are aligned at the laser light onset. **d** I, low-magnification view of the substantia nigra pars compacta (SNc) with anti-hemagglutinin (HA) protein immunohistochemistry. Scale bar, 1 mm. Asterisk represents the location of the high-magnification view. II to IV, high-magnification view of the SNc. Anti-HA (II), anti-TH immunostaining (III), and merged image (IV) are shown. Scale bar, 50 μm. **e** Top, low-magnification view of the VS with anti-HA immunohistochemistry. Scale bar, 3 mm. Asterisk represents the location of the high-magnification view. Bottom, high-magnification view of the VS indicating positive axons (arrow heads). Scale bar, 50 μm. The

immunohistochemistry shown in (**d**) and (**e**) was conducted for monkeys E and M, and yielded consistent results across both animals. **f, g** Effect of laser light stimulation at two representative sites on the monkey's choice of the first object ($n = 816$ trials in (**f**); $n = 1069$ trials in (**g**)). Conventions follow those used in Fig. 4c, d. **h** Distribution of Δchoice rates of the first object with value 4 ($n = 50$ sites). Conventions follow those used in Fig. 4e. n.s. denotes no significance ($p = 0.72$; two-tailed Wilcoxon signed-rank test). **i** Mean absolute Δchoice rate for each first object value ($n = 50$ sites). Conventions follow those used in Fig. 4f. Double asterisk denotes a significant difference ($p = 6.2 \times 10^{-38}$; one-way ANOVA and post-hoc test). **j** Gaussian functions fitted to the Δchoice rate distribution of the original data (pink curve) and the chance-level distribution of shuffled data (black curve). Conventions follow those used in Fig. 4g. (**b**) is adapted from Yun et al., Science Advances (2020)[28], licensed under CC BY-NC 4.0 (https://creativecommons.org/licenses/by-nc/4.0/). The figure has been modified from the original.

a significant effect ($p < 0.05$; two-tailed Wilcoxon rank-sum test). Moreover, there was no significant correlation between the button-release latency difference and the choice-rate difference for the first object with value 4 ($n = 50$ sites, $r = -0.03$, $p = 0.86$; two-tailed Pearson's correlation test) (Supplementary Fig. 19b). These results suggest that laser light stimulation of dopamine projections does not influence the latency to reach a decision, regardless of its effect on the monkey's action selection.

## Discussion

The ventral striatum and midbrain dopamine neurons have been regarded as neural substrates for encoding and updating the expected value of available options[1,2]. Extending beyond this literature, we found that the dopamine–ventral striatum system plays a more proactive role in deciding which option to choose. The activity of ventral striatum neurons reflected the value of the first object immediately after it was presented. Over time, however, their activity progressively shifted to reflect monkey's action selection (i.e., whether to perform the action to choose the first object). These neural dynamics suggest that the value signal in the ventral striatum gradually evolved into the choice signal as the decision was being made. Moreover, optogenetic facilitation of dopamine input to the ventral striatum as well as electrical stimulation of this region altered monkey's action selection, demonstrating a causal role of the ventral striatum and its dopaminergic modulation in decision-making. These findings advance our understanding of the dopamine–ventral striatum system in value-based decision-making by highlighting the dynamic transformation from value representation to choice encoding in the ventral striatum and the critical role of dopamine in supporting this process.

The ventral striatum has been widely recognized as a key structure for processing reward value information and regulating motivational states[3–6]. For example, a previous study demonstrated its role in representing temporally discounted values by integrating information about both reward magnitude and delay during decision-making[31]. Importantly, this study reported that ventral striatal neurons encoded the sum of discounted values across options, which may reflect a general motivational drive, whereas neurons in the dorsal striatum encoded the difference between these values, suggesting a more pivotal role for the dorsal striatum in action selection. In contrast, our present results indicate that the ventral striatum itself can also participate directly in action selection. The gradual transition we observed from value-modulated to choice-modulated signals suggests that, beyond its established role in value integration, the ventral striatum may actively contribute to transforming option-value information into action selection. However, to fully understand the division of labor between the ventral and dorsal striatum, it will be essential to examine their activity under identical behavioral conditions. Future studies that simultaneously record neuronal activity in both regions during the

same decision-making task will be crucial for elucidating how these striatal subregions cooperate to translate value into action.

The neural dynamics we observed in the ventral striatum emphasizes its role as a neural hub that bridges option valuation and action selection. Previous studies have also explored the neural mechanism that bridges option valuation and action selection. These studies mainly dealt with the prefrontal and parietal cortical areas[32–34]. For example, it has been shown that option value signals are encoded in the ventromedial prefrontal cortex and then transmitted to the dorsomedial prefrontal cortex and the intraparietal sulcus area, where neuronal activity is involved in action selection by comparing the value of each option[34]. In contrast, we found that neurons in the ventral striatum are involved in both value signaling and action selection. Their activity gradually shifted from value representation to action selection encoding, suggesting that the ventral striatum may directly derive action selection from value information. At the same time, it is important to note that neurons in the orbitofrontal cortex have been reported to encode value- and choice-related variables in decision-making situation[27,35], and that similar value-to-choice transitions have been observed in the amygdala[36,37]. Because both the orbitofrontal cortex and amygdala project directly to the ventral striatum[38,39], the signals we observed in the ventral striatum might in part be inherited from these upstream regions. Particularly, however, the relationship between the ventral striatum and prefrontal regions requires further considerations. Both regions participate in cortico-basal ganglia loop circuits, often referred to as the "limbic loop"[40,41], yet little is known about the nature of the signals exchanged between them. Computational studies have suggested that cortico-basal ganglia loop circuits, which sometimes include the dopamine system and amygdala, implement more than simple feedforward transmission of signals and highlighted the importance of iterative and recurrent processing within these circuits[42–44]. In light of our findings on the dynamic transformation from value representation to action-selection encoding in the ventral striatum, future studies should investigate how these iterative interactions between the ventral striatum and prefrontal regions contribute to decision-making, for example, through simultaneous recordings or causal perturbations of their communication.

It is also important to note that neuroscience studies on decision-making have further categorized value-related signals into distinct subtypes. For example, neurons in the dorsal striatum have been shown to encode "action value," the value associated with a potential action[45,46], whereas neurons in other brain regions encode "chosen value," the value of the option ultimately selected[27,47]. Because the task used in the present study was not designed to disentangle these subtypes, we cannot determine whether the value-modulated signal observed in the ventral striatum corresponded to action value, chosen value, or another value. Similarly, although we observed a choice-modulated signal in the ventral striatum, it remains unclear whether this signal reflects the choice of option or the choice of action. Ideally,

task designs should enable the construction of a single regression model incorporating these value- and choice-related variables, as well as other decision variables such as the sensory identity of each option and the motor components of choice, to achieve a comprehensive understanding of the neural representations underlying decision-making. Thus, while our findings establish the ventral striatum as a neural hub where value signals evolve into choice signals, the precise nature of these computations remains to be clarified. Future studies using tasks explicitly designed to dissociate these different forms of value- and choice-related signals—and other decision variables—will be essential for elucidating how the ventral striatum contributes to the full process leading to action selection.

We confirmed that the choice-modulated signal discussed above was not simply caused by the motor action (i.e., button release) associated with choosing the first object, using the control task in which the monkey was required only to release the button. This control task was designed to eliminate the process of choice in order to isolate motor-related neuronal activity in the ventral striatum. However, it also differed from the decision-making task in other important respects. In the decision-making task, the identity and value of the first object presented as an option varied unpredictably across trials, introducing uncertainty about which object would appear and what reward could be obtained. This trial-by-trial uncertainty likely evoked reward expectation and prediction error processes that were absent in the control task, where the stimulus–reward association was fixed and fully predictable. Such differences in uncertainty and reward expectation may account for the stronger neuronal modulation observed during the decision-making task. In addition to these differences, we also note that the reward structure was comparable across tasks. The reward amount per trial in the control task was set to be equivalent to value 3 or 4 in the decision-making task, which matched the expected reward magnitude in the decision-making task (3.5). However, because the duration of each trial in the control task was shorter, the overall reward rate was actually higher, potentially leading to greater motivation. Therefore, the weaker neuronal modulation observed in the control task is unlikely to reflect reduced motivation.

We observed that electrical stimulation of the ventral striatum altered the monkey's action selection. Notably, the stimulation increased the choice rate for an option at some ventral striatal stimulation sites, while decreasing it at others. These stimulation sites were interspersed throughout the ventral striatum and were not concentrated in specific regions. Such conflicting effects on the monkey's action selection might be explained by the involvement of the "direct" and "indirect" pathways of the basal ganglia[20,48,49]. Neurons in the striatum that constitute the direct pathway express the D1 dopamine receptor and are thought to regulate motor actions in an excitatory manner, whereas those forming the indirect pathway express the D2 receptor and are believed to exert an inhibitory effect on motor actions[50,51]. In particular, rodent studies selectively manipulating direct and indirect pathway neurons in the ventral striatum have demonstrated that the direct pathway promotes approach behaviors, while the indirect pathway facilitates avoidance behaviors[52–54]. Because direct and indirect pathway neurons are intermingled within the striatum, the electrical stimulation in the present study must have activated both neuron types. In addition, it is possible that more direct pathway neurons were activated at some stimulation sites, while more indirect pathway neurons were activated at others, leading to the observed opposite effects on the monkey's action selection. Future studies are called for to directly test this interpretation using cell-type–specific manipulations. Although optogenetic or chemogenetic approaches have enabled selective activation of direct- and indirect-pathway neurons in rodents[55], these techniques have not yet been fully established in non-human primates. The development of such cell-type–specific tools in primates will be essential for causally identifying the contributions of these pathways to action selection.

We also found that optogenetic facilitation of dopamine inputs to the ventral striatum altered the monkey's action selection. Importantly, this manipulation was applied "during" the decision-making process, which was prior to reward delivery. What mechanism underlies such an effect? In our previous study[28], we found that dopamine neurons encode multiple signals related to decision-making, such as the value of an option and the monkey's choice of that option, which are similar to the ventral striatum signals we observed in the present study. These dopamine signals were evoked while the monkey was making a decision, i.e., during the same period when optogenetic facilitation of dopamine input was applied in the present study. It is most likely that the decision-making-related dopamine signals transmitted to the ventral striatum would have been facilitated by our optogenetic manipulation, which could have influenced the decision-making-related signals in the ventral striatum and, consequently, altered the monkey's action selection. However, we did not directly examine the effects of optogenetic manipulation on neuronal activity in the ventral striatum. Future studies are required to investigate how the decision-making-related signals in ventral striatum neurons are modulated by dopamine input.

Our optogenetic facilitation of dopamine input increased the monkey's choice rate for an option at some stimulation sites in the ventral striatum, while decreasing it at other sites. Both stimulation sites were interspersed throughout the ventral striatum and were not concentrated in specific regions. It remains unclear why the dopamine input facilitation exerted these conflicting effects. One possibility is that the present optogenetic manipulation might have more frequently activated dopamine input to direct pathway neurons at some sites, whereas more often dopamine input to indirect pathway neurons at others. However, this biased activation cannot completely explain our results. It is well known that dopamine input to direct pathway neurons facilitates them through the D1 dopamine receptor[50,51], and that their activation promotes approach behaviors[52,54]. Thus, dopamine input to the direct pathway facilitates approach behaviors. By contrast, dopamine input to indirect pathway neurons suppresses them through the D2 receptor[50,51], and that their activation enhances avoidance behaviors[52–54]. Thus, given the reciprocal relationship between approach and avoidance, dopamine input to the indirect pathway disinhibits approach behaviors. Taken together, the dopamine inputs to both direct and indirect pathway neurons might likely increase the animal's trend to approach motivational stimuli.

Nevertheless, our results indicate that optogenetic facilitation of dopamine input to the ventral striatum produced bidirectional behavioral effects. This apparent contradiction suggests that the functional consequences of dopamine input may not be uniform across ventral striatal circuits. Several non-mutually exclusive mechanisms could underlie this heterogeneity. First, the effects of dopamine input might depend on the baseline state of the local network, such that the same dopaminergic facilitation could either enhance or suppress neuronal firing depending on ongoing activity patterns. Second, subtle differences in the subregions of the ventral striatum targeted by optogenetic stimulation (e.g., shell versus core) might contribute to the divergent behavioral effects. Third, non-canonical effects of dopamine under the specific stimulation parameters used, such as spillover activation of other receptor types or modulation of presynaptic terminals, cannot be excluded. Finally, the optogenetic manipulation was applied during the presentation of the first object, when ventral striatum neurons encode multiple signals related to value and choice. The dopaminergic facilitation might have interacted with these complex, time-varying representations, leading to variable behavioral consequences. Future studies combining cell-type-specific and temporally precise manipulations with simultaneous neuronal recordings will be essential to elucidate how dopamine input dynamically modulates value- and choice-related signals in the ventral striatum.

Please note that, although both electrical stimulation of the ventral striatum and optogenetic facilitation of dopamine inputs to this region produced bidirectional effects on the monkey's action selection, only the electrical stimulation exhibited a weak but significant anterior–posterior gradient. The positive effects tended to appear in the anterior portion, whereas the negative effects were more pronounced in the posterior portion. A recent pharmacological study in macaque monkeys also reported functional heterogeneity along this axis of the ventral striatum, showing that inactivation of its anterior portion induced hypoactive, "resting" behavior, whereas inactivation of the posterior portion elicited compulsive-like "checking" behaviors[56]. Our results raise the possibility that such anterior–posterior functional differentiation may also extend to decision-related processes. However, it remains unclear why optogenetic facilitation of dopamine inputs did not exhibit a similar gradient. One possible explanation is that electrical stimulation and pharmacological manipulation directly modulate the cell body activity of ventral striatal neurons, whereas optogenetic facilitation affects dopamine release from axon terminals. Since dopamine release mainly modulates synaptic efficacy onto striatal neurons rather than their intrinsic excitability, the anterior–posterior gradient observed with electrical stimulation might reflect the influence of non-dopaminergic afferents that directly impact neuronal firing. Additionally, the electrical stimulation sites (A23–A28) covered a slightly more anterior portion of the ventral striatum compared with the optogenetic facilitation sites (A21.5–A25), which might have contributed to the observed gradient.

Both electrical stimulation and optogenetic facilitation of dopamine inputs influenced the monkey's action selection specifically when the monkey was uncertain about whether to choose the medium-value first object (value = 4). When the object value was sufficiently high or low, the decision process became more deterministic and thus less susceptible to external perturbations. This stability likely reflects the fact that the monkeys had already learned, through extensive training, the associations between each visual object and its corresponding reward outcome. By contrast, during ambiguous decisions, the neural circuits integrating value and choice signals are likely to be in a transient, metastable state or susceptible phase, in which brief perturbations can more effectively bias the final behavioral outcome. Indeed, the medium-value first object elicited intermediate-level responses in ventral striatum neurons showing value-modulated activity. It is also possible that if the associations between visual objects and reward outcomes had not been well learned, the effects of electrical stimulation and optogenetic facilitation of dopamine inputs might have been broader and stronger. Future studies combining electrical or optogenetic stimulation with learning paradigms or systematically varying the level of decision uncertainty could further elucidate the specific computational roles of the dopamine-ventral striatum system in adaptive decision-making.

It should be noted again that our electrophysiological recordings suggest that the ventral striatum contributes not only to option valuation and action selection but also to integrating these processes as a neural hub. The ventral striatum may also participate in other processes essential for decision-making. Therefore, although electrical stimulation and optogenetic facilitation of dopamine inputs during the decision-making period altered the monkey's action selection, it remains challenging to precisely determine which component of the decision-making process was directly affected by these manipulations. Given that multiple cognitive and motor-related processes occur during the presentation of the first object, disentangling their respective contributions will require experimental designs that can temporally and functionally separate these processes. Future studies combining targeted perturbations with simultaneous neuronal recordings will be essential for clarifying how ventral striatal signals causally contribute to each stage of the decision-making process.

In summary, our findings underscore the ventral striatum's role as a neural hub that integrates option valuation and action selection. Neuronal activity in this region gradually shifted from representing the value of the first object to encoding whether to perform an action to choose it, suggesting a dynamic transformation of information during decision-making. Moreover, facilitation of dopaminergic input to the ventral striatum contributed to shaping the emerging process of action selection. Together, these results provide insight into how the ventral striatum and its dopaminergic modulation participate in bridging motivational and motor processes, and they call for future investigations into the precise mechanisms by which dopamine modulates computations in this region during value-based decision-making.

## Methods

### Animals
Three adult macaque monkeys were used in this study (monkey E: Macaca mulatta, male, 11.7 kg, 15 years old; monkey A: Macaca mulatta, male, 8.4 kg, 11 years old; monkey M: Macaca fuscata, female, 6.3 kg, 6 years old). Single-unit recording and electrical stimulation experiments were conducted in monkeys E and A. Optogenetics experiments were conducted in monkeys E and M. All animal care and experimental procedures were approved by the Animal Experiment Committee and Genetic Modification Experiment Safety Committee at the University of Tsukuba (permission number, 17-167).

### Behavioral Tasks
Behavioral tasks and data collection were conducted using the TEMPO system (Reflective Computing, WA, USA). The monkeys were seated in a primate chair, facing a computer monitor in a sound-attenuated and electrically shielded room. Eye movements were tracked with an infrared eye-tracking system (Eyelink, SR Research, Ontario, Canada) sampling at 500 Hz.

The monkeys were trained to perform a value-based decision-making task (Fig. 1a). Six visual objects were associated with different amounts of a liquid reward (water; 0.12, 0.18, 0.24, 0.30, 0.36, and 0.42 ml). For monkey E and monkey M, the visual objects were monochrome fractal images (5.2° in width and height), whereas for monkey A, they were bar stimuli (5.3° wide, 2.3° high) comprising green and magenta areas, with the color proportions indicating the reward size. Each trial began with a central fixation point (0.5° diameter) on the monitor, and the monkey was required to fixate on the point and press a button with its contralateral hand to the recording/stimulation hemisphere. Once the monkey maintained fixation and held the button for 750 ms, the fixation point disappeared, and one of the six visual objects was randomly displayed as the "first object" for 1000 ms at the center of the monitor. The monkey then had to decide whether to choose this first object by either releasing the button or keeping it pressed down. Releasing the button indicated the decision to choose the object, while keeping the button pressed down indicated the decision not to choose it. When the monkey released the button, a red open rectangle (6.3° width and height) appeared around the first object as feedback. The first object and red rectangle disappeared 1000 ms after the onset of the first object. Then, after a 400-ms fixation period, one of the six visual objects was displayed as the "second object" for 1000 ms. If the monkey had chosen the first object, it received the reward associated with it after the second object presentation. If the monkey had not chosen the first object, it could receive the reward associated with the second object by releasing the button during its presentation. When the monkey released the button, the red open rectangle appeared around the second object as feedback. Both the first and second objects were pseudo-randomly selected from the six visual objects with equal probability and independently of each other, so that the same object could appear as both the first and second objects on a given trial. Correct behavior was

indicated by a 1-kHz tone and followed by the reward delivery. Trials were aborted under the following conditions: failure to start fixation or press the button within 4000 ms, breaking fixation, releasing the button during inappropriate periods, pressing the button twice, or failing to release the button during a trial. Errors were signaled by a 100-Hz beep tone. The intertrial interval (ITI) was randomly set between 2000 and 3000 ms. The monkeys were overtrained to perform this decision-making task until their action selection became stable across daily sessions, ensuring that the associations between each visual object and its corresponding reward outcome were well learned. Therefore, no further learning was expected to occur during the recording or stimulation sessions.

The monkeys also performed a simple button-release motor task in which the monkey was not required to make decisions and was required just to release the button (Fig. 3a). Each trial began with a central fixation point (0.5° diameter) on the monitor, and the monkey was required to fixate on the point and press a button with its contralateral hand to the recording/stimulation hemisphere. After a 1000-ms fixation and button-press period, the fixation point color changed from white to gray. This color change was a go signal and the monkey was required to release the button within 2000 ms after the onset of the go signal. Correct behavior was indicated by a 1-kHz tone and followed by the reward delivery. The reward amount in the control task was set to be equivalent to value 3 or 4 in the decision-making task, which approximately matched the expected reward magnitude in the decision-making task (3.5). Trials were aborted under the following conditions: failure to start fixation or press the button within 4000 ms, breaking fixation or releasing the button during the 1000-ms fixation and button press period, or failure to release the button within 2000 ms after the go signal onset. Errors were signaled by a 100-Hz beep tone. The ITI was randomly set between 2500 and 3500 ms.

## Single-unit recording

A plastic head holder and three recording chambers were securely fixed to the skull under general anesthesia and sterile surgical conditions. Two of the chambers were placed over the frontoparietal lobes on both hemispheres, tilted laterally by 35°, and targeted to the substantia nigra pars compacta (SNc) and the ventral tegmental area (VTA). The third chamber was positioned over the midline of the frontal lobes, aimed at the ventral striatum in both hemispheres. The head holder and recording chambers were embedded in dental acrylic, which covered the top of the skull and was firmly anchored using plastic screws. After implanting the head holder and recording chambers, the monkeys underwent a magnetic resonance image (MRI) scan to determine the position of the recording electrode.

Single-unit recordings were carried out using tungsten electrodes with a single channel (impedance: 0.7 to 2.5 MΩ) (FHC, Bowdoin, USA), which were guided into the brain through a stainless-steel guide tube by an oil-driven micromanipulator (MO-97-S, Narishige, Tokyo, Japan). A grid system was used to specify recording sites, enabling recordings at 1-mm intervals between penetration points. For finer neuron mapping, a complementary grid was employed, allowing electrode penetrations between the original grid holes. On the basis of the obtained MRI and using the grid system, we inserted the tungsten electrodes into the ventral striatum (Supplementary Fig. 2). The recording sites ranged from A22 to A28 mm in monkey E and from A23 to A30 mm in monkey A along the anterior-posterior axis.

During each daily experiment, one electrode was inserted into the ventral striatum. Recording of one neuron was counted as one recording session. Therefore, zero to a few recording sessions (i.e., zero to a few neurons) were conducted per daily experiment. In total, 125 recording sessions were performed (corresponding to 125 neurons), and these data were included in the analyses.

Single-unit potentials were amplified and band-pass filtered (100 Hz to 8 kHz) using a multichannel processor (MCP-Plus 8, Alpha

Omega, Nazareth, Israel). Action potentials were isolated in real-time using a voltage-time window discrimination system (ASD, Alpha Omega, Nazareth, Israel), and the timing of each action potential was recorded with a resolution of 1 ms.

## Electrical stimulation

Electrical stimulation was generated by a stimulator (STG4000, Multi Channel Systems, Reutlingen, Germany) and applied to the ventral striatum using the same tungsten electrodes that were used for single-unit recordings. The stimulation protocol consisted of 200-μs pulses delivered at a frequency of 333 Hz. Each pulse was biphasic, with a current magnitude of 100 μA. In the value-based decision-making task, electrical stimulation was administered throughout the presentation of the first object, lasting 1000 ms. In the button-release task, stimulation was delivered for 1000 ms following the onset of the go signal. These stimulations were applied in half of the trials, while the remaining trials served as controls. The stimulation sites ranged from A23 to A25 mm in monkey E and from A24 to A28 mm in monkey A along the anterior-posterior axis, which overlapped with the electrophysiological recording sites.

## Viral vector injection

To deliver the red-light-sensitive channelrhodopsin gene (*ChRmine*) into dopamine neurons, we used an AAV2.1-hTHp2.6S-ChRmine-HA vector (1.0×10$^{13}$ genome copies per ml) that was produced via the helper-free triple transfection method and purified by affinity chromatography (GE Healthcare). AAV2.1 is a viral vector developed by a research group that includes authors of this study[57]. It efficiently infects neurons in macaque monkeys and enables high-level expression of target genes in these cells. To determine the viral vector injection sites, we first identified the substantia nigra pars compacta (SNc) and the ventral tegmental area (VTA) using MRI and mapped dopamine neuron locations within these structures through single-unit recordings. Putative dopamine neurons were identified based on established electrophysiological characteristics: a low baseline firing rate (~5 Hz), a broad spike potential that stood out from nearby neurons with high background firing rates in the substantia nigra pars reticulata, and a phasic excitation in response to a free reward. After identifying these neurons, the viral vector was injected into the SNc and VTA under general anesthesia using a microsyringe (1701RN, Hamilton, Reno, USA) equipped with a 28-gauge needle (28 G/75 mm/PT2, Hamilton, Reno, USA). Six injections were made in each hemisphere of the SNc and VTA, with injection sites spaced at least 1 mm apart. For each injection site, we introduced 0.7 to 2.0 μl of the vector at one to three different depths at a rate of 0.2 μl per minute.

## Laser light stimulation

To stimulate dopamine projections in the ventral striatum with laser light, we used an optrode (OR-250/125, Bio Research Center, Nagoya, Japan), consisting of a 250-μm-diameter optical fiber attached to a tungsten recording electrode (impedance: 0.5–2.0 MΩ, diameter: 125 μm) that extended 500 μm beyond the fiber tip. This optrode was inserted into the ventral striatum. A red laser light (638 nm wavelength, <210 mW/mm$^2$) was delivered through the optrode into the ventral striatum using a laser light source (LuxX + ® 638-100, Omicron Laserage, Rodgau-Dudenhofen, Germany). The stimulation protocol consisted of 10-ms pulses delivered at a frequency of 25 Hz. In the value-based decision-making task, laser light stimulation was administered throughout the presentation of the first object, lasting 1000 ms. In the button-release task, stimulation was delivered for 1000 ms following the onset of the go signal. These stimulations were applied in half of the trials, while the remaining trials served as controls without stimulation. The stimulation sites ranged from A22 to A25 mm in monkey E and from A21.5 to A24.5 mm in monkey M along the

anterior-posterior axis, which overlapped with the electro-physiological recording sites.

When the laser was simply turned on, light weakly leaked from the connection between the optrode and the optical fiber, which could potentially affect monkey behavior. To prevent this unintended leakage, we covered the connection with a black plastic tube and wrapped the entire assembly with a black opaque sheet. These precautions ensured that no light leakage was visible in the experimental room.

To verify the activation of dopamine neurons by red laser stimulation, we inserted the optrode into the SNc or VTA and recorded single-unit activity from putative dopamine neurons, identified by their established electrophysiological characteristics, as described above. The red laser light was then applied, and we analyzed the responses of these putative dopamine neurons to laser stimulation.

### Data analysis

To assess how the value of the first object influences the monkey's decision to release the button to choose it, we modeled the choice rate for the first object using the following logistic function:

$$P = \frac{1}{1 + \exp(-(\beta_0 + \beta_1 \times V))} \tag{1}$$

where $P$ denotes the choice rate of the first object, $V$ represents the value of the first object, and $\beta_0$ and $\beta_1$ are coefficients determined by logistic regression.

To evaluate the effects of the second object value in the previous trial on the monkey's decision to release the button to choose the first object, we modeled the choice rate for the first object using the following logistic function:

$$P_t = \frac{1}{1 + \exp\left(-(\beta_0 + \beta_1 \times V2_{t-1} + \beta_2 \times V1_t)\right)} \tag{2}$$

where $P_t$ denotes the choice rate of the first object in trial $t$, $V2_{t-1}$ represents the value of the second object in trial $t$ - $1$, $V1_t$ represents the value of the first object in trial $t$, and $\beta_0$, $\beta_1$ and $\beta_2$ are coefficients determined by logistic regression.

To calculate spike density functions (SDFs), each spike was replaced by a Gaussian kernel with $\sigma = 30$ ms.

To statistically characterize the activity of ventral striatum neurons, we fitted each neuron's activity to both value and choice models (Fig. 2d). The value model is given by:

$$F = \beta_0 + \beta_1 \times V \tag{3}$$

where $F$ represents the firing rate of each neuron, $V$ denotes the value of the first object, and $\beta_0$ and $\beta_1$ are coefficients determined by linear regression. The choice model is represented as:

$$F = \beta_0 + \beta_1 \times C \tag{4}$$

where $C$ indicates whether the monkey chose the first object (1 for chosen trials, 0 for unchosen trials). To evaluate the fit of each model, we compared the coefficient of determination (R-square) for each neuron. We used a Monte Carlo procedure to determine whether the R-square differed significantly between the two models. For each neuron, we generated a shuffled dataset by randomly reassigning the firing rate of each trial to another trial. We then fitted the baseline firing rate (0–200 ms before the first object onset) of this shuffled dataset to both models and calculated the baseline R-square difference between the models. This baseline R-square difference of the shuffled dataset was compared to the R-square difference of the original dataset during a targeted time window, and this shuffling and comparison process

was repeated 1000 times. If the R-squared difference of the original dataset exceeded that of the shuffled dataset in more than 975 out of 1000 repetitions, this indicated that the value model had a significantly higher R-square than the choice model ($p < 0.05$; two-tailed Monte Carlo test). If the value model also significantly fit the neuron's activity ($p < 0.05$; two-tailed F-test), the neuron's activity was considered to be more strongly modulated by the value of the first object (blue area in Fig. 2e). Conversely, if the R-square difference of the original dataset was smaller than that of the shuffled dataset in over 975 repetitions, the choice model was deemed to have a significantly better fit than the value model ($p < 0.05$; two-tailed Monte Carlo test). If the choice model also significantly fit the neuron's activity ($p < 0.05$; two-tailed F-test), the neuron's activity was considered to be more strongly modulated by the monkey's choice (red area in Fig. 2e). If there was no significant difference in R-square between the models ($p > 0.05$; two-tailed Monte Carlo test), but both models still significantly fit the activity ($p < 0.05$; two-tailed F-test), the neuron was considered to show an intermediate modulation between the value and the choice models (cyan area in Fig. 2e).

We performed this model comparison for each of the 150-ms sliding windows with a 1-ms step. If a neuron showed a significantly higher R-square for the value model than the choice model ($p < 0.05$; two-tailed Monte Carlo test) for 40 consecutive windows and simultaneously exhibited a significant fit with the value model across the same 40 windows ($p < 0.05$; two-tailed F-test), the activity of this period was classified as a value-modulated signal. Conversely, if a neuron showed a significantly higher R-square for the choice model than the value model ($p < 0.05$; two-tailed Monte Carlo test) for 40 consecutive windows and simultaneously exhibited a significant fit with the choice model across the same 40 windows ($p < 0.05$; two-tailed F-test), the activity of this period was classified as a choice-modulated signal. If a neuron showed no significant difference in R-square between the two models but a significant fit with both for 40 consecutive windows, the activity during this period was classified as an intermediate signal. Some neurons represented two or even three signals because their activity was classified as distinct signals during different time periods.

The model comparison analysis was also used to determine the latencies of value-modulated, intermediate, and choice-modulated signals. These latencies were defined as the start time of the first of the 40 consecutive 150-ms sliding windows with the 1-ms step where each modulation type was identified.

To confirm the validity of the classification based on the model comparison analysis, we applied a ridge regression approach, which can explicitly address multicollinearity between predictors, such as value and choice in this study. The ridge regression model was expressed as:

$$F = \beta_0 + \beta_1 \times V + \beta_2 \times C \tag{5}$$

where $F$ represents the firing rate of each neuron, $V$ denotes the value of the first object, $C$ indicates whether the monkey chose the first object (1 for chosen trials, 0 for unchosen trials). Ridge regression includes a regularization term, $\lambda(\beta_1^2 + \beta_2^2)$, in the loss function, which penalizes large coefficient estimates that can arise from correlated predictors. This regularization effectively stabilizes parameter estimation and allows for a more reliable assessment of each predictor's relative influence, even when $V$ and $C$ are correlated. This ridge regression model was fitted separately to the activity of each neuron classified as value- or choice-modulated based on the model comparison analysis. The activity ($F$) was measured within the time window during which each neuron exhibited the value- or choice-modulated signal.

To test whether the modulation of each neuron could be accounted for by the variation in reaction time (the latency of button

release), we used another ridge regression model:

$$F = \beta_0 + \beta_1 \times V + \beta_2 \times RT \qquad (6)$$

where $F$ represents the firing rate of each neuron, $V$ denotes the value of the first object, $RT$ indicates the latency of button release to choose the first object. The activity ($F$) was measured within each 150-ms window after the first object onset. To test whether the proportion of neurons showing a significant regression coefficient was larger than expected by chance, we performed a permutation-based Monte Carlo analysis. For each neuron, we shuffled the firing rates across trials and assigned them to another trial at random to generate surrogate datasets; the same ridge regression analysis was applied to each shuffled dataset. The proportion of neurons with a significant regression coefficient was then computed for each predictor. To address the multiple-comparison problem arising from repeated tests across time windows, we adopted a max-statistic permutation procedure. In each of 1000 permutations, the maximum proportion of significant neurons across all 150-ms windows was obtained and used to construct the null distribution of the chance level. The observed proportion in each window was then compared with this null distribution to determine statistical significance while controlling the family-wise error rate (FWER).

To examine the effect of the second object value in the previous trial, we used another ridge regression model:

$$F = \beta_0 + \beta_1 \times V2_{t-1} + \beta_2 \times V1_t + \beta_3 \times C_t \qquad (7)$$

where $F$ represents the firing rate of each neuron, $V2_{t-1}$ denotes the value of the second object in the previous trial ($t$-$1$), $V1_t$ represents the value of the first object in the current trial ($t$), and $C_t$ indicates whether the monkey chose the first object in the current trial ($t$) (1 for chosen trials, 0 for unchosen trials). The activity ($F$) was measured within each 150-ms window after the first object onset. To test whether the proportion of neurons showing a significant regression coefficient was larger than expected by chance, we performed the same permutation-based Monte Carlo analysis described above.

To evaluate the effects of electrical stimulation of the ventral striatum and optogenetic facilitation of dopamine projections in the ventral striatum, we calculated the difference in choice rate between stimulation and non-stimulation trials for each object value (Δchoice rate). The statistical significance of Δchoice rate was evaluated using a chi-square test for each stimulation site.

At the population level, we compared the width of the distribution of Δchoice rate, which reflects the net effect of stimulation on the monkey's action selection (even if positive and negative effects are mixed), with a chance-level distribution width using a Monte Carlo procedure. To determine the chance-level distribution width, we shuffled the stimulation and no-stimulation conditions for each trial at random and created a new dataset for each stimulation site. Using this shuffled dataset, we calculated the Δchoice rate for each stimulation site and measured the distribution width of the Δchoice rate as the chance-level distribution width. We then compared the original distribution width with the chance-level distribution width. This process of shuffling and comparison was repeated 1000 times. If the original distribution width exceeded the chance-level distribution width in more than 975 out of 1000 repetitions, this was taken as evidence that the stimulation had a statistically significant effect on the monkey's action selection, independent of whether it increased or decreased the choice rate ($p < 0.05$; two-tailed Monte Carlo test).

## Histology
Upon completion of all experiments, monkeys E and M were deeply anesthetized with secobarbital sodium (150 mg/kg) and transcardially perfused with phosphate-buffered saline (PBS), followed by 10% formalin in PBS (pH 7.4). The removed brains were postfixed in the same fresh fixative overnight at 4 °C and saturated with 30% sucrose at 4 °C. Coronal sections were cut serially at a 50-μm thickness on a freezing microtome and grouped into ten series. Every tenth section was used for individual histological analyses.

To visualize projection fibers of virus-transfected neurons, a series of every tenth sections was subjected to immunohistochemical staining for hemagglutinin (HA) protein, utilized as an optogenetic tag, using the standard avidin-biotin-peroxidase complex method with 3,3'-diaminobenzidine (DAB) as the chromogen. A primary antibody [1:500, mouse anti-HA.11 clone 16B12 monoclonal IgG (901513, BioLegend, San Diego, USA)] and a biotinylated secondary antibody [1:1000, donkey anti-mouse IgG biotin-SP-conjugated antibody (715-065-150, Jackson ImmunoResearch Laboratories, West Grove, USA)] were employed. These sections were mounted onto gelatin-coated glass slides, counterstained with Neutral red, and coverslipped after DAB staining.

To test whether virus-transfected neurons are dopaminergic, another series of every tenth section was subject to immunofluorescence staining to examine the co-expression of HA protein and Tyrosine Hydroxylase (TH), a marker for dopamine neurons. Primary antibodies [1:300, mouse anti-HA.11 clone 16B12 monoclonal IgG (901513, BioLegend, San Diego, USA); 1:1000, rabbit anti-TH polyclonal IgG (AB-152, Chemicon, Temecula, USA); 1:3000, guinea pig anti-NeuN polyclonal IgG (ABN90, Millipore, St. Louis, USA)] and secondary antibodies [1:400, donkey anti-mouse IgG conjugated with Alexa Fluor 488 (A21202, Jackson ImmunoResearch Laboratories, West Grove, USA); 1:400, donkey anti-rabbit IgG conjugated with Alexa Fluor 555 (A31572, Invitrogen, Waltham, USA); 1:200, donkey anti-guinea pig IgG conjugated with Alexa Fluor 647 (706-605-148, Jackson ImmunoResearch Laboratories, West Grove, USA)] were employed. Following staining, these sections were mounted onto gelatin-coated glass slides. Fluorescent images were acquired using a fluorescence microscope (BZ-X700, Keyence, Osaka, Japan).

## Reporting summary
Further information on research design is available in the Nature Portfolio Reporting Summary linked to this article.

## Data availability
Source data are provided with this paper.

## Code availability
No custom codes or algorithms were developed for this study. All analyses were performed using standard MATLAB (MathWorks) functions, and the analysis procedures are fully described in the 'Methods' section. All codes used in this study are available from the corresponding author upon request.

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

## Acknowledgements

We thank K. Bunzui, T. Okano, Y. Narita, S. Nishino, and N. Kajiwara for animal care; E. Tanaka for histology; M. Nakano and M. Fujiwara for assistance in the production of viral vectors; and Y. Nishimura and S. Sugawara for advice on analyses. This research was supported by MEXT KAKENHI grant number JP16H06567 (to M.M.), JP23H00412 (to M.M.), JP19K16890 (to M.N.), JP22K15627 (to M.N.), and JP22H04922 (to K. I.); JST CREST grant number JPMJCR1853 (to M.M.); and AMED PRIME grant number 23gm6510030h0001 (to M.M.).

## Author contributions

M.N. and M.M. conceived of the project and designed the experiments. M.N., M.Y., Y.W., and T.K. performed the experiments. K.I. and M.T. produced the viral vector. M.N. and M.M. analyzed the data. M.N. and M.M. wrote the manuscript. M.N., M.Y., Y.W., T.K., J.K., H.Y., K.I., M.T., and M.M. discussed the results and reviewed and edited the manuscript.

## Competing interests

The authors declare no competing interests.
