## [Transparent Peer Review file · Nature Communications]

Role of the primate ventral striatum as a neural hub bridging option valuation and action selection

Corresponding Author: Professor Masayuki Matsumoto

Version 0:

Reviewer comments:

Reviewer #1

(Remarks to the Author)

This manuscript reports activity patterns of ventral striatum neurons recorded while monkeys performed a two-step reward-guided decision task. Neurons encoded values of choice options, action selection and their combination. Electrical stimulation in the ventral striatum, directly and indirectly via optogenetic dopamine stimulation, biased behavioral action selection. The authors suggest these data indicate a broader role of ventral striatum in reinforcement learning that extends to valuation and action selection.

This is an interesting study that can make an important contribution to the literature and advance our understanding of the little explored primate ventral striatum. The experiments are conducted at high quality and the data seem robust, although the number of reported neurons ($N = 125$) is relatively low. The stimulation experiments are impressive and together with the motor-preparation controls add important insights. Overall, I am very supportive of publication. However, there are a few issues that should be addressed in a revision.

Major comments

1. There are some features related to the task design that make it an unusual model of decision-making, and that in my view limit the conclusions that can be made about decision computations. The monkey does not have to wait for the second stimulus to engage in decision-making but can already make and execute a decision at the first stimulus. This process likely involves comparing the first stimulus value against the expected value of the second option, which is the mean value of all remaining stimuli. Thus, at the time of the first stimulus, one would expect neurons to code the value of the first option, the prospective value of the second option, perhaps a correlate of the individual monkey's risk attitude (preference or aversion toward the more uncertain second stimulus), and this might transition into a choice signal. Importantly, however, the choice signal would necessarily be referenced to the button push, i.e., an action-referenced choice signal, rather than a stimulus-specific choice signal. The task therefore allows identification of option values and action choices, but not of stimulus choices, as it conflates stimulus choice with the motor response. It also does not temporally separate the internal processes of value comparison and decision-making from the process of action selection—this would require additional task periods. The abstract and manuscript should be rephrased to reflect this ambiguity. For example:

Abstract, line 23: “deciding which option to choose” should be reworded to clarify that what is investigated is action selection.

Abstract, line 27: “reflect the monkey's choice”, and “derives the choice”; these expressions should be amended to clarify that the choice that is demonstrated in the study is whether or not to perform a specific action (as this cannot be dissociated from stimulus choice in the present task design).

Discussion, line 406: “encoded the value of the option (i.e., the value of the first object)”; this should be rephrased to reflect the task design, which cannot distinguish object value coding from action value coding

2. The authors should also report the results of analysing the second-option period, as this will help clarify the specific information encoded by striatal neurons. For example, action value neurons would code the action of the button press both at first and second stimulus, whereas option coding neurons.

3. I was surprised that two pioneering studies on action value coding in primate striatum were not cited and discussed

(Samejima et al., 2005; Lau and Glimcher, 2008).

4. As some striatal neurons showed mixtures of value and choice coding, it would be more appropriate to conduct a multiple regression model that includes both variables and thus allow them to compete directly to explain variance in neuronal activity.

5. It was unclear from the methods description how the critical p-value for the F-test of model fit in the sliding window analysis was determined. This should likely be based on a similar permutation method as the comparison of R-values but I could not find this information. Did the authors impose a criterion of a critical number of significant F-tests in consecutive sliding windows?

6. The authors' interpretation that the ventral striatum might directly derive choice from value should be discussed in light of findings that neurons in the orbitofrontal cortex (Padoa-Schioppa and Assad, 2006; Ballesta and Padoa-Schioppa, 2019) and amygdala (Grabenhorst et al., 2012; Grabenhorst et al., 2023) encode subjective values for stimuli, chosen values, stimulus choices, but not necessarily chosen actions (this includes data from sequential decision tasks). Specifically, value-to-choice transitions as reported here have previously been reported in primate amygdala (Grabenhorst et al., 2012; Grabenhorst et al., 2023). Both OFC and amygdala provide direct inputs to ventral striatum. Thus, it is possible the presently observed signals could be inherited from these structures. Alternatively, if the authors can demonstrate that ventral striatal neurons code the value and choice for action (i.e., the same neurons signalling choice at first and second stimulus), irrespective of stimulus identity, this might add an interesting new aspect to our understanding of value and choice coding in this network

7. The behavioral regression in its current form is perhaps not providing a full picture of the monkeys' decision process: in addition to the value of the first option, the expected value of the second option should also be taken into account. Similarly, the monkeys' decision-making might be influenced by individual risk attitudes. It would be helpful to model these influences on choice in the regression, or, in the case of risk attitude, at least discuss whether this variable might have influenced the monkeys' choices.

8. In addition to value-to-choice transition (better labelled value-to-action transition), the authors should examine other indicators of a decision computation: (i) stronger choice signals for easier decisions (unsigned value difference) compared to difficult decisions; (ii) correlates of value comparison, e.g., transient relationship between firing rate and value difference between options.

Related to this issue: Abstract, line 32: "modulating decision computations"; decision computation has not been sufficiently demonstrated in the study; unless further evidence for decision computation in the neural data is provided, this should be changed to "bias action selection" or something similar

Minor comments.

9. Data from neurons recorded from the primate ventral striatum in decision tasks are rare. It is a shame that the authors did not document these neurons in more detail with raster plots, to show the response pattern and recording quality, as they do for the dopamine neuron in Fig. 5c, and for which the authors have a distinguished track record.

10. Fig. 3b should include standard error/deviation in addition to the mean response. It would also be helpful to additionally see these data aligned to the onset of the first stimulus, rather than button press, as this is more consistent with prior studies indicating stimulus-based choice signals

11. Line 149, typo: "stratum"

References

- Ballesta S, Padoa-Schioppa C (2019) Economic Decisions through Circuit Inhibition. *Current biology* : CB 29:3814-3824 e3815.
- Grabenhorst F, Hernadi I, Schultz W (2012) Prediction of economic choice by primate amygdala neurons. *Proc Natl Acad Sci U S A* 109:18950-18955.
- Grabenhorst F, Ponce-Alvarez A, Battaglia-Mayer A, Deco G, Schultz W (2023) A view-based decision mechanism for rewards in the primate amygdala. *Neuron* 111:3871-3884 e3814.
- Lau B, Glimcher PW (2008) Value representations in the primate striatum during matching behavior. *Neuron* 58:451-463.
- Padoa-Schioppa C, Assad JA (2006) Neurons in the orbitofrontal cortex encode economic value. *Nature* 441:223-226.
- Samejima K, Ueda Y, Doya K, Kimura M (2005) Representation of action-specific reward values in the striatum. *Science* 310:1337-1340.

Reviewer #2

(Remarks to the Author)
Summary

In this study, Neijme and colleagues record and manipulate neural activity in the ventral striatum (VS) of two macaque monkeys to study the region's role in a sequential value-based choice task. They show that the VS has an important role in the transition from option valuation to action selection signals. Two experimental manipulations using electrical stimulation and optogenetics are used to support this finding with causal evidence.

This is an interesting study that examines the role of the ventral striatum in value and choice/action coding using a combination of recording and causal approaches. It involves an impressive set of experiments. It also uses control tasks and is conducted to a high standard overall. I have a few major and minor comments that I believe would strengthen the conclusions of the work.

Major

1. By definition, in real-life, value and choice are correlated, yet the ability to dissociate the two processes is critical for concluding that the VS is responsible for the choice-to-action transformation. How did the authors address this issue? Did they simulate data to show that choice value coding and action coding can be dissociated in their study using their paradigm? It would be good to see evidence of this.
2. It seems a shame that choice is not modulated in a predictable direction when using optogenetic or electrical stimulation. However, it should be possible to show that the direction was consistent for a given recording site/area. Could you test whether the modulation is consistent across repeat stimulations for a single stimulation site to provide evidence that the choice modulation effects go beyond simply adding noise but show a consistent direction as long as the same location is studied? Related to this, why does it seem as though there is an anterior-posterior effect for the electric but not optogenetic experiment? Were the sites comparable in the two experiments or could you discuss why they might differ? Related to this analysis, I was not sure why there are seven sites in figure 4d but in the xyz figure with location there are many more sites? Could you just show the xyz plot for value 4 to match with Fig 4d?
3. I am not an expert in optogenetics stimulation, but it is commonly discussed that this technique comes with significant challenges in macaque monkeys/larger NHPs. What was done to overcome these limitations here? How can we be sure that the conclusions can be trusted? Relatedly, could the animals (for either stimulation method) possibly tell when the stimulation was on? Or if not, how can you rule out simple confound effects of the stimulation?
4. In Fig.2i, it would be interesting to have more information of the temporal dynamics. For example, did most of the transitions from value or intermediate neurons to choice neurons happen in the late period? Is it possible to provide more detail on how the three types of neurons switch over time? These dynamics seem to be at the core of what the paper can contribute.
5. Finally, does the stimulation of VS (with either electrical or optogenetic methods) influence decision reaction times (RTs) and is there any correlation between Δ Release latency in the button-release task and Δ decision RT in the decision task (Figs 4 & 5)?

Minor

1. Several studies have demonstrated the role of the VS in cost-benefit decision making, e.g., VS lesions or VS dopamine depletion reduces animals' likelihood of choosing a high-effort choice option (Hauber & Sommer, 2009; Nicola, 2010; Salamone et al. 1994). Does this relate at all to the ideas presented in this manuscript or if not, why not?
2. Please specify in the Methods which pairings of objects were used how frequently.
3. Please plot the data from each monkey separately or in separate lines in the key figures.
4. While it is brilliant to see that control experiments using simple motor tasks were performed, there is more than one difference between the task of interest and the control task. For example, in the simple motor task, there is no choice, no value processing and expectation, no modulation of outcome size and thus overall less reward on average. It would be good to acknowledge and discuss these differences in the discussion.
5. In Fig.1, please show the duration for each stage of the task in Fig.1a.
6. Why did the authors pick 'release' as their response modality?

Reviewer #3

(Remarks to the Author)

This manuscript addresses an important and timely topic: how the ventral striatum contributes to value-based decision-making. The authors used a combination of single-unit recordings, electrical stimulation, and optogenetics in nonhuman primates. The monkeys performed a sequential choice task between pairs of visual stimuli associated with different amounts of reward, and the authors report that ventral striatum activity transitions from value encoding to action selection. They further show that stimulation of the ventral striatum or its dopaminergic inputs can bias choice behavior. The multi-method approach is a strength, and the effort to dissect decision-making phases is valuable. However, several key conceptual and interpretive issues need to be addressed. In particular, the study is often framed within a reinforcement learning framework, but this is not well supported by the task design or data. Sample sizes of recorded neurons are relatively small, and the observed effects of stimulation, while interesting, are also limited. Some analyses can be improved. I believe some of these concerns can be addressed through reframing and clarification, as well as the introduction of further analysis. Below are specific comments. I hope the authors find them useful.

Major points:

1) Figure 2 shows firing rate differences that may reflect variations in reaction times, as well as other task/behavioral variables, rather than “value.” Have the authors tested for correlations between firing rates and reaction times? In general, reaction times should be considered as a covariate or explanatory variable in the analysis.

The author can, for example, build a single regression model to predict the neuronal activity of single neurons over time based on different predictors, including reaction times, reward amount, and choice, and estimate the variation in coefficients for each predictor over time, as well as the fraction of neurons that are significant. This would give better estimate of what task/behavioral variable influence neuronal activity. It would be actually also interesting to include the chosen stimulus identity in the model.

I think this approach can be helpful for assessing temporal transitions in encoding, which is exactly what the authors claim is happening (from value to choice over time).

2) I appreciate the ambition of the authors to connect their findings to a broader theoretical framework, but this is not a reinforcement learning task. Monkeys are basically choosing between overlearned stimuli. Thus, there is no learning. I think this is a very important point. The manuscript repeatedly attempts to position the findings within a reinforcement learning framework, but this interpretation is not adequately supported by the data. As a result, several claims appear overextended, which diminishes the overall strength of the study. A more cautious, data-driven interpretation would improve the clarity and impact of the work.

Some examples from the manuscript:

- “Notably, the transitions, encoding from state value to action value and further to action, were often observed within single neurons, rather than across segregated neuronal populations.” The concepts of state value and action value are not applicable in this task.

- “This finding challenges the classical notion that the ventral striatum primarily encodes value-related signals in the reinforcement learning framework”. It is unclear how the present results challenge the encoding of value-related signals in the VS, especially given the lack of learning.

- “Instead, our data suggest a broader role of the ventral striatum potentially as a neural substrate not only for value representations, but also for some aspects of policy.” This seems a bit vague. What specific aspects of “policy” are meant here?

- Also related, the authors often refer to reward prediction errors, which cannot be measured in the present study because animals were likely overtrained, and the reward was not probabilistic.

3) The effects of ventral striatum and dopaminergic neuron stimulation were absent on over 80% of trials and appeared restricted to conditions of uncertainty. This limited efficacy should be acknowledged and discussed, as it has implications for interpreting the causal role of these signals. I think that the weak effects are likely due to the absence of learning in the task. Therefore, it is unclear to what extent these results can be generalized.

4) I think the authors should clearly state that the monkeys were overtrained and that there was no learning in the task. This clarification is critical for contextualizing the neural signals and avoiding overinterpretation in terms of reinforcement learning.

Minor points:

1) I do not understand the following sentence and the data to support it: “During this period, we observed the signal dynamics of ventral striatum neurons corresponding to the time course of the decision-making process.” (line 260-261)

2) Please clarify how many electrodes were used per session, how many recording sessions were included in the analysis, and whether ventral striatal activity differed depending on whether recordings were made in the hemisphere ipsilateral or contralateral to the arm used to perform the task.

3) Caption in Figure 1: “Error bars in (b) and (c) indicate SEM, which are very small and hidden in most cases”. I really cannot see any error bar. Maybe reducing the size of the black circles can help?

Reviewer #4

(Remarks to the Author)

Nejime and colleagues investigated the role of the primate ventral striatum in value-based decision-making, specifically examining its function in the transformation of option valuation into action selection. The authors challenge the traditional view that the VS primarily encodes expected value, proposing instead a more active role in the decision process itself.

The key findings are:

VS neurons exhibit dynamic encoding, initially representing the value of an offered option (offer 1) and subsequently shifting their activity to reflect the monkey's upcoming choice (to accept or reject offer 1). This transition often occurs within individual

neurons.

Both electrical stimulation of the VS and optogenetic facilitation of dopamine input to the VS during the decision period (presentation of offer 1) significantly altered the monkeys' choices, particularly when they were uncertain (i.e., for options of intermediate value). These manipulations could bias choices both towards and away from selecting the option, depending on the specific stimulation site.

Control experiments suggest these effects are not merely due to influences on motor execution.

The authors conclude that the VS acts as a crucial neural hub that actively bridges option valuation and action selection, with dopamine playing a critical role in modulating these decision computations in real-time, extending beyond its classical role in reward prediction error signaling.

The idea that the VS itself might be involved in deriving the choice, rather than just passing value information to other regions, is particularly intriguing. The study employs a sophisticated combination of state-of-the-art techniques in non-human primates. Optogenetics in primates is still challenging, and its successful application here to dissect the role of dopamine is commendable.

While this study addresses an important question in decision neuroscience and employs sophisticated methodologies, I have substantial concerns outlined as follows.

Major comments :

1. The authors state that the task temporally separates option evaluation, decision commitment, and motor execution. However, during the presentation of the first object (offer 1), the animals must evaluate offer 1, make an accept/reject decision, and execute a motor response (hold or release the button). This conflation of evaluation, decision, and motor execution for offer 1 should be acknowledged more clearly. The primary temporal separation is between the decision on offer 1 and the revelation/receipt of offer
2. The manuscript would benefit from a more explicit model of the cognitive operations underlying the monkey's choice. The decision to accept/reject offer 1 is presumably based on a comparison of its value, $V(\text{offer 1})$, against an expected value of offer 2, $E[V(\text{offer 2})]$, which is learned from past experience. The paper does not detail how $E[V(\text{offer 2})]$ is operationalized or assumed to be updated. This is crucial because VS activity might reflect components of this expectation.
3. To better understand how $E[V(\text{offer 2})]$ might be formed and whether it's encoded, the authors could test if $V(\text{offer 2})$ from trial $n-1$ influences choice behavior for offer 1 in trial n (e.g., by splitting trials based on previous $V(\text{offer 2})$ being high vs. low) and if this variable is encoded in VS activity during the offer 1 period of trial n .
4. Consequently, during the offer 1 period, VS neurons could be encoding multiple variables beyond just $V(\text{offer 1})$ and the impending choice. These potentially include: $E[V(\text{offer 2})]$ (possibly influenced by $V(\text{offer 2})$ from trial $n-1$ or a running average), the decision variable itself (e.g., $V(\text{offer 1}) - E[V(\text{offer 2})]$), and motor preparation signals related to holding or releasing the button. This complexity makes the unique attribution of neural activity changes and stimulation effects to a single cognitive component challenging.
5. The authors use a 150ms sliding window with a 1ms step to classify neural response modulations (value, choice, intermediate). For a typical analysis window (e.g., 500ms or more), this results in a large number of statistical tests per neuron. It is crucial to clarify if and how corrections for multiple comparisons (e.g., Bonferroni or FDR) were applied. Without such corrections, the reported proportions of neuron types could be inflated.
6. As a robustness check and to complement the dynamic analysis, the authors should consider reporting results from analyses using a fixed, a priori defined time window for value encoding (e.g., early after offer 1 onset) and choice encoding (e.g., later, pre-response). This is common practice and can help validate the sliding window findings.
7. The text sometimes refers to classifying "neurons" (e.g., "value-modulated neurons"). It's more precise to state that specific responses or epochs of activity within a neuron's firing pattern are classified, as a single neuron can show different types of modulation over time, which the authors indeed report.
8. The bidirectional effects (increasing or decreasing choice probability) are reasonably attributed to potential mixed activation of direct and indirect pathway neurons. This remains speculative without cell-type specific manipulations.
9. The bidirectional effects here are more puzzling. Dopamine typically excites D1-expressing direct pathway neurons and inhibits D2-expressing indirect pathway neurons, both actions generally promoting approach. The manuscript's discussion of differential modulation of distinct VS populations is a starting point, but this warrants deeper exploration or acknowledgment of its unresolved nature. Could it relate to different baseline states, targeted VS subregions, non-canonical dopamine effects at the stimulation parameters used, or interaction with the complex encoding during offer 1 (see point 4)?
10. Given the multiple cognitive and motor-related processes potentially occurring during offer 1 presentation (as noted in point 4), interpreting precisely which aspect of the decision process is being altered by electrical or optogenetic stimulation is challenging.

11. The simple button-release task is used to argue that VS choice signals are not purely motor. However, details regarding the reward structure in this control task are missing. If the reward magnitude or expectancy in the control task significantly differed from the decision-making task, it could influence baseline VS activity and the observed modulation strength, potentially confounding the comparison in Fig. 3c. This needs to be clarified.

Minor Comments :

1. Plotting trial-by-trial choice behavior for value 1 = 4 under stimulation vs. no-stimulation conditions could reveal finer-grained dynamics, such as adaptation or sequential dependencies in the stimulation effect.
2. For correlations that are "close to significance" (e.g., Fig S6c, $p=0.11$ for opto-stim effect on choice vs. motor latency), it's advisable to acknowledge the trend and be cautious about definitively concluding "no relationship," especially with modest sample sizes for correlational analyses.
3. The manuscript should specify the time windows or criteria used to define a neuron's state (value, intermediate, choice) for calculating the transition probabilities shown in Fig 2i.
4. The 50% transition probability between "value-modulated" and "intermediate" states (and vice-versa) warrants discussion. What does this bidirectionality imply about the underlying neural process compared to the more directed transitions towards "choice-modulated" states?
5. The discussion acknowledges the role of prefrontal cortex (PFC) in transforming value to action. The findings beg the question of how VS computations are coordinated with PFC inputs. Is the VS transforming value signals received from regions like OFC/vmPFC, or is it part of a more iterative processing loop? A more nuanced discussion of these potential network interactions would enrich the paper.
6. To further situate the current findings within the field, the authors may wish to consider additional relevant studies on primate ventral striatum (VS) function in reward processing and decision-making. For example, incorporating discussion of works such as:
<https://doi.org/10.1523/JNEUROSCI.2430-24.2025>
<https://doi.org/10.1016/j.neuron.2010.11.041>

Reviewer #5

(Remarks to the Author)

Reviewer #6

(Remarks to the Author)

Version 1:

Reviewer comments:

Reviewer #1

(Remarks to the Author)

The authors have addressed all my comments.

Reviewer #2

(Remarks to the Author)

Thanks for all your work in addressing my concerns. I believe the manuscript has been significantly strengthened as a result. I have no major comments left and my minor comments below are optional small points.

Related to my previous comment #3, I wondered if it would make sense to add the precautions to prevent light leaking into the experimental room during optogenetic stimulation to the Methods in one or two sentences (e.g. to mention the black tube & black sheet).

I also think that the figure shown in response to my previous comment #4 showing transitions from value to intermediate and intermediate to choice neurons in terms of latency could make an interesting supplementary figure. I leave it up to the

authors to include it or not, according to their preference.

Reviewer #3

(Remarks to the Author)

The authors have adequately addressed all of my concerns, and I appreciate the inclusion of the additional analyses. The manuscript has improved and is now more coherently framed in light of the evidence provided. I have no further comments.

Reviewer #4

(Remarks to the Author)

The authors have addressed all of my concerns with the original manuscript.

Reviewer #5

(Remarks to the Author)

Reviewer #6

(Remarks to the Author)

Reviewer #1 (Remarks to the Author):

This manuscript reports activity patterns of ventral striatum neurons recorded while monkeys performed a two-step reward-guided decision task. Neurons encoded values of choice options, action selection and their combination. Electrical stimulation in the ventral striatum, directly and indirectly via optogenetic dopamine stimulation, biased behavioral action selection. The authors suggest these data indicate a broader role of ventral striatum in reinforcement learning that extends to valuation and action selection.

This is an interesting study that can make an important contribution to the literature and advance our understanding of the little explored primate ventral striatum. The experiments are conducted at high quality and the data seem robust, although the number of reported neurons ($N = 125$) is relatively low. The stimulation experiments are impressive and together with the motor-preparation controls add important insights. Overall, I am very supportive of publication. However, there are a few issues that should be addressed in a revision.

> Response to the above comment

We are very pleased to hear the supportive comment from the reviewer. Thank you so much.

Major comments

1. There are some features related to the task design that make it an unusual model of decision-making, and that in my view limit the conclusions that can be made about decision computations. The monkey does not have to wait for the second stimulus to engage in decision-making but can already make and execute a decision at the first stimulus. This process likely involves comparing the first stimulus value against the expected value of the second option, which is the mean value of all remaining stimuli. Thus, at the time of the first stimulus, one would expect neurons to code the value of the first option, the prospective value of the second option, perhaps a correlate of the individual monkey's risk attitude (preference or aversion toward the more uncertain second stimulus), and this might transition into a choice signal. Importantly, however, the choice signal would necessarily be referenced to the button push, i.e., an action-referenced choice signal, rather than a stimulus-specific choice signal. The task therefore

allows identification of option values and action choices, but not of stimulus choices, as it conflates stimulus choice with the motor response. It also does not temporally separate the internal processes of value comparison and decision-making from the process of action selection—this would require additional task periods. The abstract and manuscript should be rephrased to reflect this ambiguity. For example:

Abstract, line 23: “deciding which option to choose” should be reworded to clarify that what is investigated is action selection.

Abstract, line 27: “reflect the monkey’s choice”, and “derives the choice”; these expressions should be amended to clarify that the choice that is demonstrated in the study is whether or not to perform a specific action (as this cannot be dissociated from stimulus choice in the present task design).

Discussion, line 406: “encoded the value of the option (i.e., the value of the first object)”; this should be rephrased to reflect the task design, which cannot distinguish object value coding from action value coding

> Response to the above comment

We agree with the reviewer that our task design imposes limitations and does not allow us to distinguish between stimulus choice coding and action choice coding, nor between object value coding and action value coding. As suggested, we have revised the relevant descriptions throughout the manuscript.

Abstract, line 23: “deciding which option to choose” should be reworded to clarify that what is investigated is action selection.

>The original sentence, “we found that the dopamine–ventral striatum system plays a more proactive role in deciding which option to choose.”, has been revised as follows (new line 21-22):

“we found that the dopamine–ventral striatum system plays a more proactive role in action selection.”

Abstract, line 27: “reflect the monkey’s choice”, and “derives the choice”; these expressions should be amended to clarify that the choice that is demonstrated in the study is whether or not to perform a specific action (as this cannot be dissociated from stimulus choice in the present task design).

>We changed “reflect the monkey’s choice” and “derives the choice from value information” as follows (new line 26 and 27, respectively):

“reflect monkey’s action selection”

“translates the value information into the action”

Discussion, line 406: “encoded the value of the option (i.e., the value of the first object)”; this should be rephrased to reflect the task design, which cannot distinguish object value coding from action value coding

>We revised “encoded the value of the option (i.e., the value of the first object)” as follows (new line 552):

“reflected the value of the first object”, which we believe is a more neutral description of the observed phenomenon.

In addition, we have systematically revised the introduction, results, and discussion to maintain consistency. For example, phrases such as “select option” or “choose option” were replaced with *“select action”*, “decided whether to choose the option” was changed to *“decided whether to perform a specific action to choose the option”*, and “choice behavior” was replaced with *“action selection”*. Numerous other related expressions were similarly adjusted.

Finally, to clearly acknowledge the limitations of the study, we added the following paragraph to the discussion (new line 609-628):

“It is also important to note that neuroscience studies on decision-making have further categorized value-related signals into distinct subtypes. For example, neurons in the dorsal striatum have been shown to encode “action value,” the value associated with a potential action^{45,46}, whereas neurons in other brain regions encode “chosen value,” the value of the option ultimately selected^{27,47}. Because the task used in the present study was not designed to disentangle these subtypes, we cannot determine whether the value-modulated signal observed in the ventral striatum corresponded to action value, chosen value, or another value. Similarly, although we observed choice-modulated signal in the ventral striatum, it remains unclear whether this signal reflects the choice of option or the choice of action. Ideally, task designs should enable the construction of a single regression model incorporating these value- and choice-related variables, as well as other decision variables such as the sensory identity of each option and the motor components of choice, to achieve a comprehensive understanding of the neural representations underlying decision-making. Thus, while our findings establish the ventral striatum as a neural hub where value signals evolve into choice signals, the precise nature of these computations remains to be clarified. Future studies using tasks explicitly designed to dissociate these different forms of value- and choice-related signals—and other decision variables—will be

essential for elucidating how the ventral striatum contributes to the full process leading to action selection.”

2. The authors should also report the results of analysing the second-option period, as this will help clarify the specific information encoded by striatal neurons. For example, action value neurons would code the action of the button press both at first and second stimulus, whereas option coding neurons.

> Response to the above comment

We appreciate the reviewer’s suggestion to analyze neuronal activity during the second object period. Following this advice, we examined how the three neuron types (value-modulated, intermediate, and choice-modulated) identified from the first object period responded to the second object (new Supplementary Fig. 8). To address the reviewer’s example, we compared neuronal activity between trials in which the monkey chose the first object (i.e., did not release the button during the second object period) (right panels in Supplementary Fig. 8a to c) and trials in which the monkey chose the second object (i.e., released the button during the second object period) (left panels in the figures). We observed that neurons of all three types exhibited clear modulation when the monkey released the button, but little modulation when the button was not released.

This modulation pattern likely reflects task context rather than the motor act of button release itself. In trials where the first object was chosen, the second object was unavailable and did not predict future reward, which explains the weak modulation evoked by the second object. By contrast, when the first object was not chosen, the second object remained available and predictive of reward, and neurons of all three types consistently reflected the second object value. If these neurons primarily encoded the motor act of button release, their activity should have increased independently of the second object value, which was not the case.

Because the value-modulated, intermediate, and choice-modulated neuron types were originally classified based on their responses to the first object, the consistent value coding across neuron types for the second object suggests that ventral striatum neurons flexibly adjust their roles depending on context, for example, free-choice situations (when the monkey decided whether to choose the first object) versus forced-choice situations (when the monkey had no alternative but to accept the second object). We have added these results to the revised manuscript as follows

(new line 348-366):

“So far, we have focused on the neuronal modulation evoked by the first object and identified ventral striatum neurons with value-modulated, intermediate, and choice-modulated signals. How do these neurons with different types of signals respond to the second object? We did not observe clear differences in their modulations evoked by the second object across the three neuron groups (Supplementary Fig. 8). When the monkey did not choose the first object, i.e., when the second object was available and predicted the reward outcome, neuronal activity in all three groups reflected the value of the second object (Supplementary Fig. 8a to 8f). The magnitude of this modulation, measured as the regression coefficient between neuronal activity and the second object value, did not differ significantly across the three groups (neurons with positive modulation: $p = 0.41$, $F = 0.90$; neurons with negative modulation: $p = 0.41$, $F = 0.89$; one-way ANOVA) (Supplementary Fig. 8g and 8h). In contrast to the clear modulation evoked by the available second object, when the monkey chose the first object, i.e., when the second object was unavailable and no longer predicted reward, neurons of all groups showed little modulation (Supplementary Fig. 8a to 8h). Together, these results suggest that ventral striatum neurons flexibly change their roles depending on context, such as free-choice situations (when the monkey decided whether to choose the first object) versus forced-choice situations (when the monkey had no alternative but to obtain the reward associated with the second object).”

3. I was surprised that two pioneering studies on action value coding in primate striatum were not cited and discussed (Samejima et al., 2005; Lau and Glimcher, 2008).

> Response to the above comment

Thank you very much for pointing out these landmark studies. As suggested, we now cite Samejima et al. (2005) and Lau and Glimcher (2008) and incorporate them into the discussion of action value and option value. Specifically, we added the following paragraph, which was included in response to Comment 1 above, to the discussion (new line 609-628) (Ref 45 is Samejima et al. (2005), and Ref 46 is Lau and Glimcher (2008)):

“It is also important to note that neuroscience studies on decision-making have further categorized value-related signals into distinct subtypes. For example, neurons in the dorsal striatum have been shown to encode “action value,” the value associated with a potential action^{45,46}, whereas neurons in other brain regions encode “chosen value,” the value of the option ultimately selected^{27,47}. Because the task used in the present study was not

designed to disentangle these subtypes, we cannot determine whether the value-modulated signal observed in the ventral striatum corresponded to action value, chosen value, or another value. Similarly, although we observed choice-modulated signal in the ventral striatum, it remains unclear whether this signal reflects the choice of option or the choice of action. Ideally, task designs should enable the construction of a single regression model incorporating these value- and choice-related variables, as well as other decision variables such as the sensory identity of each option and the motor components of choice, to achieve a comprehensive understanding of the neural representations underlying decision-making. Thus, while our findings establish the ventral striatum as a neural hub where value signals evolve into choice signals, the precise nature of these computations remains to be clarified. Future studies using tasks explicitly designed to dissociate these different forms of value- and choice-related signals—and other decision variables—will be essential for elucidating how the ventral striatum contributes to the full process leading to action selection.”

4. As some striatal neurons showed mixtures of value and choice coding, it would be more appropriate to conduct a multiple regression model that includes both variables and thus allow them to compete directly to explain variance in neuronal activity.

> Response to the above comment

We appreciate the reviewer’s constructive suggestion to use a multiple regression model including both value and choice variables. However, in our dataset, these two variables were highly correlated (see Fig. 1b), making it statistically inappropriate to include them simultaneously in the same regression model. Under such conditions, regression coefficients cannot be reliably estimated, and any apparent variance partitioning between the two variables would be unstable and difficult to interpret.

For this reason, we primarily analyzed the contributions of value- and choice-related signals using a model comparison approach, in which the goodness of fit was compared between models representing value and choice. While we agree that it would be ideal to statistically dissociate the two signals within a single model, the strong correlation between these predictors in our task design makes this infeasible. We have clarified this point in the revised manuscript as follows (new line 205-212):

“To statistically characterize the activity of ventral striatum neurons, we conducted a model comparison analysis, in which we fitted the activity of each neuron with two models representing value and choice (Fig. 2d). We did not use a conventional multiple regression

approach with value and choice as predictors in this main analysis, because these variables were highly correlated (see Fig. 1b), making it difficult to obtain reliable parameter estimates (called “multicollinearity” problem). The model comparison analysis, which is not affected by this issue, has been successfully used in our previous study to characterize neuronal activity in other brain regions²⁸.”

On the other hand, in order to respond to this comment and other reviewers’ comments, we additionally applied a “ridge regression” approach, which can partly, but not perfectly, mitigate the issue of multicollinearity among predictors, to verify the results obtained from the model comparison analysis. This complementary analysis yielded consistent results, as described below (new line 235-254):

“To confirm the validity of this classification, we conducted an additional analysis using “ridge regression”, which can partly mitigate the issue of multicollinearity among predictors. In this model, the firing rate of each neuron was predicted by the value of the first object (value) and whether the monkey released the button to choose the object (choice), with an additional regularization term in the loss function that penalizes large coefficient estimates arising from correlated predictors. This regularization stabilizes parameter estimation and enables a more reliable valuation of each predictor’s influence. We fitted the ridge model separately to the activity of each neuron classified as value- or choice-modulated based on the model comparison analysis. The activity was measured within the time window during which each neuron exhibited the value- or choice-modulated signal. The neuronal activity classified as value-modulated showed significantly larger regression coefficients for value than did the activity classified as choice-modulated ($z = 3.1$, $p = 0.002$; two-tailed Wilcoxon rank-sum test), whereas the neuronal activity classified as choice-modulated showed significantly larger coefficients for choice than did the activity classified as value-modulated ($z = 3.2$, $p = 0.001$; two-tailed Wilcoxon rank-sum test) (Supplementary Fig. 4). These results demonstrate that our model comparison approach reliably dissociates the value- and choice-modulated signals, despite their inherent correlation. The consistency of these findings with the ridge regression further supports the robustness of our model comparison approach.”

Please note that ridge regression approach only partly mitigates the issue of multicollinearity among predictors. We therefore used this approach just to verify our main results obtained by the model comparison approach.

5. It was unclear from the methods description how the critical p-value for the F-test of model fit in the sliding window analysis was determined. This should likely be based on

a similar permutation method as the comparison of R-values but I could not find this information. Did the authors impose a criterion of a critical number of significant F-tests in consecutive sliding windows?

> Response to the above comment

We thank the reviewer for raising this important point. Using the same sliding window procedure, we directly applied the F-test in each window to determine whether the value and choice models explained significant variance in neuronal activity.

Regarding the reviewer's question about the requirement of a minimum number of significant windows in sequence, we required the significance of F-test ($p < 0.05$) to be maintained for at least 40 consecutive windows to avoid spurious results due to noise fluctuations. This criterion is the same as the criterion for the comparison of R-values. These methodological details have now been clarified in the revised manuscript as follows (new line 991-1004):

“We performed this model comparison for each of 150-ms sliding window with a 1-ms step. If a neuron showed a significantly higher R-square for the value model than the choice model ($p < 0.05$; two-tailed Monte Carlo test) for 40 consecutive windows and simultaneously exhibited a significant fit with the value model across the same 40 windows ($p < 0.05$; two-tailed F-test), the activity of this period was classified as a value-modulated signal. Conversely, if a neuron showed a significantly higher R-square for the choice model than the value model ($p < 0.05$; two-tailed Monte Carlo test) for 40 consecutive windows and simultaneously exhibited a significant fit with the choice model across the same 40 windows ($p < 0.05$; two-tailed F-test), the activity of this period was classified as a choice-modulated signal. If a neuron showed no significant difference in R-square between the two models but significant fit with both for 40 consecutive windows, the activity during this period was classified as an intermediate signal. Some neurons represented two or even three signals because their activity was classified as distinct signals during different time periods.”

6. The authors' interpretation that the ventral striatum might directly derive choice from value should be discussed in light of findings that neurons in the orbitofrontal cortex (Padoa-Schioppa and Assad, 2006; Ballesta and Padoa-Schioppa, 2019) and amygdala (Grabenhorst et al., 2012; Grabenhorst et al., 2023) encode subjective values for stimuli, chosen values, stimulus choices, but not necessarily chosen actions (this includes data from sequential decision tasks). Specifically, value-to-choice transitions as reported here

have previously been reported in primate amygdala (Grabenhorst et al., 2012; Grabenhorst et al., 2023). Both OFC and amygdala provide direct inputs to ventral striatum. Thus, it is possible the presently observed signals could be inherited from these structures. Alternatively, if the authors can demonstrate that ventral striatal neurons code the value and choice for action (i.e., the same neurons signalling choice at first and second stimulus), irrespective of stimulus identity, this might add an interesting new aspect to our understanding of value and choice coding in this network

> Response to the above comment:

Thank you very much for pointing out the important studies in the orbitofrontal cortex (OFC) and amygdala, and for suggesting the new analysis to further clarify the principles of value and choice coding in the ventral striatum. In the response to major comment 2, which is described above, we conducted the suggested analysis examining how ventral striatum neurons classified during the first object period responded to the second object. We found that value-modulated, intermediate, and choice-modulated neurons all consistently encoded the value of the second object when it was available and predictive of reward, but showed little modulation when the second object was unavailable and no longer associated with future reward. Importantly, this pattern indicates that ventral striatum neurons flexibly switch between value and choice coding depending on task context. As mentioned in the response to major comment 2, we have added these results to the revised manuscript (new line 348-366).

Regarding to the comment, “*Both OFC and amygdala provide direct inputs to ventral striatum. Thus, it is possible the presently observed signals could be inherited from these structures.*”, we totally agree this possibility. We therefore revised a discussion paragraph to raise this possibility in the revised manuscript, in which the suggested studies are cited, as follows (new line 581-608):

“The neural dynamics we observed in the ventral striatum emphasizes its role as a neural hub that bridges option valuation and action selection. Previous studies have also explored the neural mechanism that bridges option valuation and action selection. These studies mainly dealt with the prefrontal and parietal cortical areas³²⁻³⁴. For example, it has been shown that option value signals are encoded in the ventromedial prefrontal cortex and then transmitted to the dorsomedial prefrontal cortex and the intraparietal sulcus area, where neuronal activity is involved in action selection by comparing the value of each option³⁴. In contrast, we found that neurons in the ventral striatum are involved in both value signaling and action selection. Their activity gradually shifted from value representation to action

selection encoding, suggesting that the ventral striatum may directly derive action selection from value information. At the same time, it is important to note that neurons in the orbitofrontal cortex have been reported to encode value- and choice-related variables in decision-making situation^{27,35}, and that similar value-to-choice transitions have been observed in the amygdala^{36,37}. Because both the orbitofrontal cortex and amygdala project directly to the ventral striatum^{38,39}, the signals we observed in the ventral striatum might in part be inherited from these upstream regions. Particularly, however, the relationship between the ventral striatum and prefrontal regions requires further considerations. Both regions participate in cortico-basal ganglia loop circuits, often referred to as the “limbic loop”^{40,41}, yet little is known about the nature of the signals exchanged between them. Computational studies have suggested that cortico-basal ganglia loop circuits, which sometimes include the dopamine system and amygdala, implement more than simple feedforward transmission of signals and highlighted the importance of iterative and recurrent processing within these circuits⁴²⁻⁴⁴. In light of our findings on the dynamic transformation from value representation to action selection encoding in the ventral striatum, future studies should investigate how such iterative interactions between the ventral striatum and prefrontal regions contribute to decision-making, for example, through simultaneous recordings or causal perturbations of their communication.”

7. The behavioral regression in its current form is perhaps not providing a full picture of the monkeys' decision process: in addition to the value of the first option, the expected value of the second option should also be taken into account. Similarly, the monkeys' decision-making might be influenced by individual risk attitudes. It would be helpful to model these influences on choice in the regression, or, in the case of risk attitude, at least discuss whether this variable might have influenced the monkeys' choices.

> Response to the above comment:

Thank you very much for pointing out the possibility that the expected value of the second object and individual risk attitudes influence the monkey's action selection. We first examined the effect of the expected value. We presume that the monkey's expectation of the second object value depends on its recent experience on the second object. Using a logistic linear regression with the second object value in the previous trial (t-1) and the first object value in the current trial (t), we found that the value of the previous second object affected the monkey's action selection in a manner opposite to the value of the current first object. That is, the higher the value

of the previous second object, the less likely the monkey was to choose the current first object. This opposing effect of the current and previous values suggests that when the previous second object had been highly rewarding, the monkeys anticipated the possibility of another valuable second object and were therefore more inclined to reject the current first object. We have added these results to the revised manuscript as follows (new line 157-169):

“We also found that the monkey’s decision was influenced by recent experience. Specifically, the value of the second object in the previous trial negatively predicted whether the monkey would choose the first object in the current trial. A logistic regression analysis revealed a weak but significantly negative regression coefficient between the value of the previous second object and the choice rate of the current first object ($n = 143$ sessions, $z = -7.2$, $p = 8.4 \times 10^{-13}$; two-tailed Wilcoxon signed-rank test), while the regression coefficient for the current first object value was significantly positive ($n = 143$ sessions, $z = 10.4$, $p = 3.2 \times 10^{-25}$; two-tailed Wilcoxon signed-rank test) (Supplementary Fig. 1b). Thus, the higher the value of the previous second object, the less likely the monkeys were to choose the current first object. This opposing effect of the current and previous values suggests that when the previous second object had been highly rewarding, the monkeys anticipated the possibility of another valuable second object and were therefore more inclined to reject the current first object.”

We next examined whether individual risk attitudes affected choice behavior. In principle, risk-seeking monkeys may reject the first object more often in anticipation of receiving a larger reward from the second object. We therefore quantified risk attitude as the first object value that was chosen with 50% probability, with a higher value indicating greater risk seeking. However, no significant differences in risk attitude were observed across the three monkeys. This suggests that the observed choice patterns were not driven by individual differences in risk preference, but rather reflected a consistent process of option evaluation and action selection within the task. We have added these results to the revised manuscript as follows (new line 130-140):

“Because the task design allowed the monkeys to either commit early to the first object or wait for the second one, their decisions could, in principle, be influenced by individual risk attitudes. For example, risk-seeking monkeys might reject the first object more often in anticipation of receiving a larger reward from the second object. We quantified risk attitude as the value of the first object that was chosen with 50% probability. A higher value indicated a stronger risk-seeking tendency. However, no significant differences in risk attitude were observed across the three monkeys ($n = 54, 71, \text{ and } 18$ sessions for monkeys E, A and M, respectively, $p = 0.07$, $F = 2.78$; one-way ANOVA) (Supplementary Fig. 1a). This indicates

that the observed choice patterns were not influenced by individual differences in risk preference but were instead a consistent consequence of option valuation and action selection within the task.”

8. In addition to value-to-choice transition (better labelled value-to-action transition), the authors should examine other indicators of a decision computation: (i) stronger choice signals for easier decisions (unsigned value difference) compared to difficult decisions; (ii) correlates of value comparison, e.g., transient relationship between firing rate and value difference between options.

Related to this issue: Abstract, line 32: “modulating decision computations”; decision computation has not been sufficiently demonstrated in the study; unless further evidence for decision computation in the neural data is provided, this should be changed to “bias action selection” or something similar.

> Response to the above comment:

Thank you very much for raising this important point. In response to major comment 7 described above, we found that the value of the second object in the previous trial, which could influence the expected value of the second object in the current trial, affected the monkeys’ choice of the current first object. Specifically, the higher the value of the previous second object, the less likely the monkeys were to choose the current first object. This finding suggests that when the previous second object had been highly rewarding, the monkeys anticipated the possibility of another valuable second object and were therefore more inclined to reject the current first object by comparing this first object with the potential second object.

To address the reviewer’s suggestion regarding decision computations, we analyzed trials in which the value of the first object was 4. In these trials, the monkeys sometimes chose and sometimes rejected the first object, reflecting difficult decision situations. In contrast, in other value conditions, the monkeys almost always either chose or rejected the first object. Thus, only the condition with the first object value of 4 allowed us to examine neural correlates of difficult decision as the reviewer suggested.

We calculated the magnitude of the choice-modulated signal evoked by the first object with value 4, separately for each second object value in the previous trial (see the figure below). The magnitude was also shown separately for the 33 choice-modulated neurons with positive modulation (activity increased when the monkey

chose the first object) (a and b in the figure) and the 23 neurons with negative modulation (activity decreased when the monkey chose the first object) (c and d in the figure). The time window to calculate the magnitude (gray shaded area in a and c, 500–1000 ms) was determined based on the averaged activity pattern.

When the previous second object value was small (e.g., value = 1), the decision about whether to choose the current first object with value 4 could be easier (unsigned value difference = 3). In contrast, when the previous second-object value was 4, the decision could be more difficult (unsigned value difference = 0). However, the magnitude of the choice-modulated signal was not significantly influenced by the decision difficulty (i.e., by the previous second object value) for either neurons with positive modulation or those with negative modulation ($p > 0.05$; one-way ANOVA).

We also calculated the correlation between the magnitude of neuronal activity and the value difference for these positively and negatively modulated neurons using trials in the same situation (i.e., first object chosen trials with its value of 4). We selected these trials because the monkey’s action selection fluctuated (i.e., difficult decisions were included). The magnitude of neuronal activity was calculated using the same time window (500–1000 ms). Of the 33 choice-modulated neurons with positive modulation, only 3 neurons exhibited a significant correlation coefficient between the magnitude of neuronal activity and the value difference ($p < 0.05$; correlation analysis). Of the 23 choice-modulated neurons with negative modulation, only 2 neurons

exhibited a significant correlation coefficient ($p < 0.05$; correlation analysis).

These results suggest that ventral striatum neurons did not explicitly represent the value difference between the first object in the current trial and the second object in the previous trial, which could influence the expected value of the second object in the current trial. Behaviorally, we observed that the value of the previous second object affected the monkey's choice of the current first object, indicating that the monkeys may perform such value comparison computations to some extent. However, the behavioral effect of the previous second object value was relatively weak compared with the effect of the current first object value, which may explain the absence of a detectable neural correlate under our recording conditions.

Although we do not plan to include these negative results in the main text due to their limited statistical power and interpretability, we would be willing to include them in the revised manuscript if the reviewer strongly recommends doing so.

Furthermore, to address the reviewer's comment regarding the phrase "*modulating decision computations*" in the abstract, we revised the sentence to more accurately reflect our findings, as follows (new line 29-32):

"Our findings reveal a novel function of the ventral striatum as a neural hub that bridges option valuation and action selection, and demonstrate the contribution of dopamine in the process leading to action selection within this region."

We have also revised related sentences throughout the manuscript accordingly.

Minor comments.

9. Data from neurons recorded from the primate ventral striatum in decision tasks are rare. It is a shame that the authors did not document these neurons in more detail with raster plots, to show the response pattern and recording quality, as they do for the dopamine neuron in Fig. 5c, and for which the authors have a distinguished track record.

> Response to the above comment:

We fully agree with the reviewer that raster plots demonstrate the quality of our data. Accordingly, we have included raster plots in Fig. 2a–2c of the revised manuscript.

10. Fig. 3b should include standard error/deviation in addition to the mean response. It would also be helpful to additionally see these data aligned to the onset of the first stimulus, rather than button press, as this is more consistent with prior studies

indicating stimulus-based choice signals

> Response to the above comment:

As the reviewer suggested, we have included SEM and SDFs aligned at the first object onset (at the go signal onset in the button-release task) in Fig. 3 and Supplementary Fig. 6 of the revised manuscript.

11. Line 149, typo: “stratum”

> Response to the above comment:

Thank you very much for carefully checking our manuscript. We corrected the typo in the revised manuscript.

Reviewer #2 (Remarks to the Author):

Summary

In this study, Neijme and colleagues record and manipulate neural activity in the ventral striatum (VS) of two macaque monkeys to study the region's role in a sequential value-based choice task. They show that the VS has an important role in the transition from option valuation to action selection signals. Two experimental manipulations using electrical stimulation and optogenetics are used to support this finding with causal evidence.

This is an interesting study that examines the role of the ventral striatum in value and choice/action coding using a combination of recording and causal approaches. It involves an impressive set of experiments. It also uses control tasks and is conducted to a high standard overall. I have a few major and minor comments that I believe would strengthen the conclusions of the work.

> Response to the above comment

We are very pleased to hear the supportive comment from the reviewer. Thank you so much.

Major

1. By definition, in real-life, value and choice are correlated, yet the ability to dissociate the two processes is critical for concluding that the VS is responsible for the choice-to-action transformation. How did the authors address this issue? Did they simulate data to show that choice value coding and action coding can be dissociated in their study using their paradigm? It would be good to see evidence of this.

> Response to the above comment

We greatly appreciate the reviewer's insightful comment on this critical issue. In our task, the monkeys were required to decide whether to accept a single option (i.e., the first object) presented on each trial (Fig. 1a). As expected, the higher the value of the first object, the more likely the monkey was to accept it (Fig. 1b). Consequently, the first object value and the monkey's choice were positively correlated. Such situations

naturally arise in real-life decision-making contexts. This correlation, however, poses an analytical challenge: when two explanatory variables are highly correlated, conventional multiple regression analyses cannot reliably estimate their independent contributions because of “multicollinearity”.

To address this issue in our original manuscript, we avoided using a standard multiple regression including both value and choice as predictors in the main analysis. Instead, we employed a model comparison approach, in which each neuron’s activity was separately fitted with two competing models, a value model and a choice model (see the figure below and Fig. 2d in our manuscript), and their explanatory power was statistically compared. This approach allowed us to identify whether neuronal activity was better explained by the value of the first object or by the monkey’s choice, without assuming independence between the two predictors. Using this procedure, we classified the activity of each recorded neuron into three types: value-modulated, choice-modulated, and intermediate signals

As the reviewer noted, this model comparison procedure is original, and therefore its robustness should be validated. To this end, we conducted an additional analysis using “ridge regression”, which can partly address multicollinearity between predictors. The ridge regression model was expressed as:

$$F = \beta_0 + \beta_1 \times V + \beta_2 \times C$$

where F represents the firing rate of each neuron, V denotes the value of the first object, C indicates whether the monkey chose the first object (1 for chosen trials, 0 for unchosen trials). Ridge regression includes a regularization term, $\lambda(\beta_1^2 + \beta_2^2)$, in the loss function, which penalizes large coefficient estimates that can arise from correlated predictors. This regularization effectively stabilizes parameter estimation and allows for a more reliable assessment of each predictor’s relative influence, even

when *V* and *C* are correlated.

We fitted the ridge model separately to the activity of each neuron classified as value- or choice-modulated based on the model comparison analysis. The activity was measured within the time window during which each neuron exhibited the value- or choice-modulated signal. The neuronal activity classified as value-modulated showed significantly larger regression coefficients for value than did the activity classified as choice-modulated ($z = 3.1$, $p = 0.002$; two-tailed Wilcoxon rank-sum test), whereas the neuronal activity classified as choice-modulated showed significantly larger coefficients for choice than did the activity classified as value-modulated ($z = 3.2$, $p = 0.001$; two-tailed Wilcoxon rank-sum test) (see Supplementary Fig. 4).

These results confirm that our model comparison approach reliably distinguishes neuronal signals modulated by the value of the first object and those modulated by the monkey's choice, despite the inherent correlation between the two variables. The consistency of results obtained using ridge regression further supports the validity and robustness of our analytical framework for dissociating value- and choice-modulated signals in ventral striatum neurons. We have included this new analysis and its results in the revised manuscript as follows (new line 235-254):

“To confirm the validity of this classification, we conducted an additional analysis using “ridge regression”, which can partly mitigate the issue of multicollinearity among predictors. In this model, the firing rate of each neuron was predicted by the value of the first object (value) and whether the monkey released the button to choose the object (choice), with an additional regularization term in the loss function that penalizes large coefficient estimates arising from correlated predictors. This regularization stabilizes parameter estimation and enables a more reliable valuation of each predictor’s influence. We fitted the ridge model separately to the activity of each neuron classified as value- or choice-modulated based on the model comparison analysis. The activity was measured within the time window during which each neuron exhibited the value- or choice-modulated signal. The neuronal activity classified as value-modulated showed significantly larger regression coefficients for value than did the activity classified as choice-modulated ($z = 3.1$, $p = 0.002$; two-tailed Wilcoxon rank-sum test), whereas the neuronal activity classified as choice-modulated showed significantly larger coefficients for choice than did the activity classified as value-modulated ($z = 3.2$, $p = 0.001$; two-tailed Wilcoxon rank-sum test) (Supplementary Fig. 4). These results demonstrate that our model comparison approach reliably dissociates the value- and choice-modulated signals, despite their inherent correlation. The consistency of these findings with the ridge regression further supports the robustness of our model comparison approach.”

2. It seems a shame that choice is not modulated in a predictable direction when using optogenetic or electrical stimulation. However, it should be possible to show that the direction was consistent for a given recording site/area. Could you test whether the modulation is consistent across repeat stimulations for a single stimulation site to provide evidence that the choice modulation effects go beyond simply adding noise but show a consistent direction as long as the same location is studied? Related to this, why does it seem as though there an anterior-posterior effect for the electric but not optogenetic experiment? Were the sites comparable in the two experiments or could you discuss why they might differ? Related to this analysis, I was not sure why there are seven sites in figure 4d but in the xyz figure with location there are many more sites? Could you just show the xyz plot for value 4 to match with Fig 4d?

> Response to the above comment

We thank the reviewer for this valuable suggestion. In order to respond to the comment, “*Could you test whether the modulation is consistent across repeat stimulations for a single stimulation site to provide evidence that the choice modulation effects go beyond simply adding noise but show a consistent direction as long as the same location is studied?*”, we performed an additional analysis to test whether the direction of choice modulation was consistent for each single stimulation site. Specifically, we examined whether the positive or negative effects of stimulation were maintained across the early and late halves of the trials for each site. The results showed that the direction of these effects was consistent within individual sites (see Supplementary Fig. 10 for electrical stimulation and Supplementary Fig. 15 for optogenetic stimulation), suggesting that the observed modulation reflects stable site-specific effects rather than random trial-by-trial fluctuations. The revised text reads as follows (new line 398-401 for electrical stimulation and new line 501-505 for optogenetic stimulation; the statistical significance is shown in Supplementary Fig. 10 and 15, respectively):

“We confirmed that the direction of these positive and negative effects was consistent across the early and late halves of the trials for each stimulation site (Supplementary Fig. 10), suggesting that these effects were not due to random trial-by-trial fluctuations in the monkey’s action selection.”

“We confirmed that the direction of these positive and negative effects was consistent across the early and late halves of the trials for each stimulation site (Supplementary Fig. 15), suggesting that these effects were not due to random trial-by-trial fluctuations in the monkey’s

action selection.”

In order to respond to the comment, *“Related to this, why does it seem as though there is an anterior–posterior effect for the electric but not optogenetic experiment? Were the sites comparable in the two experiments or could you discuss why they might differ?”*, we have added a discussion paragraph emphasizing the difference between electrical stimulation (which directly modulates cell body activity) and optogenetic stimulation (which affects dopamine release that modulates synaptic efficacy onto striatal neurons). We also referred to a pharmacological study that inactivated the ventral striatum and reported a similar anterior–posterior gradient, supporting the notion that such functional heterogeneity may extend to decision-related processes. We believe that the sites of electrical stimulation (A23–A28) and those of optogenetic stimulation (A21.5–A25) are largely comparable, although the electrical stimulation covered a slightly more anterior portion of the ventral striatum, which might explain the observed gradient. The newly added discussion paragraph reads as follows (new line 725-745):

“Please note that, although both electrical stimulation of the ventral striatum and optogenetic facilitation of dopamine inputs to this region produced bidirectional effects on the monkey’s action selection, only the electrical stimulation exhibited a weak but significant anterior–posterior gradient. The positive effects tended to appear in the anterior portion, whereas the negative effects were more pronounced in the posterior portion. A recent pharmacological study in macaque monkeys also reported functional heterogeneity along this axis of the ventral striatum, showing that inactivation of its anterior portion induced hypoactive, “resting” behavior, whereas inactivation of the posterior portion elicited compulsive-like “checking” behaviors⁵⁶. Our results raise the possibility that such anterior–posterior functional differentiation may also extend to decision-related processes. However, it remains unclear why optogenetic facilitation of dopamine inputs did not exhibit a similar gradient. One possible explanation is that electrical stimulation and pharmacological manipulation directly modulate the cell body activity of ventral striatal neurons, whereas optogenetic facilitation affects dopamine release from axon terminals. Since dopamine release mainly modulates synaptic efficacy onto striatal neurons rather than their intrinsic excitability, the anterior–posterior gradient observed with electrical stimulation might reflect the influence of non-dopaminergic afferents that directly impact neuronal firing. Additionally, the electrical stimulation sites (A23–A28) covered a slightly more anterior portion of the ventral striatum compared with the optogenetic facilitation sites (A21.5–A25), which might have contributed to the observed gradient.”

The reviewer raised the additional concern, *“Related to this analysis, I was not*

sure why there are seven sites in figure 4d but in the xyz figure with location there are many more sites? Could you just show the xyz plot for value 4 to match with Fig 4d?"

We completely agree that the original xyz plot could be confusing to readers. In the initial version, both significant and non-significant data were plotted in the same format without explicit distinction. This was because we assumed that the statistical significance at individual sites might have been underestimated due to the limited number of analyzable trials (approximately 25–50 trials per site in which the first object with value 4 was presented and electrical stimulation was delivered). Therefore, we decided to display all data equally in order to illustrate the overall trend of the spatial distribution of stimulation effects. Accordingly, our main focus was placed on the population-level analyses shown in Figures 4f and 4g, and Figures 5i and 5j. However, as the reviewer rightly pointed out, the lack of visual distinction between significant and non-significant data could cause confusion and create an apparent discrepancy between the number of sites shown in Figure 4e and 5h and the xyz plot. To address this concern, we have revised the xyz plots so that significant and non-significant sites are now clearly differentiated, as shown in the updated versions of Supplementary Figures 9a and 14a.

3. I am not an expert in optogenetics stimulation, but it is commonly discussed that this technique comes with significant challenges in macaque monkeys/larger NHPs. What was done to overcome these limitations here? How can we be sure that the conclusions can be trusted? Relatedly, could the animals (for either stimulation method) possibly tell when the stimulation was on? Or if not, how can you rule out simple confound effects of the stimulation?

> Response to the above comment

We appreciate the reviewer's important point regarding the challenges of applying optogenetic stimulation in macaque monkeys. We agree that this issue was not sufficiently explained in the original manuscript. One of the major difficulties in macaques, compared with rodents, is the relatively low infection efficiency of viral vectors commonly used in optogenetics. To address this limitation, a research group that includes authors of the present study developed AAV2.1, a viral vector that efficiently infects neurons in the macaque brain and achieves robust expression of target genes. In this study, we used this vector system. We have added this information to the revised manuscript, citing the original report of AAV2.1 (new line

907-909):

“AAV2.1 is a viral vector developed by a research group that includes authors of this study⁵⁷. It efficiently infects neurons in macaque monkeys and enables high-level expression of target genes in these cells.”

The reviewer also raised an important concern about whether the animals could detect when stimulation was applied, for example through unintended leakage of the stimulation laser light into the experimental room. Indeed, if the laser were simply turned on, light could weakly leak at the connection between the stimulation probe (optrode) and the optical fiber (see the following picture).

To prevent this, we covered the connection with a black tube and wrapped the entire assembly with a black sheet, as illustrated in the following pictures. These precautions ensured that no light leakage was visible in the experimental room. We therefore believe that confounding effects due to light leakage can be ruled out, and we are not aware of other possible confounds related to the stimulation.

4. In Fig.2i, it would be interesting to have more information of the temporal dynamics.

For example, did most of the transitions from value or intermediate neurons to choice neurons happen in the late period? Is it possible to provide more detail on how the three types of neurons switch over time? These dynamics seem to be at the core of what the paper can contribute.

> Response to the above comment

We thank the reviewer for this valuable and insightful suggestion. Following the reviewer's advice, we examined the temporal dynamics of the transitions among the three functional neuron types by calculating the latencies of signal transitions from value to intermediate, from intermediate to choice, and from value to choice (see the figure below). The transition latency was defined as the start time of each second signal.

As expected, we observed that the transition latency was shorter for value-to-intermediate transitions than for intermediate-to-choice or value-to-choice transitions (see the vertical dotted line in each figure, indicating the mean \pm SD of the latency). However, these differences were small and did not reach statistical significance ($p > 0.05$, Wilcoxon rank-sum test).

The most likely reason for the lack of statistical significance is the limited number of neurons that exhibited these signal transitions. While the observed tendency is consistent with a sequential flow of information from value to choice representations, the current dataset does not allow a definitive temporal characterization of these transitions. Although we do not plan to include this negative result in the main text due to the limited sample size, we would be happy to incorporate the detailed analysis and corresponding figure in the revised manuscript if the reviewer considers it important for completeness.

5. Finally, does the stimulation of VS (with either electrical or optogenetic methods)

influence decision reaction times (RTs) and is there any correlation between Δ Release latency in the button-release task and Δ decision RT in the decision task (Figs 4 & 5)?

> Response to the above comment

We appreciate the reviewer's insightful comment regarding the potential effects of ventral striatum (VS) stimulation on decision reaction time (the latency of button release in the decision-making task). To address this point, we analyzed the effects of both electrical and optogenetic stimulations on the button-release latency to choose an option in the decision-making task.

For electrical stimulation, we found that stimulation did not clearly affect button-release latency. Here we analyzed the button-release latency only for the first object with value 4 because the electrical stimulation affected the monkey's action selection only for this value condition. Although a marginally significant difference was observed at the population level, the magnitude of this effect was small, and only one stimulation site showed a significant effect. Furthermore, there was no significant correlation between the effect on the button-release latency and the effect on the monkey's action selection. These results suggest that electrical stimulation of the ventral striatum may weakly influence the latency to reach a decision, but its effect (if any) is minimal and independent of which action the monkey selects. We have added these data to the revised manuscript as follows (new line 441-452):

“Even in the decision-making task, electrical stimulation did not clearly affect button-release latency (Supplementary Fig. 13a). Although the latency difference between stimulation and non-stimulation trials reached marginal significance at the population level ($n = 56$ sites, $z = -2.45$, $p = 0.0144$; two-tailed Wilcoxon signed-rank test), the difference was small, and only one of the 56 stimulation sites showed a significant effect ($p < 0.05$; two-tailed Wilcoxon rank-sum test). Moreover, there was no significant correlation between the button-release latency difference in the decision-making task and the choice-rate difference for the first object with value 4 ($n = 56$ sites, $r = 0.08$, $p = 0.54$; two-tailed Pearson's correlation test) (Supplementary Fig. 13b). These results suggest that electrical stimulation of the ventral striatum might weakly influence the latency to reach a decision, but its impact (if any) is minimal and independent of which action the monkey selects.”

Similarly, optogenetic stimulation of dopamine projections did not significantly alter button-release latency in the decision-making task, and no individual stimulation sites exhibited a significant effect. There was no significant correlation between the effect on the button-release latency and the effect on the monkey's action selection. We have added these data to the revised manuscript as follows (new line 536-545):

“As in the button-release task, laser light stimulation did not affect button-release latency in the decision-making task (Supplementary Fig. 18a). The latency difference between stimulation and non-stimulation trials was not significant ($n = 50$ sites, $z = -0.56$, $p = 0.57$; two-tailed Wilcoxon signed-rank test), and no stimulation sites showed a significant effect ($p < 0.05$; two-tailed Wilcoxon rank-sum test). Moreover, there was no significant correlation between the button-release latency difference and the choice-rate difference for the first object with value 4 ($n = 50$ sites, $r = -0.03$, $p = 0.86$; two-tailed Pearson’s correlation test) (Supplementary Fig. 18b). These results suggest that laser light stimulation of dopamine projections does not influence the latency to reach a decision, regardless of its effect on the monkey’s action selection.”

Finally, as the reviewer suggested, we examined “the correlation between Δ release latency in the button-release task and Δ decision RT in the decision-making task”. No significant correlation was found for either electrical or optogenetic stimulation (see the figure below). This result is consistent with the absence or weakness of stimulation effects on button-release latency in both tasks. We did not include this additional result in the revised manuscript because they are largely redundant with the analyses already presented, but we would be happy to include them if the reviewer considers them useful for completeness.

Minor

1. Several studies have demonstrated the role of the VS in cost-benefit decision making,

e.g., VS lesions or VS dopamine depletion reduces animals' likelihood of choosing a high-effort choice option (Hauber & Sommer, 2009; Nicola, 2010; Salamone et al. 1994). Does this relate at all to the ideas presented in this manuscript or if not, why not?

> Response to the above comment

We thank the reviewer for raising this important point and for referring to studies demonstrating the involvement of the ventral striatum (VS) in cost–benefit decision making (Hauber & Sommer, 2009; Nicola, 2010; Salamone et al., 1994). We fully agree that these studies are highly relevant to our research in highlighting the VS's role in regulating choice behavior.

However, our study focuses on a different aspect of decision making, specifically, the transformation from option valuation to action selection. To investigate this process, it is essential to record and manipulate neuronal activity with precise temporal control during the decision period itself. In contrast, studies employing VS lesions or dopamine depletion lack such temporal specificity and thus cannot directly address the dynamic neural mechanisms that mediate value-to-action transformation during an ongoing decision. Instead, these studies seem to primarily assess the effects on general motivation, which can manifest independently of specific decision timing.

It is widely accepted that dopamine neurons convey reward prediction error (RPE) signals, which are elicited by outcomes after a decision and are thought to influence future decision processes. The studies cited by the reviewer do not exclude the possibility that the observed behavioral effects partly reflect alterations in such dopamine-mediated learning or motivational mechanisms, because these studies lack neuronal recording and manipulation techniques with precise temporal resolution. By contrast, our work investigates how the dopamine–VS system influences the ongoing decision process itself, before any reward feedback occurs. In this sense, our study addresses a complementary but conceptually distinct question regarding the immediate, within-trial contribution of dopamine to decision formation.

Because our main objective is to elucidate this temporally specific mechanism, we have not included this discussion in the revised manuscript. However, if the reviewer considers that explicitly contrasting our findings with prior lesion or dopamine depletion studies would strengthen the manuscript's conceptual framing, we would be happy to add a brief discussion highlighting this distinction in the revised version.

2. Please specify in the Methods which pairings of objects were used how frequently.

> Response to the above comment

This information is important for readers to understand the task design. We have added it to the Methods section of the revised manuscript as follows (new line 827-830):

“Both the first and second objects were pseudo-randomly selected from the six visual objects with equal probability and independently of each other, so that the same object could appear as both the first and second objects on a given trial.”

3. Please plot the data from each monkey separately or in separate lines in the key figures.

> Response to the above comment

We appreciate this valuable suggestion. Showing the data for each monkey is indeed important for demonstrating the reproducibility of our findings. In the revised manuscript, we have now presented key figures separately for each monkey, illustrating the proportions of neurons encoding value-modulated, choice-modulated, and intermediate signals (Supplementary Fig. 3a and 3b), as well as the latencies of these signals (Supplementary Fig. 3c and 3d). These data are critical for demonstrating the transition from option valuation representation to action selection encoding in the ventral striatum.

In addition, we have separately plotted, for each monkey, the effects of electrical and optogenetic stimulations (Supplementary Fig. 11 and Supplementary Fig. 16, respectively), which are essential to show the causal contributions of the ventral striatum and dopaminergic inputs to decision-making. We found that the results were consistent across monkeys, supporting the robustness and reproducibility of our conclusions.

4. While it is brilliant to see that control experiments using simple motor tasks were performed, there is more than one difference between the task of interest and the control task. For example, in the simple motor task, there is no choice, no value processing and expectation, no modulation of outcome size and thus overall less reward on average. It

would be good to acknowledge and discuss these differences in the discussion.

> Response to the above comment

We thank the reviewer for this insightful comment. We fully agree that the control task differed from the decision-making task in several respects beyond the presence or absence of choice, and that these differences could influence the observed neuronal modulation. The control task was intentionally designed to remove the choice process in order to isolate motor-related neural activity in the ventral striatum. However, in the decision-making task, the identity and value of the first object varied unpredictably across trials, introducing uncertainty about which option would appear and what reward could be obtained. This trial-by-trial uncertainty likely engaged reward expectation and prediction error processes that were absent in the control task, where the stimulus–reward association was fixed and fully predictable. Such differences in uncertainty and reward expectation may explain why neuronal modulation was stronger in the decision-making task.

We also note that the reward structure was comparable across the two tasks. The reward amount per trial in the control task was set to be equivalent to value 3 or 4 in the decision-making task, matching the expected reward magnitude in the decision-making task (3.5). Because each trial in the control task was shorter, the overall reward rate was actually higher, which could lead to greater motivation. Therefore, the weaker neuronal modulation in the control task is unlikely to be due to reduced motivation. We have added these discussions to the revised manuscript as follows (new line 629-648):

“We confirmed that the choice-modulated signal discussed above was not simply caused by the motor action (i.e., button release) associated with choosing the first object, using the control task in which the monkey was required only to release the button. This control task was designed to eliminate the process of choice in order to isolate motor-related neuronal activity in the ventral striatum. However, it also differed from the decision-making task in other important respects. In the decision-making task, the identity and value of the first object presented as an option varied unpredictably across trials, introducing uncertainty about which object would appear and what reward could be obtained. This trial-by-trial uncertainty likely evoked reward expectation and prediction error processes that were absent in the control task, where the stimulus–reward association was fixed and fully predictable. Such differences in uncertainty and reward expectation may account for the stronger neuronal modulation observed during the decision-making task. In addition to these differences, we also note that the reward structure was comparable across tasks.

The reward amount per trial in the control task was set to be equivalent to value 3 or 4 in the decision-making task, which matched the expected reward magnitude in the decision-making task (3.5). However, because the duration of each trial in the control task was shorter, the overall reward rate was actually higher, potentially leading to greater motivation. Therefore, the weaker neuronal modulation observed in the control task is unlikely to reflect reduced motivation.”

5. In Fig.1, please show the duration for each stage of the task in Fig.1a.

> Response to the above comment

We have added the durations in Fig. 1a of the revised manuscript.

6. Why did the authors pick ‘release’ as their response modality?

> Response to the above comment

We appreciate the reviewer’s question regarding the choice of “release” as the response modality. In the early stages of training, the monkeys were initially trained to make their choice by pressing a button. However, under this design, they sometimes attempted to press the button again when a highly valuable second object appeared, even after having already chosen the first object. To prevent such double-choice behavior and to ensure a clear and unambiguous response, we adopted a “button release” as the choice action. Once the monkey releases the button, it cannot make a second release within the same trial, which effectively enforces a single, temporally well-defined choice. This modification greatly improved task performance.

Reviewer #3 (Remarks to the Author):

This manuscript addresses an important and timely topic: how the ventral striatum contributes to value-based decision-making. The authors used a combination of single-unit recordings, electrical stimulation, and optogenetics in nonhuman primates. The monkeys performed a sequential choice task between pairs of visual stimuli associated with different amounts of reward, and the authors report that ventral striatum activity transitions from value encoding to action selection. They further show that stimulation of the ventral striatum or its dopaminergic inputs can bias choice behavior. The multi-method approach is a strength, and the effort to dissect decision-making phases is valuable. However, several key conceptual and interpretive issues need to be addressed. In particular, the study is often framed within a reinforcement learning framework, but this is not well supported by the task design or data. Sample sizes of recorded neurons are relatively small, and the observed effects of stimulation, while interesting, are also limited. Some analyses can be improved. I believe some of these concerns can be addressed through reframing and clarification, as well as the introduction of further analysis. Below are specific comments. I hope the authors find them useful.

> Response to the above comment

We are very pleased to hear the supportive comment and appreciate the following thoughtful comments, which have helped us improve our study.

Major points:

1) Figure 2 shows firing rate differences that may reflect variations in reaction times, as well as other task/behavioral variables, rather than “value.” Have the authors tested for correlations between firing rates and reaction times? In general, reaction times should be considered as a covariate or explanatory variable in the analysis.

The author can, for example, build a single regression model to predict the neuronal activity of single neurons over time based on different predictors, including reaction times, reward amount, and choice, and estimate the variation in coefficients for each predictor over time, as well as the fraction of neurons that are significant. This would give better estimate of what task/behavioral variable influence neuronal activity. It would be actually also interesting to include the chosen stimulus identity in the model. I think this approach can be helpful for assessing temporal transitions in encoding,

which is exactly what the authors claim is happening (from value to choice over time).

> Response to the above comment

We thank the reviewer for this important and constructive comment. To examine whether the observed neuronal modulation could be explained by reaction time (RT), we conducted an additional analysis using “ridge regression”, a regularized regression method that can partly mitigate the problem of multicollinearity between correlated predictors. In our dataset, the first object value and RT (the latency of button release to choose the first object) were correlated (Fig. 1c), making it statistically inappropriate to include both in a standard multiple regression model. Ridge regression introduces a small penalty term to the loss function, stabilizing coefficient estimation and allowing a more reliable assessment of each predictor’s contribution under such correlated conditions.

The results showed that the influence of RT on neuronal activity was minimal, confirming that the value-related modulation reported in Figure 2 cannot be explained by variations in reaction time. The corresponding description has been added to the revised manuscript (new line 255-270):

“Using another ridge regression, we tested whether the modulation of each neuron could be accounted for by the variation in reaction time (i.e., the latency of button release). Here, the firing rate of each neuron was predicted by the value of the first object (value) and the button-release latency (latency), which were also correlated (Fig. 1c). We excluded the predictor of whether the monkey released the button to choose the first object (choice) from this analysis, because the button-release latency existed only when the monkey released the button (i.e., when the monkey did not release the button, making it impossible to include such trials in the regression analysis). Whereas the proportion of neurons with a significant regression coefficient for value was significantly higher than chance for all calculation time windows (each 150-ms window after the first object onset) except immediately after the onset ($p > 0.05$; two-tailed Monte Carlo test) (Supplementary Fig. 5a), the proportion of neurons with a significant regression coefficient for latency was very close to chance over time, slightly above for some time windows ($p < 0.05$; two-tailed Monte Carlo test) but below for others ($p > 0.05$; two-tailed Monte Carlo test) (Supplementary Fig. 5b). These results suggest that the influence of button-release latency on neural activity, if any, was minimal.”

The reviewer further suggested building a single regression model including multiple behavioral and task variables (e.g., reaction times, reward amount, choice, and stimulus identity) to better assess the temporal dynamics of encoding. We fully agree that such a unified model could provide valuable insights. However, several

aspects of our task design and data structure make it difficult to include all these predictors in a single model without introducing severe statistical and interpretational issues. First, reward value and choice were strongly correlated in our task (Fig. 1b), resulting in severe multicollinearity, which prevents reliable estimation of regression coefficients and interpretation of their relative contributions. Second, reaction time (RT) existed only in trials where the monkey released the button to accept the first object, and thus RT and choice could not be included simultaneously, as reject trials had no measurable RT. Third, because each visual object was uniquely associated with a specific reward amount, the “stimulus identity” and “reward value” variables were perfectly confounded, making it impossible to separate their effects statistically.

Despite these inherent limitations, our task design offers a critical advantage. In this task, option valuation, action selection, and action execution occurred in close temporal succession during the presentation of the first object. This structure provided a unique opportunity to examine the neural mechanisms underlying each of these processes and the temporal transitions between them within a single, continuous decision period. Thus, while statistical separation of certain behavioral variables was not feasible, this tight coupling between valuation and action processes allowed us to capture the natural evolution of neural signals from value representation to action selection coding—precisely the phenomenon our study aims to characterize.

We again agree that the single regression model including multiple behavioral and task variables suggested by the reviewer could, in principle, provide further insights into the neuronal representations underlying decision-making. Therefore, despite the constraints, we performed two complementary analyses designed to address the reviewer’s concern while maintaining statistical validity:

- (1) As described above, we examined whether neuronal modulation could be explained by variations in RT using a ridge regression model that included value and latency as predictors. The results showed minimal RT influence on neuronal activity.
- (2) We also tested whether neurons represented the value of the second object in the previous trial, in addition to the value of the first object in the current trial and the monkey’s choice, to assess whether past valuation processes contributed to current decision-related activity, using a ridge regression model that included these variables as predictors. At the behavioral level, the value of the previous second object influenced the monkey’s decisions (Supplementary Fig. 1b): the higher the value of the previous second object, the less likely the monkeys were to choose the current first object. This suggests that when the previous second

object had been highly rewarding, the monkeys anticipated the possibility of another valuable second object and therefore more inclined to reject the current first object. The ridge regression analysis showed that while the proportions of neurons significantly encoding the current value and choice were above chance, the proportion for the previous value remained close to the chance level, indicating weak representation of the previous second object value. The relevant description has been added to the revised manuscript as follows (new line 322-347):

“To understand the neural mechanisms underlying option valuation, action selection, and the transition between them, we have analyzed neuronal activity encoding the value of the first object and whether the monkey decided to release the button to choose this object during the decision-making period (i.e., during the presentation of the first object). However, other neural processes contributing to decision formation might also occur in the ventral striatum during the same period. For example, we observed that not only the value of the first object, but also the value of the second object in the previous trial influenced the monkey’s action selection (Supplementary Fig. 1b), suggesting that valuation or maintenance of the previous second object might also take place in the ventral striatum during decision making. To test this possibility, we conducted another ridge regression analysis in which the firing rate of each neuron was predicted by the value of the previous second object (previous value), as well as the value of the current first object (current value) and whether the monkey chose the first object (choice). As expected from the model comparison analysis, the proportions of neurons with significant regression coefficients for current value and choice increased after the first object onset and became significantly higher than chance ($p < 0.05$, two-tailed Monte Carlo test) (Supplementary Fig. 7a and 7b). On the other hand, the proportion of neurons with a significant regression coefficient for previous value was close to the chance level over time, slightly above for some time windows ($p < 0.05$; two-tailed Monte Carlo test) but below for others ($p > 0.05$; two-tailed Monte Carlo test) (Supplementary Fig. 7c), suggesting that the ventral striatum may not strongly represent or maintain the value of the previous second object. However, at the behavioral level, the influence of the previous second object value on the monkey’s choices was much weaker than that of the current first object value. Therefore, it may be difficult to detect neuronal activity corresponding to such a weak behavioral effect of the previous second object value in the ventral striatum.”

Both of the above analyses used ridge regression. While ridge regression cannot fully eliminate multicollinearity, its regularization term allows us to “partially” mitigate this problem and obtain more stable coefficient estimates under correlated conditions.

For this reason, these analyses are presented as complementary and supportive, rather than definitive variance-partitioning tests. Nonetheless, the consistency of the ridge regression results with those obtained from the model comparison analysis reinforces the robustness of our main conclusions.

In these ridge regression analyses, we presented the proportion of neurons with a significant regression coefficient for each predictor, but not the coefficients themselves. We did not display the regression coefficients because individual neurons exhibited both positive and negative coefficients, reflecting heterogeneous coding directions (e.g., some neurons increased their activity with higher value, while others decreased it). Averaging such coefficients across neurons would therefore cancel out meaningful effects and obscure the underlying coding diversity. In addition, since the main conclusions of this analysis were already well supported by the proportions of significantly modulated neurons, presenting the regression coefficients was not necessary for further interpretation. However, we would be happy to provide these coefficients if the reviewer considers them informative.

Recognizing the importance of the reviewer's suggestion, we have also noted this point in the revised Discussion section. As described there, although our task design makes it difficult to include all relevant variables (e.g., value, choice, reaction time, and stimulus identity) within a single regression model due to strong correlations and missing data in certain conditions, we fully agree that such an approach would provide a more comprehensive understanding of the neural representations underlying decision-making. Accordingly, we added the following statement to the Discussion (new line 618–624):

“Ideally, task designs should enable the construction of a single regression model incorporating these value- and choice-related variables, as well as other decision variables such as the sensory identity of each option and the motor components of choice, to achieve a comprehensive understanding of the neural representations underlying decision-making. Thus, while our findings establish the ventral striatum as a neural hub where value signals evolve into choice signals, the precise nature of these computations remains to be clarified.”

2) I appreciate the ambition of the authors to connect their findings to a broader theoretical framework, but this is not a reinforcement learning task. Monkeys are basically choosing between overlearned stimuli. Thus, there is no learning. I think this is a very important point. The manuscript repeatedly attempts to position the findings within a reinforcement learning framework, but this interpretation is not adequately supported by the data. As a result, several

claims appear overextended, which diminishes the overall strength of the study. A more cautious, data-driven interpretation would improve the clarity and impact of the work.

> Response to the above comment

We appreciate the reviewer’s insightful comment. We agree that our task does not involve a learning component, and therefore it is not appropriate to directly interpret our findings within a reinforcement learning framework. In line with this concern, we have removed suggestions, discussions, and conclusions that explicitly linked our data to reinforcement learning. Below, we provide examples of the revisions we made in response to the reviewer’s points.

Some examples from the manuscript:

- “Notably, the transitions, encoding from state value to action value and further to action, were often observed within single neurons, rather than across segregated neuronal populations.” The concepts of state value and action value are not applicable in this task.

> As the reviewer correctly noted, the concepts of state value and action value are not applicable to this task. We have deleted the paragraph in the discussion section in which we referred to the relationship between our findings and the reinforcement learning framework, including the terms “state value” and “action value.” These terms have also been removed entirely from the manuscript.

- “This finding challenges the classical notion that the ventral striatum primarily encodes value-related signals in the reinforcement learning framework”. It is unclear how the present results challenge the encoding of value-related signals in the VS, especially given the lack of learning.

> We have deleted the discussion paragraph including this sentence.

- “Instead, our data suggest a broader role of the ventral striatum potentially as a neural substrate not only for value representations, but also for some aspects of policy.” This seems a bit vague. What specific aspects of “policy” are meant here?

> We thank the reviewer for pointing out this issue. In the original manuscript, we used the term “policy” in the context of reinforcement learning to suggest that ventral

striatal activity might relate to decision strategies beyond value representation. However, as the reviewer correctly noted, our task does not involve learning, and thus the use of this term was misleading. To avoid overinterpretation, we have deleted the discussion paragraph containing this sentence, and we have removed any mention of “policy” from the manuscript.

- Also related, the authors often refer to reward prediction errors, which cannot be measured in the present study because animals were likely overtrained, and the reward was not probabilistic.

> We agree with the reviewer’s assessment that reward prediction errors cannot be properly evaluated in our task, as the animals were overtrained and the reward schedule was deterministic. In the revised manuscript, “reward prediction error” appeared only in a general context because dopamine neurons are widely known to encode reward prediction errors. We did not intend to link our own results directly to reward prediction error signals. We now restrict our discussion to interpretations that are directly supported by our data.

3) The effects of ventral striatum and dopaminergic neuron stimulation were absent on over 80% of trials and appeared restricted to conditions of uncertainty. This limited efficacy should be acknowledged and discussed, as it has implications for interpreting the causal role of these signals. I think that the weak effects are likely due to the absence of learning in the task. Therefore, it is unclear to what extent these results can be generalized.

> Response to the above comment

We thank the reviewer for this thoughtful comment regarding the limited effects of electrical and optogenetic stimulations and their potential relationship to learning. We agree that the selective effects observed under uncertain conditions have important implications for interpreting the causal role of the ventral striatum and dopamine inputs and for the generalizability of our findings. We have now included the following paragraph in the Discussion to address this issue (new line 746-763):

“Both electrical stimulation and optogenetic facilitation of dopamine inputs influenced the monkey’s action selection specifically when the monkey was uncertain about whether to choose the medium-value first object (value = 4). When the object value was sufficiently

high or low, the decision process became more deterministic and thus less susceptible to external perturbations. This stability likely reflects the fact that the monkeys had already learned, through extensive training, the associations between each visual object and its corresponding reward outcome. By contrast, during ambiguous decisions, the neural circuits integrating value and choice signals are likely to be in a transient, metastable state or susceptible phase, in which brief perturbations can more effectively bias the final behavioral outcome. Indeed, the medium-value first object elicited intermediate-level responses in ventral striatum neurons showing value-modulated activity. It is also possible that if the associations between visual objects and reward outcomes had not been well learned, the effects of electrical stimulation and optogenetic facilitation of dopamine inputs might have been broader and stronger. Future studies combining electrical or optogenetic stimulation with learning paradigms or systematically varying the level of decision uncertainty could further elucidate the specific computational roles of the dopamine-ventral striatum system in adaptive decision-making.”

This additional discussion explicitly acknowledges the limited efficacy of stimulation effects and clarifies their potential dependence on both learning and decision uncertainty. While our current findings highlight how the dopamine–ventral striatum system contributes to ongoing decision-making under well-learned conditions, future studies manipulating learning state or uncertainty level will be essential to determine how broadly these mechanisms generalize across different behavioral contexts.

4) I think the authors should clearly state that the monkeys were overtrained and that there was no learning in the task. This clarification is critical for contextualizing the neural signals and avoiding overinterpretation in terms of reinforcement learning.

> Response to the above comment

We appreciate the reviewer’s helpful suggestion. We fully agree that clarifying the training status of the monkeys is important to appropriately contextualize the observed neural signals and to avoid overinterpretation in terms of reinforcement learning processes. As suggested, we have now clearly stated in the revised manuscript that the monkeys were overtrained and that no further learning occurred during the task. The following sentence has been added to the revised manuscript (new line 835-839):

“The monkeys were overtrained to perform this decision-making task until their action

selection became stable across daily sessions, ensuring that the associations between each visual object and its corresponding reward outcome were well learned. Therefore, no further learning was expected to occur during the recording or stimulation sessions.”

We also included additional discussion regarding the learning state, as suggested in the previous review comment (see our response to Comment 3).

Minor points:

1) I do not understand the following sentence and the data to support it: “During this period, we observed the signal dynamics of ventral striatum neurons corresponding to the time course of the decision-making process.” (line 260-261)

> Response to the above comment

We thank the reviewer for pointing out this unclear sentence. We agree that the original wording was ambiguous and did not clearly convey our intended meaning. To improve clarity, we have revised the sentence as follows (new line 373-376):

“During this period, we found that ventral striatum neurons exhibited value-modulated, intermediate, and choice-modulated signals, which dynamically evolved as the decision process progressed from option valuation to action selection.”

2) Please clarify how many electrodes were used per session, how many recording sessions were included in the analysis, and whether ventral striatal activity differed depending on whether recordings were made in the hemisphere ipsilateral or contralateral to the arm used to perform the task.

> Response to the above comment

We thank the reviewer for this helpful request for clarification. We used one single-channel electrode in each daily experiment. Recording of one neuron was counted as one recording session. Therefore, zero to a few recording sessions (i.e., zero to a few neurons) were conducted per daily experiment. In total, 125 recording sessions were performed (corresponding to 125 neurons), and these data were included in the analyses. We have added this information to the revised manuscript as follows (line 878-882):

“During each daily experiment, one electrode was inserted into the ventral striatum.

Recording of one neuron was counted as one recording session. Therefore, zero to a few recording sessions (i.e., zero to a few neurons) were conducted per daily experiment. In total, 125 recording sessions were performed (corresponding to 125 neurons), and these data were included in the analyses.”

Regarding the question about ipsilateral and contralateral differences, we conducted single-unit recordings (as well as stimulation experiments) exclusively in the hemisphere contralateral to the arm used to perform the task (as described on new line 843-844 of the manuscript). The contralateral hemisphere was targeted because it is generally considered to have a stronger functional coupling with motor output. Even within the ventral striatum, Sawada et al. (2015) *Science*, which is cited in our manuscript, reported that the ventral striatum contributes to effortful motor behaviors (precision gripping in animals with spinal cord injuries) by integrating motivational drive with motor coordination, and that this effect was observed only on the contralateral side. Therefore, we did not examine hemispheric differences between contralateral and ipsilateral recordings in this study.

However, as far as we know, there is currently no evidence that the ventral striatum does not contribute to ipsilateral action selection in decision-making contexts. Future studies directly comparing both hemispheres will be valuable to determine whether similar or distinct patterns of neural activity and causal effects are present in the ipsilateral ventral striatum. Although we have not included this discussion in the revised manuscript to maintain focus on the main findings, we would be happy to incorporate a brief statement on this point if the reviewer considers it important for completeness.

3) Caption in Figure 1: “Error bars in (b) and (c) indicate SEM, which are very small and hidden in most cases”. I really cannot see any error bar. Maybe reducing the size of the black circles can help?

> Response to the above comment

We reduced the size. Now, the error bars are visible.

Reviewer #4 (Remarks to the Author):

Nejime and colleagues investigated the role of the primate ventral striatum in value-based decision-making, specifically examining its function in the transformation of option valuation into action selection. The authors challenge the traditional view that the VS primarily encodes expected value, proposing instead a more active role in the decision process itself.

The key findings are:

VS neurons exhibit dynamic encoding, initially representing the value of an offered option (offer 1) and subsequently shifting their activity to reflect the monkey's upcoming choice (to accept or reject offer 1). This transition often occurs within individual neurons.

Both electrical stimulation of the VS and optogenetic facilitation of dopamine input to the VS during the decision period (presentation of offer 1) significantly altered the monkeys' choices, particularly when they were uncertain (i.e., for options of intermediate value). These manipulations could bias choices both towards and away from selecting the option, depending on the specific stimulation site.

Control experiments suggest these effects are not merely due to influences on motor execution.

The authors conclude that the VS acts as a crucial neural hub that actively bridges option valuation and action selection, with dopamine playing a critical role in modulating these decision computations in real-time, extending beyond its classical role in reward prediction error signaling.

The idea that the VS itself might be involved in deriving the choice, rather than just passing value information to other regions, is particularly intriguing. The study employs a sophisticated combination of state-of-the-art techniques in non-human primates. Optogenetics in primates is still challenging, and its successful application here to dissect the role of dopamine is commendable.

While this study addresses an important question in decision neuroscience and employs sophisticated methodologies, I have substantial concerns outlined as follows.

> Response to the above comment:

We sincerely appreciate the following thoughtful comments, which have helped us improve our study.

Major comments :

1. The authors state that the task temporally separates option evaluation, decision commitment, and motor execution. However, during the presentation of the first object (offer 1), the animals must evaluate offer 1, make an accept/reject decision, and execute a motor response (hold or release the button). This conflation of evaluation, decision, and motor execution for offer 1 should be acknowledged more clearly. The primary temporal separation is between the decision on offer 1 and the revelation/receipt of offer

> Response to the above comment:

Thank you very much for pointing out this important issue. We agree that our original description was misleading, as the option evaluation, decision commitment, and motor execution processes cannot be demonstrated to be temporally separated in our task. Instead, a key advantage of the task design is that these processes occur in close succession during the presentation of the first object. This temporal structure allows us to investigate not only the neural mechanisms underlying each process but also the transitions between them. To address this concern, we have removed sentences referring to temporal separation and revised the text to emphasize the advantage of our task as follows (new line 174-177):

“In this task, option valuation, action selection, and action execution occurred in close succession during the presentation of the first object, providing an opportunity to examine the neural mechanisms underlying each process and the transitions between them.”

2. The manuscript would benefit from a more explicit model of the cognitive operations underlying the monkey's choice. The decision to accept/reject offer 1 is presumably based on a comparison of its value, $V(\text{offer 1})$, against an expected value of offer 2, $E[V(\text{offer 2})]$, which is learned from past experience. The paper does not detail how $E[V(\text{offer 2})]$ is operationalized or assumed to be updated. This is crucial because VS activity might reflect components of this expectation.

> Response to the above comment:

We thank the reviewer for this insightful comment. We agree that the monkey's decision to accept or reject the first object (offer 1) is likely based on a comparison between its value, $V(\text{offer 1})$, and an expected value of the second object, $E[V(\text{offer 2})]$, which may be formed from past experience. Indeed, our behavioral analyses revealed that the value of the second object in the "previous trial" influenced the monkey's choice behavior in a manner consistent with such comparison. The corresponding description has been added to the revised manuscript as follows (new line 157-169):

"We also found that the monkey's decision was influenced by recent experience. Specifically, the value of the second object in the previous trial negatively predicted whether the monkey would choose the first object in the current trial. A logistic regression analysis revealed a weak but significantly negative regression coefficient between the value of the previous second object and the choice rate of the current first object ($n = 143$ sessions, $z = -7.2$, $p = 8.4 \times 10^{-13}$; two-tailed Wilcoxon signed-rank test), while the regression coefficient for the current first object value was significantly positive ($n = 143$ sessions, $z = 10.4$, $p = 3.2 \times 10^{-25}$; two-tailed Wilcoxon signed-rank test) (Supplementary Fig. 1b). Thus, the higher the value of the previous second object, the less likely the monkeys were to choose the current first object. This opposing effect of the current and previous values suggests that when the previous second object had been highly rewarding, the monkeys anticipated the possibility of another valuable second object and were therefore more inclined to reject the current first object."

To further examine whether the ventral striatum represented this previous second object value during the decision-making period in the current trial (i.e., during the presentation of the current first object), we performed a "ridge regression" analysis in which the firing rate of each neuron was predicted by the value of the previous second object, the value of the current first object, and whether the monkey chose this first object. We used ridge regression because these predictors were correlated, making it statistically inappropriate to include them in a standard multiple regression model. Under such correlated conditions, regression coefficients cannot be reliably estimated, as any apparent variance partitioning between predictors becomes unstable due to "multicollinearity". Ridge regression introduces a small regularization term to the loss function, which stabilizes coefficient estimation and allows partial, but not perfect, mitigation of this issue. Therefore, this analysis should be regarded as a complementary validation rather than a primary variance-partitioning test.

The results showed that, while many ventral striatum neurons significantly encoded the current first object value and the monkey's choice (as expected from our

original analyses), only a few neurons represented the previous second object value. The corresponding description has been added to the revised manuscript as follows (new line 322-347):

“To understand the neural mechanisms underlying option valuation, action selection, and the transition between them, we have analyzed neuronal activity encoding the value of the first object and whether the monkey decided to release the button to choose this object during the decision-making period (i.e., during the presentation of the first object). However, other neural processes contributing to decision formation might also occur in the ventral striatum during the same period. For example, we observed that not only the value of the first object, but also the value of the second object in the previous trial influenced the monkey’s action selection (Supplementary Fig. 1b), suggesting that valuation or maintenance of the previous second object might also take place in the ventral striatum during decision making. To test this possibility, we conducted another ridge regression analysis in which the firing rate of each neuron was predicted by the value of the previous second object (previous value), as well as the value of the current first object (current value) and whether the monkey chose the first object (choice). As expected from the model comparison analysis, the proportions of neurons with significant regression coefficients for current value and choice increased after the first object onset and became significantly higher than chance ($p < 0.05$, two-tailed Monte Carlo test) (Supplementary Fig. 7a and 7b). On the other hand, the proportion of neurons with a significant regression coefficient for previous value was close to the chance level over time, slightly above for some time windows ($p < 0.05$; two-tailed Monte Carlo test) but below for others ($p > 0.05$; two-tailed Monte Carlo test) (Supplementary Fig. 7c), suggesting that the ventral striatum may not strongly represent or maintain the value of the previous second object. However, at the behavioral level, the influence of the previous second object value on the monkey’s choices was much weaker than that of the current first object value. Therefore, it may be difficult to detect neuronal activity corresponding to such a weak behavioral effect of the previous second object value in the ventral striatum.”

These analyses also address the points raised in Comment 3 below.

3. To better understand how $E[V(\text{offer } 2)]$ might be formed and whether it's encoded, the authors could test if $V(\text{offer } 2)$ from trial $n-1$ influences choice behavior for offer 1 in trial n (e.g., by splitting trials based on previous $V(\text{offer } 2)$ being high vs. low) and if this variable is encoded in VS activity during the offer 1 period of trial n .

> Response to the above comment:

We thank the reviewer for this constructive suggestion. We have indeed examined whether the value of the second object in the previous trial (previous $V(\text{offer } 2)$) influenced the monkey's choice behavior and whether this information was represented in ventral striatal activity during the offer 1 period (i.e., the presentation of the first object) of the current trial. The detailed results are presented in our response to Comment 2 above.

Briefly, a logistic regression analysis showed that the previous second object value negatively predicted the probability of choosing the current first object, indicating that monkeys anticipated the potential value of a future second object based on past experience. At the neural level, a ridge regression analysis—including the value of the second object in the previous trial, the value of the first object in the current trial, and the monkey's choice—revealed that while many neurons significantly encoded the current first object value and the choice, only a small proportion (near chance level) encoded the previous second object value. Please refer to the response to Comment 2 for further details and interpretation.

4. Consequently, during the offer 1 period, VS neurons could be encoding multiple variables beyond just $V(\text{offer } 1)$ and the impending choice. These potentially include: $E[V(\text{offer } 2)]$ (possibly influenced by $V(\text{offer } 2)$ from trial $n-1$ or a running average), the decision variable itself (e.g., $V(\text{offer } 1) - E[V(\text{offer } 2)]$), and motor preparation signals related to holding or releasing the button. This complexity makes the unique attribution of neural activity changes and stimulation effects to a single cognitive component challenging.

> Response to the above comment:

We appreciate the reviewer's thoughtful comment regarding the potential complexity of neural representations during the offer 1 period. We fully agree that ventral striatal (VS) neurons could, in principle, encode multiple decision-related variables such as $V(\text{offer } 1)$, $E[V(\text{offer } 2)]$, the decision variable $V(\text{offer } 1) - E[V(\text{offer } 2)]$, and motor preparation signals related to the forthcoming action. However, in our dataset, these variables were highly correlated with one another. Specifically, $V(\text{offer } 1)$, $E[V(\text{offer } 2)]$ (approximated by $V(\text{offer } 2)$ from trial $n-1$), and the decision variable $V(\text{offer } 1) - E[V(\text{offer } 2)]$ are mathematically and empirically interdependent, making it statistically inappropriate to include all of them simultaneously in a single regression model due to severe multicollinearity.

Given this limitation, we adopted a complementary approach. As described in our responses to Comments 2 and 3 above, we examined whether the value of the previous second object $V(\text{offer } 2)_{n-1}$ influenced both behavior and neuronal activity. Although behavioral analyses revealed a weak but significant negative effect of $V(\text{offer } 2)_{n-1}$ on the current choice, neural analyses showed that only a very small proportion of neurons encoded this variable, with the proportion remaining close to chance level. Based on this result, it is reasonable to expect that the representation of the comparison signal $V(\text{offer } 1) - E[V(\text{offer } 2)]$ would also be weak in the ventral striatum, since its computation depends on the reliable encoding of both terms. Indeed, when we explicitly tested this possibility using a separate regression model in which the firing rate of each neuron was predicted only by $V(\text{offer } 1) - E[V(\text{offer } 2)]$, only 9 out of 125 neurons (7.2%) showed significant regression coefficients, a proportion not significantly different from chance ($p > 0.05$, two-tailed Monte Carlo test). These results suggest that ventral striatum neurons primarily encode the current object value and choice-related activity, rather than an explicit decision variable computed as $V(\text{offer } 1) - E[V(\text{offer } 2)]$.

As seen in the response to Comments 2 and 3 above, we described that ventral striatum neurons only weakly (if any) represented $E[V(\text{offer } 2)]$ (approximated by $V(\text{offer } 2)$ from trial $n-1$). But we did not show the data from the regression model in which the firing rate of each neuron was predicted only by $V(\text{offer } 1) - E[V(\text{offer } 2)]$, because the results can be expected from the weak representation of $E[V(\text{offer } 2)]$. However, if the reviewer considers this additional analysis important for clarity, we would be happy to include it in the revised manuscript.

Furthermore, we examined whether ventral striatum activity reflected motor preparation signals, such as the latency of button release (reaction time) when choosing the first object (we have already analyzed the effect of holding or releasing the button on neuronal activity in Fig. 3). Using a ridge regression model that included the current object value and button-release latency as predictors, we found that the proportion of neurons with significant regression coefficients for latency was close to the chance level across time (Supplementary Fig. 5b). These results indicate that the ventral striatum activity does not reflect motor preparation or execution. We described these results in the revised manuscript as follows (new line 255-270):

“Using another ridge regression, we tested whether the modulation of each neuron could be accounted for by the variation in reaction time (i.e., the latency of button release). Here, the firing rate of each neuron was predicted by the value of the first object (value) and the button-release latency (latency), which were also correlated (Fig. 1c). We excluded the

predictor of whether the monkey released the button to choose the first object (choice) from this analysis, because the button-release latency existed only when the monkey released the button (i.e., when the monkey did not release the button, making it impossible to include such trials in the regression analysis). Whereas the proportion of neurons with a significant regression coefficient for value was significantly higher than chance for all calculation time windows (each 150-ms window after the first object onset) except immediately after the onset ($p > 0.05$; two-tailed Monte Carlo test) (Supplementary Fig. 5a), the proportion of neurons with a significant regression coefficient for latency was very close to chance over time, slightly above for some time windows ($p < 0.05$; two-tailed Monte Carlo test) but below for others ($p > 0.05$; two-tailed Monte Carlo test) (Supplementary Fig. 5b). These results suggest that the influence of button-release latency on neural activity, if any, was minimal.”

Together, these analyses address the reviewer’s concern and support the conclusion that ventral striatum neurons predominantly represent value- and choice-related signals during decision formation, with minimal influence from prior value expectations or motor-related factors.

5. The authors use a 150ms sliding window with a 1ms step to classify neural response modulations (value, choice, intermediate). For a typical analysis window (e.g., 500ms or more), this results in a large number of statistical tests per neuron. It is crucial to clarify if and how corrections for multiple comparisons (e.g., Bonferroni or FDR) were applied. Without such corrections, the reported proportions of neuron types could be inflated.

> Response to the above comment:

As the reviewer correctly pointed out, we used a 150-ms sliding window with a 1-ms step and tested statistical significance for each window using a threshold of $p < 0.05$. This method was chosen to examine the temporal dynamics of neuronal activity, specifically, when and how each neuron exhibited different types of modulation.

For this analysis, we further applied an additional criterion requiring that at least 40 consecutive windows exhibit the same type of significant modulation (i.e., value-type, choice-type, or intermediate-type modulation). This consecutive-window criterion effectively reduces the likelihood of false positives that could arise from multiple comparisons across overlapping windows. Similar approaches have been adopted in many previous studies (e.g., Grabenhorst and Báez-Mendoza, Nat. Commun., 2025), where consecutive significant windows are treated as a cluster-based correction method.

We note that the rationale behind this approach is that adjacent sliding windows are not statistically independent, because neural activity typically changes gradually over time. Consequently, applying conventional multiple-comparison corrections such as Bonferroni or FDR, which assume statistical independence among tests, would be overly conservative and substantially increase the likelihood of false negatives.

To emphasize ‘40 consecutive windows’, we revised the original manuscript as follows (new line 991-1004):

“We performed this model comparison for each of 150-ms sliding window with a 1-ms step. If a neuron showed a significantly higher R-square for the value model than the choice model ($p < 0.05$; two-tailed Monte Carlo test) for 40 consecutive windows and simultaneously exhibited a significant fit with the value model across the same 40 windows ($p < 0.05$; two-tailed F-test), the activity of this period was classified as a value-modulated signal. Conversely, if a neuron showed a significantly higher R-square for the choice model than the value model ($p < 0.05$; two-tailed Monte Carlo test) for 40 consecutive windows and simultaneously exhibited a significant fit with the choice model across the same 40 windows ($p < 0.05$; two-tailed F-test), the activity of this period was classified as a choice-modulated signal. If a neuron showed no significant difference in R-square between the two models but significant fit with both for 40 consecutive windows, the activity during this period was classified as an intermediate signal. Some neurons represented two or even three signals because their activity was classified as distinct signals during different time periods.”

6. As a robustness check and to complement the dynamic analysis, the authors should consider reporting results from analyses using a fixed, a priori defined time window for value encoding (e.g., early after offer 1 onset) and choice encoding (e.g., later, pre-response). This is common practice and can help validate the sliding window findings.

> Response to the above comment:

We appreciate the reviewer’s valuable suggestion. As a robustness check to complement our dynamic analysis, we conducted an additional analysis using fixed, a priori–defined time windows. Specifically, we divided the 1000-ms period of the first object presentation into four equal quadrants (0–250, 250–500, 500–750, and 750–1000 ms) and applied the same model comparison procedure to each window. This allowed us to classify recorded neurons as value-modulated, intermediate, or choice-modulated within each fixed time window. Please see the figure below, in which the

proportions of value-modulated, intermediate, and choice-modulated neurons are shown for each quadrant.

The proportion of value-modulated neurons was the largest in the 0–250 ms window and decreased thereafter, whereas the proportion of intermediate neurons peaked in the 250–500 ms window and then declined. The proportion of choice-modulated neurons gradually increased and became dominant in the 500–750 ms window. These temporal patterns, obtained using the fixed time windows, consistently reproduced the value-to-choice transition observed in our original sliding-window analysis, supporting the robustness of our findings.

While we believe this additional analysis validates the main results, we feel that including it in the manuscript would be somewhat redundant, given the already extensive supplementary materials (18 supplementary figures). However, if the reviewer considers this additional analysis important for clarity, we would be happy to include it in the revised manuscript.

7. The text sometimes refers to classifying "neurons" (e.g., "value-modulated neurons"). It's more precise to state that specific responses or epochs of activity within a neuron's firing pattern are classified, as a single neuron can show different types of modulation over time, which the authors indeed report.

> Response to the above comment:

We fully agree with the reviewer’s point. In the revised manuscript, we now classify neuronal activity, not the neurons themselves, into value-modulated, intermediate, or choice-modulated signals, as follows (new line 216-234):

“If neurons showed a significantly better fit with the value model than the choice model, their activity was considered to be more largely modulated by the value of the first object ($p < 0.05$; two-tailed Monte Carlo test) (blue area in Fig. 2e, hereafter called “value-modulated” signal). If neurons showed a significantly better fit with the choice model than the value model, their activity was considered to be more largely modulated by the monkey’s action selection ($p < 0.05$; two-tailed Monte Carlo test) (red area in Fig. 2e, hereafter called “choice-modulated” signal). If neurons did not show a significantly better fit with either model ($p > 0.05$; two-tailed Monte Carlo test) but showed significant fits with both models ($p < 0.05$; two-tailed F test), their activity was considered to exhibit an intermediate modulation between the value and the choice models (cyan area in Fig. 2e, hereafter called “intermediate” signal). To examine the temporal profile of neuronal activity, this model comparison analysis was conducted throughout the presentation of the first object using a 150-ms sliding window with a 1-ms step. Consequently, we identified 42 neurons with the value-modulated signal, 56 neurons with the choice-modulated signal, and 81 neurons with the intermediate signal (Fig. 2f, see also Supplementary Fig. 3a and 3b for individual monkeys). Notably, some of the recorded neurons exhibited two or three types of signals because these neurons represented distinct signals for different periods during the object presentation.”

We have revised all relevant descriptions throughout the manuscript to consistently refer to neuronal activity or signals, rather than to neurons themselves.

8. The bidirectional effects (increasing or decreasing choice probability) are reasonably attributed to potential mixed activation of direct and indirect pathway neurons. This remains speculative without cell-type specific manipulations.

> Response to the above comment:

We agree with the reviewer that the interpretation regarding mixed activation of direct and indirect pathway neurons remains speculative without cell-type-specific manipulations. To clarify this point, we have added the following description to the relevant paragraph to acknowledge this limitation and to highlight directions for future research, as follows (new line 667-672):

“Future studies are called for to directly test this interpretation using cell-type-specific manipulations. Although optogenetic or chemogenetic approaches have enabled selective

activation of direct- and indirect-pathway neurons in rodents⁵⁵, these techniques have not yet been fully established in non-human primates. The development of such cell-type-specific tools in primates will be essential for causally identifying the contributions of these pathways to action selection.”

9. The bidirectional effects here are more puzzling. Dopamine typically excites D1-expressing direct pathway neurons and inhibits D2-expressing indirect pathway neurons, both actions generally promoting approach. The manuscript's discussion of differential modulation of distinct VS populations is a starting point, but this warrants deeper exploration or acknowledgment of its unresolved nature. Could it relate to different baseline states, targeted VS subregions, non-canonical dopamine effects at the stimulation parameters used, or interaction with the complex encoding during offer 1 (see point 4)?

> Response to the above comment:

We thank the reviewer for this insightful comment. We agree that the bidirectional behavioral effects of optogenetic facilitation of dopamine inputs are indeed puzzling and cannot be fully explained by canonical dopaminergic mechanisms. In the revised manuscript, we have expanded the discussion to explore potential explanations and to explicitly acknowledge the unresolved nature of these effects. The revised paragraph now reads as follows (new line 704-724):

“... Taken together, the dopamine inputs to both direct and indirect pathway neurons might likely increase the animal's trend to approach motivational stimuli.

Nevertheless, our results indicate that optogenetic facilitation of dopamine input to the ventral striatum produced bidirectional behavioral effects. This apparent contradiction suggests that the functional consequences of dopamine input may not be uniform across ventral striatal circuits. Several non-mutually exclusive mechanisms could underlie this heterogeneity. First, the effects of dopamine input might depend on the baseline state of the local network, such that the same dopaminergic facilitation could either enhance or suppress neuronal firing depending on ongoing activity patterns. Second, subtle differences in the subregions of the ventral striatum targeted by optogenetic stimulation (e.g., shell versus core) might contribute to the divergent behavioral effects. Third, non-canonical effects of dopamine under the specific stimulation parameters used, such as spillover activation of other receptor types or modulation of presynaptic terminals, cannot be excluded. Finally, the optogenetic manipulation was applied during the presentation of the first object, when ventral striatum neurons encode multiple signals related to value and choice. The dopaminergic facilitation

might have interacted with these complex, time-varying representations, leading to variable behavioral consequences. Future studies combining cell-type-specific and temporally precise manipulations with simultaneous neuronal recordings will be essential to elucidate how dopamine input dynamically modulates value- and choice-related signals in the ventral striatum.”

10. Given the multiple cognitive and motor-related processes potentially occurring during offer 1 presentation (as noted in point 4), interpreting precisely which aspect of the decision process is being altered by electrical or optogenetic stimulation is challenging.

> Response to the above comment:

We fully agree with the reviewer that multiple cognitive and motor-related processes likely occur during the presentation of the first object, making it challenging to determine which specific component of the decision-making process was directly affected by electrical or optogenetic stimulation. To address this point, we have revised the discussion to explicitly acknowledge this limitation and clarify our interpretation, as follows (page 764-776):

“It should be noted again that our electrophysiological recordings suggest that the ventral striatum contributes not only to option valuation and action selection, but also to integrating these processes as a neural hub. The ventral striatum may also participate in other processes essential for decision-making. Therefore, although electrical stimulation and optogenetic facilitation of dopamine inputs during the decision-making period altered the monkey’s action selection, it remains challenging to precisely determine which component of the decision-making process was directly affected by these manipulations. Given that multiple cognitive and motor-related processes occur during the presentation of the first object, disentangling their respective contributions will require experimental designs that can temporally and functionally separate these processes. Future studies combining targeted perturbations with simultaneous neuronal recordings will be essential for clarifying how ventral striatal signals causally contribute to each stage of the decision-making process.”

11. The simple button-release task is used to argue that VS choice signals are not purely motor. However, details regarding the reward structure in this control task are missing. If the reward magnitude or expectancy in the control task significantly differed from the

decision-making task, it could influence baseline VS activity and the observed modulation strength, potentially confounding the comparison in Fig. 3c. This needs to be clarified.

> Response to the above comment:

We thank the reviewer for pointing out this important clarification. We fully agree that the difference in expected reward magnitude between the decision-making and control tasks could confound the comparison of neuronal modulation. To minimize this possibility, we used comparable reward magnitudes across the two tasks. Specifically, in the control task, the amount of reward per trial was set to be equivalent to that of value 3 or 4 in the decision-making task, which approximately matched the expected value of the decision-making task (3.5). Therefore, we believe that the stronger neuronal modulation observed in the decision-making task cannot be attributed to the difference in expected reward magnitude. We have added the following description to the Methods section (page 848-850):

“The reward amount in the control task was set to be equivalent to value 3 or 4 in the decision-making task, which approximately matched the expected reward magnitude in the decision-making task (3.5).”

We also added a discussion related to this issue as follows (new line 629-648):

“We confirmed that the choice-modulated signal discussed above was not simply caused by the motor action (i.e., button release) associated with choosing the first object, using the control task in which the monkey was required only to release the button. This control task was designed to eliminate the process of choice in order to isolate motor-related neuronal activity in the ventral striatum. However, it also differed from the decision-making task in other important respects. In the decision-making task, the identity and value of the first object presented as an option varied unpredictably across trials, introducing uncertainty about which object would appear and what reward could be obtained. This trial-by-trial uncertainty likely evoked reward expectation and prediction error processes that were absent in the control task, where the stimulus–reward association was fixed and fully predictable. Such differences in uncertainty and reward expectation may account for the stronger neuronal modulation observed during the decision-making task. In addition to these differences, we also note that the reward structure was comparable across tasks. The reward amount per trial in the control task was set to be equivalent to value 3 or 4 in the decision-making task, which matched the expected reward magnitude in the decision-making task (3.5). However, because the duration of each trial in the control task was shorter, the overall reward rate was actually higher, potentially leading to greater motivation. Therefore, the weaker neuronal modulation

observed in the control task is unlikely to reflect reduced motivation.”

Minor Comments :

1. Plotting trial-by-trial choice behavior for value 1 = 4 under stimulation vs. no-stimulation conditions could reveal finer-grained dynamics, such as adaptation or sequential dependencies in the stimulation effect.

> Response to the above comment:

We thank the reviewer for this helpful suggestion. Following the comment, we have now visualized the trial-by-trial choice behavior for first object value = 4 under stimulation and no-stimulation conditions as raster plots in the new Figures 4 and 5. These plots allow readers to observe the temporal dynamics of choice modulation, including potential adaptation or sequential dependencies related to stimulation.

2. For correlations that are "close to significance" (e.g., Fig S6c, $p=0.11$ for opto-stim effect on choice vs. motor latency), it's advisable to acknowledge the trend and be cautious about definitively concluding "no relationship," especially with modest sample sizes for correlational analyses.

> Response to the above comment:

We thank the reviewer for the helpful suggestion. In the revised manuscript, we have modified the description to acknowledge the modest negative trend in the correlation while avoiding an overly definitive statement of “no relationship.” Specifically, we now describe that although the correlation between the difference in button-release latency and the difference in choice rate of the first object with value of 4 did not reach statistical significance, a weak negative trend was observed. The revised text reads as follows (new line 527-535):

“Please note that there was no significant correlation between the difference in button-release latency and the difference in choice rate of the first object with value of 4, although the weak negative correlation was observed ($n = 50$ sites, $r = -0.23$, $p = 0.11$; two-tailed Pearson’s correlation test) (Supplementary Fig. 17c). This subtle trend indicates only a limited association, if any, between the effect of laser light stimulation on the motor action and the effect on the monkey’s action selection. These results therefore suggest that the

effect of laser light stimulation on dopamine projections on the monkey's action selection is unlikely to be fully explained by its impact on motor action.”

3. The manuscript should specify the time windows or criteria used to define a neuron's state (value, intermediate, choice) for calculating the transition probabilities shown in Fig 2i.

> Response to the above comment:

We appreciate the reviewer's insightful comment. The criteria and time windows used to define each neuronal state (value-modulated, choice-modulated, or intermediate) were originally described in the Methods section. However, we agree that this description could be made clearer. In the revised manuscript, we have rewritten this part to improve clarity and readability. Specifically, we now provide a concise explanation that each neuron's activity was statistically evaluated by comparing the fits of a value model and a choice model using R-square values obtained from 150-ms sliding windows with a 1-ms step, assessed by Monte Carlo and F-tests. Based on the results over 40 consecutive windows, neuronal activity was classified as value-modulated, choice-modulated, or intermediate. These criteria were also used to determine the latencies of each modulation type. The revised text now reads as follows (new line 991-1008):

“We performed this model comparison for each of 150-ms sliding window with a 1-ms step. If a neuron showed a significantly higher R-square for the value model than the choice model ($p < 0.05$; two-tailed Monte Carlo test) for 40 consecutive windows and simultaneously exhibited a significant fit with the value model across the same 40 windows ($p < 0.05$; two-tailed F-test), the activity of this period was classified as a value-modulated signal. Conversely, if a neuron showed a significantly higher R-square for the choice model than the value model ($p < 0.05$; two-tailed Monte Carlo test) for 40 consecutive windows and simultaneously exhibited a significant fit with the choice model across the same 40 windows ($p < 0.05$; two-tailed F-test), the activity of this period was classified as a choice-modulated signal. If a neuron showed no significant difference in R-square between the two models but significant fit with both for 40 consecutive windows, the activity during this period was classified as an intermediate signal. Some neurons represented two or even three signals because their activity was classified as distinct signals during different time periods.

The model comparison analysis was also used to determine the latencies of value-

modulated, intermediate, and choice-modulated signals. These latencies were defined as the start time of the first of the 40 consecutive 150-ms sliding windows with the 1-ms step where each modulation type was identified.”

4. The 50% transition probability between "value-modulated" and "intermediate" states (and vice-versa) warrants discussion. What does this bidirectionality imply about the underlying neural process compared to the more directed transitions towards "choice-modulated" states?

> Response to the above comment:

We appreciate the reviewer’s insightful comment regarding the bidirectional transition between the value-modulated and intermediate signals. We agree that this finding warrants clarification. In the revised manuscript, we now note that the equal transition probability between these two states suggests that they may not form a strictly sequential process, but could instead reflect bidirectional or overlapping computations during value evaluation. We also mention that such transitions are likely coordinated at the population level rather than within single neurons (we found that the latency of the value-modulated signal was shorter than that of the intermediate signal at the population level). The revised text now reads as follows (new line 300-304):

“The equal transition probability between the value-modulated and intermediate signals suggests that these two states may not form a strictly sequential process but could reflect bidirectional or overlapping computations during value valuation. This transition is therefore likely to be coordinated at the population level rather than within single neurons.”

5. The discussion acknowledges the role of prefrontal cortex (PFC) in transforming value to action. The findings beg the question of how VS computations are coordinated with PFC inputs. Is the VS transforming value signals received from regions like OFC/vmPFC, or is it part of a more iterative processing loop? A more nuanced discussion of these potential network interactions would enrich the paper.

> Response to the above comment:

We thank the reviewer for this insightful comment regarding the network-level interactions between the ventral striatum (VS) and the prefrontal cortex (PFC). In the

revised Discussion, we have expanded this section to incorporate a more nuanced consideration of how VS computations may be coordinated with cortical inputs. Specifically, we now note that both the VS and PFC are components of cortico-basal ganglia loop circuits, and that the nature of the signals exchanged between these regions remains poorly understood. We further emphasize that computational studies have proposed that cortico-basal ganglia loops implement more than simple feedforward transmission and instead support iterative and recurrent processing within the loop. This addition provides a framework in which the VS could both transform value signals received from cortical regions such as the OFC/vmPFC and actively participate in iterative value–choice computations. We also added that future studies should investigate these interactions by simultaneously recording from both regions or by causally perturbing their communication. The revised text reads as follows (new line 581-608):

“The neural dynamics we observed in the ventral striatum emphasizes its role as a neural hub that bridges option valuation and action selection. Previous studies have also explored the neural mechanism that bridges option valuation and action selection. These studies mainly dealt with the prefrontal and parietal cortical areas³²⁻³⁴. For example, it has been shown that option value signals are encoded in the ventromedial prefrontal cortex and then transmitted to the dorsomedial prefrontal cortex and the intraparietal sulcus area, where neuronal activity is involved in action selection by comparing the value of each option³⁴. In contrast, we found that neurons in the ventral striatum are involved in both value signaling and action selection. Their activity gradually shifted from value representation to action selection encoding, suggesting that the ventral striatum may directly derive action selection from value information. At the same time, it is important to note that neurons in the orbitofrontal cortex have been reported to encode value- and choice-related variables in decision-making situation^{27,35}, and that similar value-to-choice transitions have been observed in the amygdala^{36,37}. Because both the orbitofrontal cortex and amygdala project directly to the ventral striatum^{38,39}, the signals we observed in the ventral striatum might in part be inherited from these upstream regions. Particularly, however, the relationship between the ventral striatum and prefrontal regions requires further considerations. Both regions participate in cortico-basal ganglia loop circuits, often referred to as the “limbic loop”^{40,41}, yet little is known about the nature of the signals exchanged between them. Computational studies have suggested that cortico-basal ganglia loop circuits, which sometimes include the dopamine system and amygdala, implement more than simple feedforward transmission of signals and highlighted the importance of iterative and recurrent processing within these circuits⁴²⁻⁴⁴. In light of our

findings on the dynamic transformation from value representation to action selection encoding in the ventral striatum, future studies should investigate how such iterative interactions between the ventral striatum and prefrontal regions contribute to decision-making, for example, through simultaneous recordings or causal perturbations of their communication.”

6. To further situate the current findings within the field, the authors may wish to consider additional relevant studies on primate ventral striatum (VS) function in reward processing and decision-making. For example, incorporating discussion of works such as: <https://doi.org/10.1523/JNEUROSCI.2430-24.2025> <https://doi.org/10.1016/j.neuron.2010.11.041>

> Response to the above comment:

We thank the reviewer for this insightful suggestion. In the revised manuscript, we have expanded the Discussion section to more comprehensively situate our findings within the broader literature on the primate ventral striatum (VS). Specifically, we now refer to and discuss the studies by Cai et al. (2011) Neuron (Ref 31 in our manuscript) and Iwaaki et al. (2025) J Neurosci (Ref 56 in our manuscript) as follows (new line 564-580 and 725-742, respectively):

“The ventral striatum has been widely recognized as a key structure for processing reward value information and regulating motivational states³⁻⁶. For example, a previous study demonstrated its role in representing temporally discounted values by integrating information about both reward magnitude and delay during decision-making³¹. Importantly, this study reported that ventral striatal neurons encoded the sum of discounted values across options, which may reflect a general motivational drive, whereas neurons in the dorsal striatum encoded the difference between these values, suggesting a more pivotal role for the dorsal striatum in action selection. In contrast, our present results indicate that the ventral striatum itself can also participate directly in action selection. The gradual transition we observed from the value-modulated to choice-modulated signals suggests that, beyond its established role in value integration, the ventral striatum may actively contribute to transforming option value information into action selection. However, to fully understand the division of labor between the ventral and dorsal striatum, it will be essential to examine their activity under identical behavioral conditions. Future studies that simultaneously record neuronal activity in both regions during the same decision-making task will be crucial for elucidating how these striatal subregions cooperate to translate value

into action.”

“Please note that, although both electrical stimulation of the ventral striatum and optogenetic facilitation of dopamine inputs to this region produced bidirectional effects on the monkey’s action selection, only the electrical stimulation exhibited a weak but significant anterior–posterior gradient. The positive effects tended to appear in the anterior portion, whereas the negative effects were more pronounced in the posterior portion. A recent pharmacological study in macaque monkeys also reported functional heterogeneity along this axis of the ventral striatum, showing that inactivation of its anterior portion induced hypoactive, “resting” behavior, whereas inactivation of the posterior portion elicited compulsive-like “checking” behaviors⁵⁶. Our results raise the possibility that such anterior–posterior functional differentiation may also extend to decision-related processes. However, it remains unclear why optogenetic facilitation of dopamine inputs did not exhibit a similar gradient. One possible explanation is that electrical stimulation and pharmacological manipulation directly modulate the cell body activity of ventral striatal neurons, whereas optogenetic facilitation affects dopamine release from axon terminals. Since dopamine release mainly modulates synaptic efficacy onto striatal neurons rather than their intrinsic excitability, the anterior–posterior gradient observed with electrical stimulation might reflect the influence of non-dopaminergic afferents that directly impact neuronal firing.”

Reviewer #5 (Remarks to the Author):

Reviewer #6 (Remarks to the Author):

Reviewer #1 (Remarks to the Author):

The authors have addressed all my comments.

> Response to the above comment

We appreciate your positive feedback and are glad that our revisions have addressed all of your comments.

Reviewer #2 (Remarks to the Author):

Thanks for all your work in addressing my concerns. I believe the manuscript has been significantly strengthened as a result. I have no major comments left and my minor comments below are optional small points.

> Response to the above comment

We appreciate your positive feedback and are glad that our revisions have addressed your concerns.

Related to my previous comment #3, I wondered if it would make sense to add the precautions to prevent light leaking into the experimental room during optogenetic stimulation to the Methods in one or two sentences (e.g. to mention the black tube & black sheet).

> Response to the above comment

We agree that describing these precautions is important for the reproducibility of the study. We have therefore added the following text to the Methods section (new lines 947–951):

“When the laser was simply turned on, light weakly leaked from the connection between the optrode and the optical fiber, which could potentially affect monkey behavior. To prevent this unintended leakage, we covered the connection with a black plastic tube and wrapped the entire assembly with a black opaque sheet. These precautions ensured that no light leakage was visible in the experimental room.”

I also think that the figure shown in response to my previous comment #4 showing transitions from value to intermediate and intermediate to choice neurons in terms of

latency could make an interesting supplementary figure. I leave it up to the authors to include it or not, according to their preference.

> Response to the above comment

We agree that readers may find the latency of transitions of interest. In response to the reviewer's suggestion, we have included this data as Supplementary Fig. 6 in the revised manuscript.

Reviewer #3 (Remarks to the Author):

The authors have adequately addressed all of my concerns, and I appreciate the inclusion of the additional analyses. The manuscript has improved and is now more coherently framed in light of the evidence provided. I have no further comments.

> Response to the above comment

We appreciate your positive feedback and are glad that our revisions have addressed all of your concerns.

Reviewer #4 (Remarks to the Author):

The authors have addressed all of my concerns with the original manuscript.

> Response to the above comment

We appreciate your positive feedback and are glad that our revisions have addressed all of your concerns.

Reviewer #5 (Remarks to the Author):

Reviewer #6 (Remarks to the Author):
